https://doi.org/10.1038/s41467-021-26131-z　**OPEN**

# mTOR-related synaptic pathology causes autism spectrum disorder-associated functional hyperconnectivity

Marco Pagani [1,2], Noemi Barsotti[3], Alice Bertero[1,3], Stavros Trakoshis[4,5], Laura Ulysse[6], Andrea Locarno[7], Ieva Miseviciute[7], Alessia De Felice[1], Carola Canella[1], Kaustubh Supekar[8], Alberto Galbusera [1], Vinod Menon[8], Raffaella Tonini [7], Gustavo Deco[6,9], Michael V. Lombardo [5,10,11], Massimo Pasqualetti [1,3,11] & Alessandro Gozzi [1,11 ✉]

Postmortem studies have revealed increased density of excitatory synapses in the brains of individuals with autism spectrum disorder (ASD), with a putative link to aberrant mTOR-dependent synaptic pruning. ASD is also characterized by atypical macroscale functional connectivity as measured with resting-state fMRI (rsfMRI). These observations raise the question of whether excess of synapses causes aberrant functional connectivity in ASD. Using rsfMRI, electrophysiology and in silico modelling in Tsc2 haploinsufficient mice, we show that mTOR-dependent increased spine density is associated with ASD-like stereotypies and cortico-striatal hyperconnectivity. These deficits are completely rescued by pharmacological inhibition of mTOR. Notably, we further demonstrate that children with idiopathic ASD exhibit analogous cortical-striatal hyperconnectivity, and document that this connectivity fingerprint is enriched for ASD-dysregulated genes interacting with mTOR or Tsc2. Finally, we show that the identified transcriptomic signature is predominantly expressed in a subset of children with autism, thereby defining a segregable autism subtype. Our findings causally link mTOR-related synaptic pathology to large-scale network aberrations, revealing a unifying multi-scale framework that mechanistically reconciles developmental synaptopathy and functional hyperconnectivity in autism.

[1] Functional Neuroimaging Laboratory, Istituto Italiano di Tecnologia, Center for Neuroscience and Cognitive Systems @ University of Trento, Rovereto, Italy. [2] Autism Center, Child Mind Institute, New York, NY, USA. [3] Department of Biology, Unit of Cell and Developmental Biology, University of Pisa, Pisa, Italy. [4] Department of Psychology, University of Cyprus, Nicosia, Cyprus. [5] Laboratory for Autism and Neurodevelopmental Disorders, Istituto Italiano di Tecnologia, Center for Neuroscience and Cognitive Systems @ University of Trento, Rovereto, Italy. [6] Center for Brain and Cognition, Computational Neuroscience Group, Department of Information and Communication Technologies, Barcelona, Spain. [7] Neuromodulation of Cortical and Subcortical Circuits Laboratory, Istituto Italiano di Tecnologia, Genova, Italy. [8] Stanford University, Stanford, CA, USA. [9] Institució Catalana de la Recerca i Estudis Avançats (ICREA), Universitat Pompeu Fabra, Barcelona, Spain. [10] Autism Research Centre, University of Cambridge, Cambridge, UK. [11] These authors jointly supervised this work: Michael V. Lombardo, Massimo Pasqualetti, Alessandro Gozzi. ✉email: alessandro.gozzi@iit.it

Recent advances in neurogenetics have begun to shed light on the complex etiology of autism spectrum disorders (ASD)[1]. Despite the daunting phenotypic and etiologic complexity that characterizes ASD[2], multiple syndromic forms of ASD have been found to encompass mutations in genes that affect translational control, protein synthesis, and the structure, transmission, or plasticity of synapses[3,4]. These observations have led to the hypothesis that dysfunctional synaptic pruning and homeostasis might contribute to ASD pathology[3,5]. Postmortem histological examinations support this notion, as the presence of increased dendritic spine density in brain tissue of ASD patients has been repeatedly observed[6–8]. Crucially, one such study[7] has suggested a link between postmortem dendritic synaptic surplus in ASD and hyperactivity of the mammalian target of rapamycin (mTOR) pathway[5]. mTOR is a key regulator of synaptic protein synthesis[9], and aberrations in mTOR signaling have been linked to synaptic and neuroanatomical abnormalities[10,11] that are associated with both syndromic (e.g., tuberous sclerosis[12,13]) and idiopathic ASD[14–16]. Further mechanistic investigations by Tang et al.[7], suggest that the observed excitatory spine surplus could be the result of mTOR-driven impaired macro-autophagy and deficient spine pruning, thus establishing a mechanistic link between a prominent signaling abnormality in ASD[12,16] and prevalent postmortem correlate of synaptic pathology in these disorders.

At the macroscopic level, several studies have highlighted the presence of aberrant functional connectivity in ASD as measured with resting-state fMRI (rsfMRI)[17–22], electroencephalography[23], or near-infrared spectroscopy[24]. These observations suggest that specific symptoms and clinical manifestations of ASD could at least partly reflect interareal brain synchronization[25,26]. Although the exact relationship between microscopic, mesoscopic, and large-scale network activity remains poorly understood, the linear integrative function served by dendritic spines[27] may be critical to the establishment of large-scale interareal coupling. Recent investigations of neuronal–microglia signaling in rodents support this notion, showing that developmental synaptic pruning critically governs the maturation of large-scale functional connectivity, biasing ASD-relevant sensory-motor and socio-communicative behavior[28]. These observations point at a putative etiopathological link between translational control, synaptic pathophysiology, and brain-wide dysconnectivity, raising the question of whether dendritic spine abnormalities that characterize ASD (e.g., via mTOR hyperactivity[7]) could primarily cause functional connectivity aberrancies relevant for these disorders. If this was the case, prior conceptualizations of ASD as a "developmental synaptopathy"[29] or "dysconnectivity syndrome"[30] could be reconciled within a unique multiscale framework.

Here, we probe a causal mechanistic link between mTOR-related synaptic pathology and aberrant functional connectivity alterations in ASD. We test this hypothesis by combining rsfMRI, viral tracing, in silico modeling, morphological and electrophysiological recordings in haploinsufficient tuberous sclerosis complex 2 mice (Tsc2[+/−]), a mouse line mechanistically reconstituting mTOR-dependent synaptic surplus observed in postmortem ASD investigations[7]. We show that mTOR-dependent increased spine density is associated with motor stereotypies and functional hyperconnectivity in evolutionarily conserved associative networks and that these dysfunctions can be completely rescued by pharmacological inhibition of mTOR activity. In keeping with the key involvement of mTOR-pathway dysfunction in idiopathic ASD[7,14,16], we further show that a similar connectivity fingerprint can be identified in rsfMRI scans of a subset of idiopathic ASD cases, and document that this hyperconnectivity pattern is enriched in genes that are dysregulated in the cortical transcriptome of ASD and interact with mTOR and Tsc2

at the protein level. Our work establishes a mechanistic link between mTOR-related synaptic pathology and functional hyperconnectivity in ASD, and defines a multiscale cross-species platform enabling the decoding of ASD-specific pathophysiological traits from connectomics alterations in ASD populations.

## Results

**Increased spine density and cortico-limbic hyperconnectivity in Tsc2[+/−] mice.** Previous research has linked synaptic surplus in ASD to aberrant mTOR signaling, showing that mTOR over-activity in Tsc2-deficient mice mechanistically recapitulates the increased synaptic density observed in postmortem ASD brain tissue[7]. To probe whether ASD-relevant mTOR-dependent synaptic pathology is associated with a specific brain-wide signature of functional dysconnectivity, we carried out rsfMRI connectivity mapping in Tsc2[+/−] mutant mice. To ascertain the presence of synaptic alterations in Tsc2[+/−] mutants, we first measured dendritic spine density in the insular cortex, a region relevant to social dysfunction in autism[21,31], and found that Tsc2[+/−] mice exhibit higher spine density compared to control littermates ($P = 0.021$, Fig. 1a and $P < 0.001$, Supplementary Fig. S1). We next used rsfMRI to map brain-wide functional connectivity in juvenile (P28) Tsc2[+/−] mutants. The use of pre-pubertal mice allowed us to identify a connectivity signature unaffected by puberty-induced synaptic and network remodeling[32], which would as such be more indicative of the network states that characterize early developmental pathology in ASD.

Global connectivity (GC) analysis revealed prominent foci of increased connectivity in the PFC and insular cortex of Tsc2[+/−] mice (Fig. 1b). The regions affected are core nodes of the mouse default mode and salience networks, two evolutionarily conserved systems[33] that have been widely implicated in ASD[21,31]. To identify the target regions associated with these connectivity changes, we next carried out network probing of PFC and in the insular cortex by using seed-based mapping. This analysis revealed that Tsc2[+/−] mice exhibit functional over-synchronization between the prefrontal cortex and the posterior cingulate, anterior insula, and cortical–striatal components of the DMN (Fig. 1c). Similarly, the anterior insula was oversynchronized with prefrontal regions and the retrosplenial cortex (Fig. 1d). These findings show that mTOR-dependent synaptic alterations in Tsc2[+/−] mutants are associated with a distinctive functional hyperconnectivity signature encompassing translationally relevant integrative networks.

**mTOR-related functional hyperconnectivity can be modeled in silico by increasing synaptic coupling.** Previous studies have revealed that macroscale white matter rearrangement[34] or mesoscale axonal rewiring[35,36] could lead to brain-wide rsfMRI dysconnectivity. To rule out a meso- or macrostructural origin for the observed rsfMRI changes, we first used diffusion tensor imaging to map macroscale white matter organization in the same cohort of juvenile mice undergoing rsfMRI mapping. Voxelwise and regional assessments of fractional anisotropy (FA), a parameter sensitive to microscale white matter integrity[35], revealed the presence of largely preserved microstructure in all the major fiber tracts of Tsc2[+/−] mutants ($q > 0.24$, FDR-corrected, Fig. 2a). The lack of regional FA differences also argues against the presence of major alterations in whole-brain white matter topography, as these would be appreciable in the form of large regional FA differences[37]. To rule out the presence of more subtle myelination deficits in Tsc2[+/−] mice, we carried out measurements of myelin basic protein (MBP) density and coherency[38] using immunostaining methods. In keeping with our FA measurements, MBP staining did not show any genotype-dependent alterations in the corpus callosum or in gray-matter

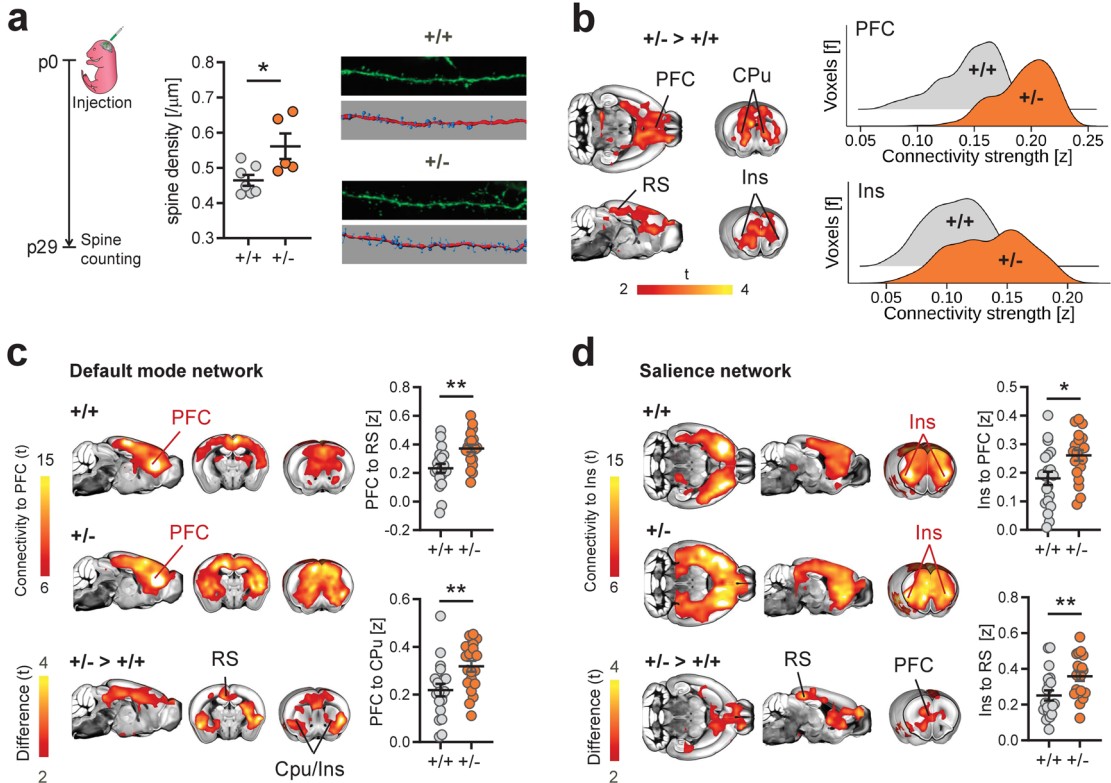

**Fig. 1 Dendritic spine surplus is associated with increased rsfMRI hyperconnectivity in $Tsc2^{+/-}$ mice. a** Experimental design of spine density measurements. Intergroup comparisons showed a surplus of dendritic spine density in $Tsc2^{+/-}$ $(+/-)$ mutant mice. Representative confocal images of Layer 5 dendritic spines in $+/+$ and $+/-$ mice are also reported (unpaired $t$ test, two-sided, $t = 2.72$, $P = 0.021$. $n = 7$ $(+/+)$ and $n = 5$ $(+/-)$ mice). **b** Voxelwise rsfMRI mapping revealed widespread increases in long-range connectivity in cortico-limbic and striatal areas of $Tsc2^{+/-}$ mice (red–yellow, left). Global histogram analysis confirmed these findings, revealing a marked increase in the number of voxels showing stronger long-range rsfMRI connectivity in prefrontal and insular regions of $Tsc2^{+/-}$ mice (right). **c** Spatial extension of the mouse default-mode network as probed using a seed region in the prefrontal cortex (PFC). Between-group comparisons (bottom) revealed prominent prefronto-cortical and striatal hyperconnectivity in $Tsc2^{+/-}$ mice as compared to control littermates (red–yellow, bottom). Regional quantifications of this effects confirmed increased prefrontal rsfMRI connectivity with retrosplenial (RS, unpaired $t$ test, two-sided, $t = 3.40$, $P = 0.002$) and striatal (CPu, unpaired $t$ test, two-sided, $t = 2.88$, $P = 0.006$) areas in $Tsc2^{+/-}$ mice. **d** Spatial extension of the mouse salience network as probed using a seed region in the anterior insular cortex (Ins). Between-group comparisons revealed foci of increased connectivity in prefrontal (PFC, unpaired $t$ test, two-sided, $t = 2.69$, $P = 0.011$) and retrosplenial cortices (RS, unpaired $t$ test, two-sided, $t = 2.70$, $P = 0.009$) of $Tsc2^{+/-}$ mice (red–yellow, bottom). Regional quantifications of these effects confirmed increased connectivity between anterior insula and prefrontal and retrosplenial cortices in $Tsc2^{+/-}$ mice. Error bars represent SEM. Cpu striatum, Ins insular cortex, PFC prefrontal cortex, RS retrosplenial cortex. $*P < 0.05$, $**P < 0.01$. $n = 20$ mice for each group in panel (**b**), (**c**), and (**d**). Source data are provided as a Source Data file.

regions (PFC, hippocampus) as assessed with densitometric quantifications and MBP coherency mapping (Supplementary Fig. S2). These results support the lack of gross micro- and microstructural white matter (WM) reorganization in these mice. To probe axonal mesoscale structure in $Tsc2^{+/-}$ mutants, we next carried out retrograde axonal tracing using recombinant rabies virus[36] in the medial prefrontal cortex, a region exhibiting prominent rsfMRI hyperconnectivity in $Tsc2^{+/-}$ mutants. Quantification of retrogradely labeled cells in neocortical, striatal, and sub-thalamic areas revealed largely comparable projection frequency in all the examined regions across genotypes ($q > 0.77$, FDR-corrected, Fig. 2b and Supplementary Fig. S3). Furthermore, in keeping with prior investigations[39], high-resolution anatomical imaging did not reveal any major morphological changes in the brain of $Tsc2^{+/-}$ mutants (Supplementary Fig. S4). Taken together, these results argue against the presence of gross macro- and mesoscale structural or axonal abnormalities in $Tsc2^{+/-}$ mice.

The lack of major structural alterations predictive of the observed rsfMRI hyperconnectivity points at a possible mechanistic link between mTOR-related spine surplus and increased rsfMRI coupling in $Tsc2^{+/-}$ mutants. To test this hypothesis, we first probed spine functionality in $Tsc2^{+/-}$ mutants using

intracellular electrophysiological recordings, under the assumption that only nonsilent, functionally mature synapses could contribute to the establishment of aberrant interareal coupling[40]. We thus measured the ratio between AMPA and NMDA excitatory postsynaptic currents in neurons from the prefrontal cortex and caudate putamen, two brain regions showing prominent rsfMRI hyperconnectivity in $Tsc2^{+/-}$ mutants. AMPA/NMDA ratio is a metric sensitive to synaptic maturation, and prior research has shown that dendritic spine pruning and circuital refinement during early development results in a steady increase in AMPA/NMDA ratio reflecting the removal of immature or silent synapses[41,42]. Our measurements revealed that the synaptic AMPA/NMDA ratio was broadly comparable across genotypes in both the examined regions (PFC: $P = 0.43$, CPU: $P = 0.97$, Fig. 2c), suggesting that most dendritic spines in $Tsc2^{+/-}$ mutants are functionally mature (i.e., non silent).

The absence of conspicuous anatomical rewiring, along with the presence of largely preserved synaptic maturation in $Tsc2^{+/-}$ mice support a theoretical model in which macroscale interareal functional hyperconnectivity could reflect an increased integration of excitatory input mediated by over-abundant synapses, resulting in large-scale over-synchronization mediated by locally

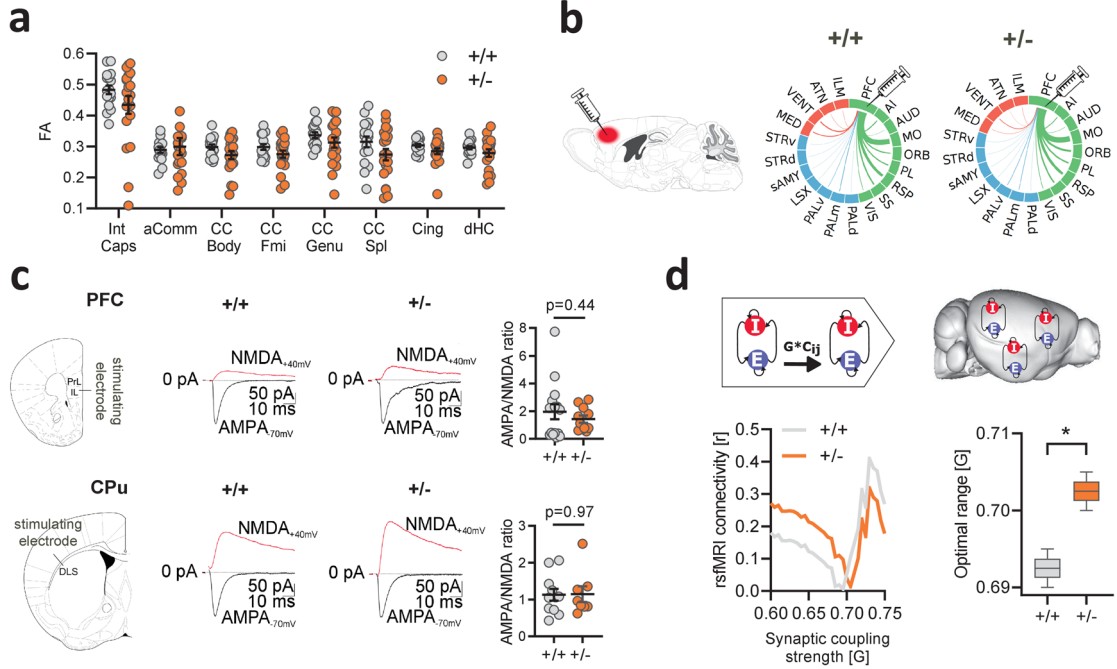

**Fig. 2 Increased synaptic coupling explains functional hyperconnectivity in Tsc2$^{+/-}$ mice. a** Fractional anisotropy (FA) quantification in the anterior commissure, corpus callosum, cingulate, dorsal hippocampus, and internal capsule as assessed with diffusion-weighted imaging. Intergroup comparisons showed preserved FA in Tsc2$^{+/-}$ mutant mice with respect to control littermates. Int Caps internal capsule, aComm anterior commissure, CC Body body of the corpus callosum, CC Fmi forcep minor of the corpus callosum, CC Genu genu of the corpus callosum, CC Spl splenium of the corpus callosum, Cing cingulum, dHC dorsal hippocampus. n = 20 mice for each group. Error bars represent SEM. **b** Regional quantification of retrogradely labeled cells as probed with the injection of recombinant rabies virus in the prefrontal cortex. The thickness of the links in the circular layouts is proportional to the relative number of labeled cells. Cortical areas are depicted in green, subcortical regions in blue, and thalamic areas in red. No differences were observed between in Tsc2$^{+/-}$ and Tsc2$^{+/+}$ mice in any of the examined regions. PFC prefrontal cortex, AI agranular insular cortex, AUD auditory cortex, MO motor cortex, ORB orbitofrontal cortex, PL prelimbic cortex, RSP retrosplenial cortex, SS somatosensory cortex, VIS visual cortex, PALd globus pallidus (dorsal part), PALm globus pallidus (medial part), PALv globus pallidus (ventral part), LSX lateral septal complex, sAMY striatum-like amygdalar nuclei, STRd striatum (dorsal part), STRv striatum (ventral part), MED medial thalamus, VENT ventral thalamus, ATN anterior thalamus, ILM intralaminar thalamus. **c** Intracellular electrophysiological recordings in prefrontal and striatal neurons of Tsc2$^{+/-}$ mutant and control mice. Intergroup comparisons revealed comparable AMPA/NMDA ratio in Tsc2$^{+/-}$ and Tsc2$^{+/+}$ mice in both the regions probed (PFC, unpaired t test, two-sided, t = 0.79, P = 0.44; CPu, unpaired t test, two-sided, t = 0.07, P = 0.97). PFC prefrontal cortex, PrL prelimbic, IL infralimbic, CPu striatum, DLS dorsolateral striatum. PFC, n = 15 mice (+/+), n = 11 mice (+/−); CPu, n = 11 mice (+/+), n = 8 mice (+/−). Error bars represent SEM. **d** Whole-brain network modeling[44] was carried out to probe a putative mechanistic link between interareal coupling strength (G) to empirical measurements of whole-brain rsfMRI connectivity (left). Optimal G was selected on the basis of the minimal difference between modeled and empirical rsfMRI connectivity (y axis) for both genotypes (middle). Intergroup differences (permutation-based unpaired t-test, two-sided, $G_{diff}$ = 0.010, P = 0.030) revealed higher optimal G in Tsc2$^{+/-}$ vs. control mice (right). n = 20 mice for each group. Box and whiskers: Tsc2$^{+/+}$, min = 0.6900, max = 0.6950, centre = 0.6925, bounds of box and whiskers = 0.6912–0.6933, quartiles; Tsc2$^{+/-}$, min = 0.7000, max = 0.7050, centre = 0.7025, bounds of box and whiskers = 0.7012–0.7033, quartiles. *P < 0.05. Source data are provided as a Source Data file.

and remotely projecting excitatory cells. Such a model would be consistent with the established function of dendritic spine synapses as linear integrators of inputs in distributed cortical circuits[40]. The recent development of whole-brain computational models of rsfMRI connectivity[43,44] allowed us to test the theoretical validity of this assumption via in silico manipulations of macroscale interareal coupling strength. To this aim, we first established a whole-brain network model of rsfMRI connectivity using a high-resolution parcellation of the mouse brain connectome[44]. We next carried out in silico modeling at varying interareal coupling strength (parameter G) to identify the values that most accurately predicted our empirical rsfMRI measurements in control and Tsc2$^{+/-}$ mutant mice. In keeping with our hypothesis, we found that optimal coupling strength in Tsc2$^{+/-}$ mutants was significantly larger than the corresponding value in Tsc2$^{+/+}$ control animals (P = 0.03, Fig. 2d). This finding supports a putative association between mTOR-related synaptic overabundance and rsfMRI hyperconnectivity, suggesting that rsfMRI over-synchronization in high connection density components of

the DMN and salience networks[43] may emerge out of a generalized increase in interareal coupling strength.

**Pharmacological inhibition of mTOR rescues synaptic surplus and functional hyperconnectivity in Tsc2$^{+/-}$ mice.** If mTOR-related dendritic spine surplus is mechanistically implicated in the establishment of rsfMRI hyperconnectivity, pharmacological normalization of mTOR hyperactivity should rescue both synaptic overabundance and aberrant rsfMRI coupling. To test this hypothesis we pharmacologically treated Tsc2$^{+/-}$ and control mice with the mTOR inhibitor rapamycin during their fourth postnatal week as in ref. [7]. As predicted, quantification of dendritic spine density revealed that rapamycin treatment rescued spine density in Tsc2$^{+/-}$ mice to levels comparable to Tsc2$^{+/+}$ control mice (Fig. 3a and P < 0.001, Supplementary Fig. S1). Remarkably, rsfMRI mapping in rapamycin-treated Tsc2$^{+/-}$ mice revealed a complete rescue of the hyperconnectivity phenotype, entailing a marked reduction of long-range functional connectivity in the same neocortical and striatal regions the are

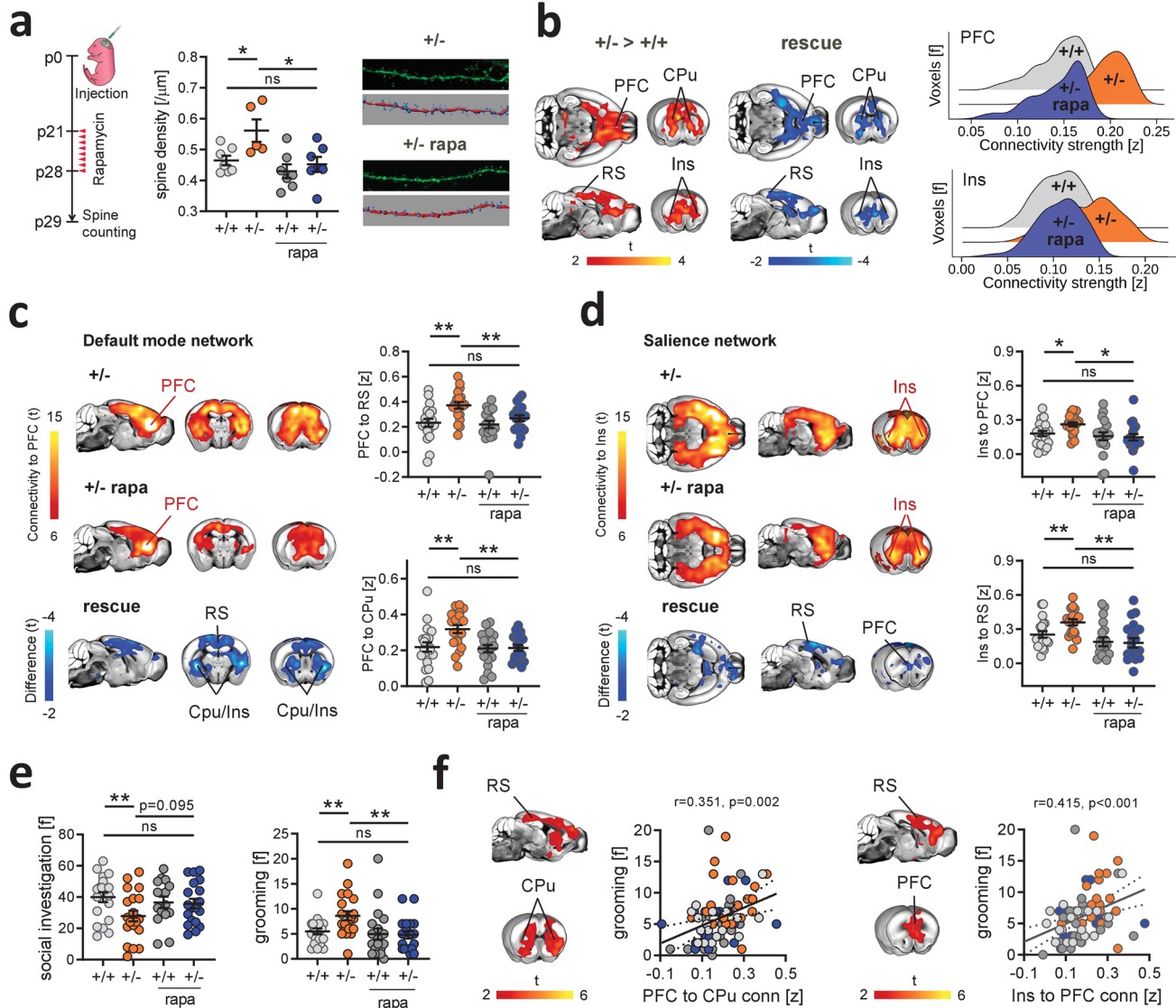

**Fig. 3 Pharmacological mTOR inhibition rescues synaptic surplus, functional hyperconnectivity, and autism-like behavior in $Tsc2^{+/-}$ mice.**
**a** Experimental design of spine density measurements (left). Developmental treatment with the mTOR inhibitor rapamycin normalized dendritic spine density in rapamycin-treated $Tsc2^{+/-}$ mice (+/− rapa) to the level of vehicle-treated $Tsc2^{+/+}$ control mice (+/+). $n = 7$ mice (+/+, +/− rapa, and +/+ rapa) and $n = 5$ mice (+/−). **b** Voxelwise rsfMRI mapping showed a pattern of decreased long-range functional connectivity in rapamycin-treated $Tsc2^{+/-}$ mice (blue), recapitulating the anatomical regions showing increased rsfMRI connectivity in $Tsc2^{+/-}$ mice (red–yellow coloring, left). Rescue here denotes the effect of rapamycin in $Tsc2^{+/-}$ mutants i.e., areas of decreased rsfMRI connectivity in rapamycin-treated vs. vehicle-treated $Tsc2^{+/-}$ mice. Global histogram analysis confirmed the rescue of long-range hyperconnectivity in prefrontal and insular regions of rapamycin-treated mutants (right). **c** Spatial extension of the mouse default-mode network in the vehicle- (top) and rapamycin- (bottom) treated $Tsc2^{+/-}$ mice. Between-group connectivity mapping and regional quantifications revealed that treatment with rapamycin rescued prefrontal rsfMRI hyperconnectivity in $Tsc2^{+/-}$ mice to levels comparable to control mice (right). $n = 20$ mice for each group. **d** Spatial extension of the salience network in the vehicle- (top) and rapamycin- (bottom) treated $Tsc2^{+/-}$ mice. Rapamycin completely rescued salience network hyperconnectivity in rapamycin-treated $Tsc2^{+/-}$ mice. Rescue in (**c**) and (**d**) denotes the effect of rapamycin in $Tsc2^{+/-}$ mutants i.e., areas of decreased rsfMRI connectivity in rapamycin-treated vs. vehicle-treated $Tsc2^{+/-}$ mice. $n = 20$ mice for each group. **e** Rapamycin also rescued altered self-grooming and social behaviors in $Tsc2^{+/-}$ rapa mice. Grooming, mice, $n = 20$ (+/+), $n = 20$ (+/−), $n = 19$ (+/+ rapa), $n = 19$ (+/− rapa). Social investigation, mice, $n = 20$ (+/+), $n = 20$ (+/−), $n = 15$ (+/+ rapa), $n = 19$ (+/− rapa). **f** Voxelwise correlation mapping revealed a significant positive correlation between prefrontal (left, Pearson's $r$, $r = 0.351$, $P = 0.002$) and insular (right, Pearson's $r$, $r = 0.415$, $P < 0.001$) rsfMRI connectivity, and motor stereotypes. PFC prefrontal cortex, CPu striatum, RS retrosplenial cortex, Ins insular cortex. Scatterplots represent the quantification of these correlations in the striatum and PFC, respectively. *$q < 0.05$, **$q < 0.01$, all FDR-corrected. Error bars represent SEM. Source data are provided as a Source Data file.

characterized by rsfMRI hyperconnectivity in vehicle-treated $Tsc2^{+/-}$ mutants (Fig. 3b). In keeping with this, rapamycin also rescued default-mode network (Fig. 3c) and salience network (Fig. 3d) rsfMRI connectivity in $Tsc2^{+/-}$-treated mice, as probed with prefrontal and anterior insular seed-based correlation analysis, respectively. These results corroborate the mechanistic

specificity of our findings, supporting a causal link between ASD-relevant mTOR-dependent synaptic pathology and rsfMRI hyperconnectivity in $Tsc2^{+/-}$ mice.

To assess whether these connectivity findings are behaviorally relevant, we next measured social behaviors and motor stereotypies in rapamycin and vehicle-treated $Tsc2^{+/-}$ mice using a

male–male interaction and self-grooming test, respectively. In keeping with previous findings[7,45], $Tsc2^{+/-}$ mutants exhibited significantly decreased social investigation and increased stereotypical grooming behavior, reconstituting two core ASD-like traits in mice. Notably, rapamycin treatment in $Tsc2^{+/-}$ mutants completely rescued both these ASD-like behaviors to the level of control mice (Fig. 3e). To further relate rsfMRI hyperconnectivity to the observed behavioral impairments, we next calculated the voxelwise correlation between socio-motor behavioral scores and prefrontal-DMN and insular-SN connectivity maps. Interestingly, this analysis did not reveal any significant correlation between social scores and rsfMRI hyperconnectivity ($|t| > 2$, $P > 0.05$, cluster-corrected). However, fronto–striatal–cortical hyperconnectivity was prominently associated with repetitive motor behavior (Fig. 3f). These findings suggest that mTOR-dependent cortico-striatal rsfMRI hyperconnectivity in $Tsc2^{+/-}$ mice is associated with increased ASD-like motor stereotypies, but are epiphenomenal to socio-behavioral impairments.

**Cortico-striatal hyperconnectivity in ASD is associated with an mTOR-related transcriptomic signature.** Convergent investigations point at a putative involvement of hyperactivity of the mTOR pathway in idiopathic ASD[7,14,16]. Based on these observations, we reasoned that an mTOR-related hyperconnectivity fingerprint recapitulating the connectional features observed in the mouse would be similarly identifiable in rsfMRI brain scans of children with ASD. To test this hypothesis, we leveraged the cross-species translatability of our rsfMRI mapping to identify an analogous signature of fronto–striato-insular hyperconnectivity in children with ASD. To this aim, we carried out spatially unbiased mapping of rsfMRI long-range connectivity in a large cohort of children with ASD ($n = 163$, age range 6–13 years old) and age-matched typically developing individuals (TDs, $n = 168$) available within the ABIDE collection[46]. Demographic and clinical information of individuals included in our analyses are in Table 1.

To identify the brain regions that most recurrently showed overconnectivity in ASD, we first carried out long-range rsfMRI connectivity mapping for each ABIDE's collection site and we then combined the site-specific brain maps in a single frequency map, retaining only voxels exhibiting Cohen's $d > 0.2$ in the ASD population. In a striking resemblance to our mouse results (cf. Fig. 1), this analysis revealed the presence of prevalent bilateral foci of increased long-range rsfMRI connectivity in the anterior insula, prefrontal cortex, and in the basal ganglia of children with ASD (Fig. 4a). Violin plots in Fig. 4a also indicate extended positive tails in the ASD group for nearly all sites, potentially indicating that a subset of idiopathic ASD cases might drive the between-group connectivity differences. To pinpoint anatomical hotspots of functional hyperconnectivity at the population level, we next aggregated all datasets across each participating site to perform a case–control mega-analysis of long-range rsfMRI connectivity. This analysis revealed the presence of bilateral foci of increased long-range functional connectivity in the anterior insular cortex of children with ASD ($t > 2$, $P < 0.05$, Fig. 4b). To map the functional topography of the corresponding insular networks, we then carried out seed-based correlation mapping using the identified area of insular hyperconnectivity as a seed region (Supplementary Fig. S5). The obtained case–control difference map revealed a pattern of rsfMRI overconnectivity in ASD highly consistent with network rsfMRI mapping in the mouse, with evidence of increased functional rsfMRI coupling between the anterior insula and prefrontal cortices, precuneus, angular gyrus, and basal ganglia (Fig. 4c). Importantly, rsfMRI hyperconnectivity was detectable also after removing all brain

**Table 1 Demographics and clinical scores.**

| | TD | ASD |
|---|---|---|
| Age (years) | 10.4 ± 1.5 | 10.5 ± 1.5 |
| Sex | 136 M/32 F | 139 M/24 F |
| IQ** | 103 ± 12 | 111 ± 18 |
| Reciprocal social interaction | --- | 20.0 ± 5.5 |
| Language and communication | --- | 16.1 ± 4.6 |
| Restricted, repetitive, and stereotyped behaviors and interests | --- | 6.4 ± 2.6 |

$**P < 0.01$, unpaired $t$ test.

scans collected at the site with the highest occurrence of hyperconnected ASD individuals (Stanford University [SU]), suggesting that our results are not primarily driven by the site with the largest differences (Supplementary Fig. S6). Similarly, the identified signature was also detectable after strict censoring of head motion in human ASD scans (Supplementary Fig. S7). Collectively, these analyses reveal a characteristic functional connectivity fingerprint in ASD that topographically recapitulates mTOR-dependent rsfMRI over-synchronization observed in rodents.

The cross-species identification of a topographically conserved connectivity fingerprint is consistent with possible involvement of dysregulated mTOR signaling in the emergence of fronto–striato–insular rsfMRI hyperconnectivity in ASD. To explore this hypothesis, we first used a gene expression decoding analysis to isolate genes whose spatial expression patterns across the cortex match the topology of human-derived ASD insular hyperconnectivity phenotypes[47–49]. This insular hyperconnectivity-relevant list of genes was then tested to identify whether it is highly enriched for genes known to be dysregulated in the ASD cortical transcriptome[50] and capable of interacting at the protein level with mTOR or Tsc2. Notably, we found that ASD-dysregulated and mTOR–Tsc2 interacting genes are significantly enriched amongst the genes that are spatially expressed within the identified insular hyperconnectivity phenotype (OR = 2.07, $P = 0.023$, Fig. 4e). This finding establishes a possible mechanistic link between mTOR-pathway dysfunction and the fronto-cortico-striatal hyperconnectivity observed in idiopathic ASD cases.

**mTOR transcriptomic signature is predominantly expressed in a subset of functionally hyperconnected ASD individuals.** While previous work has implicated aberrant mTOR signaling in idiopathic ASD[7,14], the daunting etiological heterogeneity underlying these disorders implies that only a fraction of ASD individuals within a multisite dataset like ABIDE would be expected to be affected by mTOR-related dysfunction. Indeed, Fig. 4a shows that within each ABIDE site, there are extended positive tails on the distributions, supporting the idea that case–control hyperconnectivity might be driven by a subset of idiopathic ASD cases. This observation supports the prediction that the identified population-level mTOR signature of hyperconnectivity may be most relevant in a subset of idiopathic ASD cases (Fig. 5a). To test this hypothesis, we carried out a cluster analysis of rsfMRI insular networks to parse ASD individuals into homogeneous neuro-functional subtypes, defined by their corresponding insular connectivity profile. Agglomerative hierarchical clustering revealed that children with ASD can be grouped into four subgroups (Fig. 5b). To map how functional connectivity is altered in each of these subgroups, we compared the insular connectivity of each ASD subgroup with that of TD controls. The

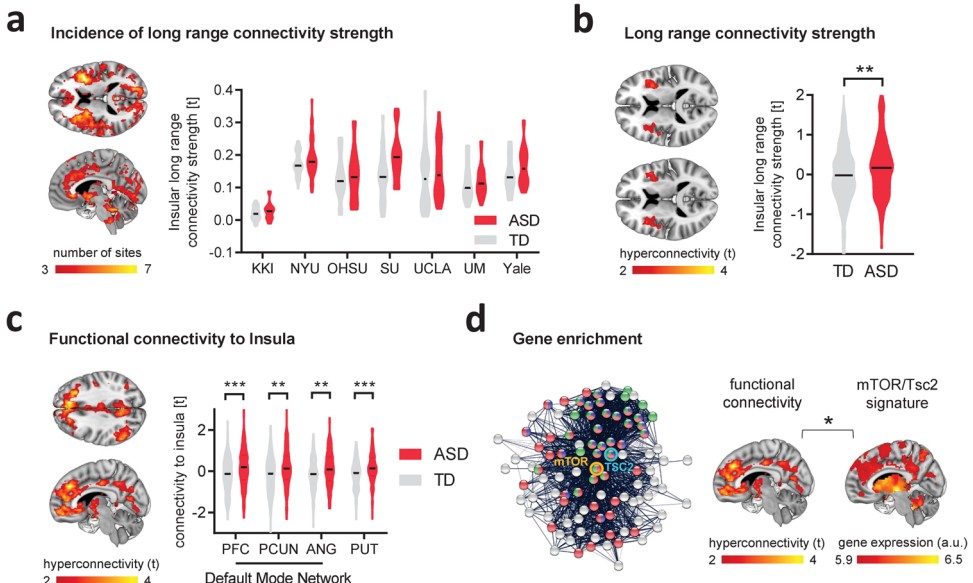

**Fig. 4 Fronto-insular-striatal hyperconnectivity in ASD children is associated with a Tsc2/mTOR gene expression signature. a** Voxelwise long-range connectivity mapping showed prominent rsfMRI hyperconnectivity in the anterior insular cortex of children with ASD vs. TDs. Red–yellow represents the occurrence rate i.e., the number of datasets presenting rsfMRI overconnectivity (Cohen's $d > 0.2$, left). Region-wise quantification confirmed that long-range rsfMRI connectivity of the anterior insular cortex is consistently increased in all seven sites of the ABIDE-I collection (Cohen's $d$ range, min = 0.06 [UM], max = 0.75 [SU], right). **b** Long-range rsfMRI connectivity mapping of the aggregated ABIDE datasets confirmed the presence of significant bilateral foci of functional hyperconnectivity in the insular cortex of ASD vs TD children (unpaired $t$ test, two-sided, $t = 3.04$, $P = 0.003$). Error bars represent SEM. **c** Seed-based probing shows functional over-synchronization between anterior insula, and basal ganglia, and prefrontal cortices in children with ASD vs. TDs. Regional quantification confirmed increased rsfMRI connectivity between insula and PFC (unpaired $t$ test, two-sided, $t = 3.94$, $P < 0.001$), precuneus (unpaired $t$ test, two-sided, $t = 3.26$, $P = 0.002$), angular gyrus (unpaired $t$ test, two-sided, $t = 3.13$, $P = 0.002$) and putamen (unpaired $t$ test, two-sided, $t = 4.33$, $P < 0.001$) in ASD vs. TDs. PFC prefrontal cortex, PCUN precuneus, ANG angular gyrus, PUT putamen. Error bars represent SEM. **d** Network-based visualization of Tsc2-mTOR interactome used for gene enrichment. The pictured interactome is limited to 100 genes for visualization purposes. Tsc2 seed gene is highlighted in light blue and mTOR seed gene in orange. Nodes that belongs to "Reactome mTOR signaling" are in green, those that are part of the "KEGG mTOR signaling pathway" are in red, and those that belong to "GO:0032006 regulation of TOR signaling" are in blue (left). The ASD functional hyperconnected phenotype is enriched for genes that are part of the Tsc2-mTOR interactome and reported to be dysregulated in ASD. *$P < 0.05$, **$P < 0.01$. ***$P < 0.001$. Source data, including gene lists, are provided as a Source Data file.

resulting neuro-subtypes were characterized by distinctive connectional signatures. Specifically, subtype 1 ($n = 19$) exhibited a global pattern of rsfMRI hyperconnectivity, while subtype 2 ($n = 21$) encompassed focal rsfMRI hyperconnectivity in the prefrontal, striatal, and cerebellar areas, broadly recapitulating key features of the hyperconnectivity signature identified at the population level. Subtype 3 ($n = 19$) showed widespread hypoconnectivity, while subtype 4 ($n = 102$) presented weak foci of frontal hyperconnectivity (Fig. 5c and Supplementary Fig. S8). Crucially, gene decoding and enrichment analyses revealed that subtype 2 was robustly and specifically enriched for mTOR/Tsc2-related transcripts (OR = 2.37, $P = 0.006$, $q = 0.024$, FDR-corrected, Fig. 5d). No enrichments were observed for any of the other three neuro-subtypes ($q > 0.44$, FDR-corrected, all comparisons). Similarly, no significant gene enrichment was observed when we aggregated subgroups 1, 3, and 4 together into a single group (OR = 1.33, $P = 0.41$), suggesting that the identified group-level mTOR signature from the ASD vs. TD case–control comparison is primarily driven by individuals belonging to subtype 2. Altogether, these findings reveal a segregable functionally defined mTOR-related ASD neuro-subtype, corroborating a possible causal link between dysfunctional mTOR signaling and fronto–striato–insular hyperconnectivity in ASD.

## Discussion

Here, we probe mechanistic links between ASD-relevant synaptic pathology and altered macroscale functional coupling in ASD. We show that mice with mTOR-dependent synaptic dysregulation exhibit a distinct functional hyperconnectivity signature that can be rescued by mTOR inhibition. Leveraging rsfMRI data in both rodents and humans, we report the identification of topographically analogous functional connectivity alterations in a subset of human idiopathic ASD cases, and document that this connectional signature is characterized by robustly enriched ASD-dysregulated and mTOR-pathway genes. These findings mechanistically reconcile developmental synaptopathy and aberrant functional connectivity in ASD into a multiscale frame that unifies two distinct and prominent conceptualizations of autism pathology.

Mutations in genes affecting synaptic pruning and homeostasis represent a key risk factor for the development of ASD[3,4]. In keeping with this, postmortem investigations have revealed altered synaptic density in ASD[6–8], suggesting that the pathogenesis of autism, or a substantial fraction of its heterogeneous expression, may be ascribed to synaptic dysfunction. While prior studies have dissected the signaling cascade produced by multiple forms of ASD-relevant synaptopathy (reviewed by Bagni et al.[51]), research on the brain-wide correlates of dysfunctional synaptic homeostasis in ASD has been lacking. Canonical cellular interpretations of increased basal dendrite spine density have putatively linked this trait to enhanced local excitatory connectivity, a feature of ASD[52] proposed to underlie neocortical excitation/inhibition imbalance and cause failure in differentiating signals from noise[7,51]. Our observation that mTOR-related synaptic excess leads to brain-wide functional hyperconnectivity crucially complements microscale modeling and advances our

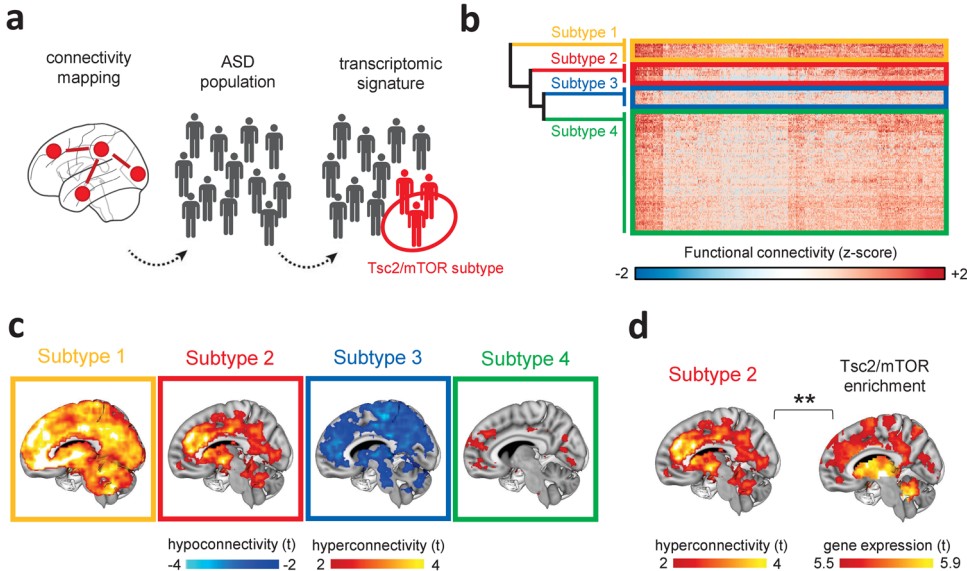

**Fig. 5 Identification of a Tsc2/mTOR-enriched hyperconnected ASD neuro-subtype. a** Illustration of the neuro-functional subtyping we used to identify the sub-population of ASD individuals enriched for Tsc2-mTOR network genes. RsfMRI connectivity mapping was applied to the heterogeneous population of ASD individuals followed by hierarchical clustering. Gene decoding and enrichment analysis were then applied to identify the ASD subtype that is most enriched for Tsc2/mTOR network genes. **b** The heatmap shows the four clusters identified with agglomerative hierarchical clustering. **c** Intergroup comparisons showed clear subtype-specific functional connectivity fingerprints, i.e., alterations of rsfMRI connectivity in ASD vs. TDs for each cluster. **d** Gene enrichment highlighted that only ASD subtype 2 was significantly enriched for mTOR/Tsc2 genes. **P < 0.01.

understanding of synaptopathy in ASD, suggesting that disrupted synaptic homeostasis can lead to multiscale alterations in neural connectivity. This notion is consistent with prior observations[7] that Tsc2 mutants display increased synaptic density in Layer V pyramidal neurons, a subtype of cells that serve as key orchestrators of network-wide functional connectivity[53,54]. The identification by us and others[7] of increased spine density in multiple cortical regions of Tsc2 mice suggests that mTOR-dependent spine pathology may affect large cortical territories. This observation is consistent with the spatial overextension of DMN network connectivity observed in Tsc2 mutants to include peripheral motor-sensory districts of these systems (Fig. 1c). Future investigations are warranted to assess whether this relationship can be generalized to other brain systems.

Prior animal research supports an association between pruning defects, altered spine density, and disrupted brain-wide connectivity. For example, investigations in mice characterized by microglia-deficient spine pruning revealed that removal of unnecessary synapses is required for the refinement of rsfMRI connectivity during development[28]. Our work expands these investigations by mechanistically linking a prevalent postmortem synaptic phenotype to a translationally relevant hyperconnectivity signature in ASD, and by documenting that, in the case of mTOR signaling, normalization of dendritic spine pathology is accompanied by the complete rescue of aberrant functional connectivity. Interestingly, cross-comparisons of synaptic and connectivity findings in mouse models of syndromic autism, such as Shank3B[55,56], 16p11.2[35], and Cntnap2[36,57] suggest that diminished dendritic spine density may lead to decreased brain-wide rsfMRI connectivity. While these observations would be consistent with a monotonic relationship between spine density and functional coupling in ASD, our poor understanding of the drivers of brain-wide functional coupling[58], together with the complex developmental physiology of synapses as well as their ability to homeostatically adapt to circuit dysfunction and brain wiring[51,59] suggest caution in the extension of this possible relationship to other forms of synaptopathy.

A non-biunivocal relationship may involve also mTOR-related signaling, as Fmr1$^{-/y}$ and Tsc2$^{+/-}$ mice, two mouse lines in which over-active mTOR activity has been reported[60], are characterized by opposing metabotropic glutamate-receptor-mediated protein synthesis[5] and divergent connectivity profiles, with evidence of rsfMRI under-connectivity in Fmr1-deficient mice[61]. While seemingly paradoxical, this observation is not at odds with the model described here because, as opposed to what we observed in Tsc2 mutants (Fig. 2), Fmr1 mice are characterized by strongly immature synapses[62] and preserved (or only marginally altered) spine density[63], two features that may conceivably lead to decreased interareal coupling strength. It should also be noted that the significance of mTOR hyperactivity in Fmr1 remains to be fully established, as demonstrated by rapamycin's inability to improve pathology in Fmr1$^{-/y}$ mice[60]. Collectively, these observations support a view in which the density and functionality of dendritic spines can crucially contribute to etiologically relevant functional dysconnectivity in autism. However, the direction and location of the ensuing dysfunctional coupling are likely to be strongly biased by additional pathophysiological components within the ASD spectrum, including maladaptive microcircuit homeostasis (i.e., E/I imbalance[64,65]), developmental miswiring[36], or alterations in modulatory activity governing brain-wide coupling[66]. Moreover, more subtle maturational mechanisms could conceivably compound synaptic abnormalities or independently contribute to the generation of the aberrant functional coupling. These might include mTOR-related maturational alterations in striatal excitability[67,68] or maladaptive synaptic plasticity and metabolic activity[69], all of which may impact rsfMRI BOLD signal.

The translational advantages of rsfMRI prompted us to identify a clinical correlate of the functional hyperconnectivity observed in the mouse via first-of-its-kind cross-species decoding of a distinctive connectivity signature observed in the rodent model. We found that a distinctive pattern of fronto-insular-striatal hyperconnectivity can be identified in idiopathic ASD. The observation of strikingly conserved cross-species circuital dysfunction, and its enrichment in ASD-dysregulated mTOR-interacting genes

support the validity of our approach, establishing a possible mechanistic link between mTOR-pathway hyperactivation and fronto-insular-striatal hyperconnectivity endophenotypes in idiopathic autism. These findings advance our understanding of ASD pathology in humans, revealing a neural circuitry that is sensitive to mTOR-related maturational derailment in segregable subpopulations of patients.

An interesting result of our investigations is the presence of hyperconnectivity in shared substrates of the salience and default-mode networks, such as the medial prefrontal cortex. While these two networks are clearly segregable with rsfMRI connectivity mapping, a degree of anatomical intersection between the two systems is apparent in rodents[33,70] and, to a lower extent, in humans[71]. The larger anatomical overlap in rodents is consistent with the lower extension and reduced regional differentiation of the cortical mantle in this species[72], and with the anatomical configuration of the rodent anterior cingulate as a relay-station linking striatal circuits to other higher-order association cortices[43]. In keeping with this notion, rsfMRI connectivity abnormalities in Tsc2 mutants mostly encompassed striatal regions, the PFC, the retrosplenial cortex and insular areas (Fig. 1b). These results delineate fronto-cortico-striatal hyperconnectivity as a core feature of mTOR-related connectopathy, and suggest that aberrant coupling between these substrates can differentially affect the topographical organization of ASD-relevant networks such as the salience and default-mode networks.

Our findings are broadly consistent with EEG data in children harboring Tsc2 mutations, in which broadly increased inter-regional synchronization has been recently reported[73]. These parallel findings are important, given that our cross-species measurements were carried out under different brain states, e.g., in lightly anesthetized rodents and awake ASD patients. Importantly, our approach also suggests that functional neuroimaging measurements can crucially aid deconstructions of ASD heterogeneity into distinct synaptopathy-based functional neuro-subtypes. Such translational links can aid in the development of circuit- and mechanism-specific therapeutic approaches[74]. This notion is further corroborated by recent cross-etiological mapping of rsfMRI in 16 mouse autism models, which resulted in the identification of four distinct connectivity neuro-subtypes characterized by specific, often diverging, network dysfunction fingerprints[75]. Notably, findings in mouse models are paralleled by the observation in the present study of four distinct connectivity profiles in the insular cortex, some of which exhibiting clearly opposing functional topography. Future applications of the cross-species, multiscale research platform we describe here might help unravel the connectional heterogeneity of ASD by permitting the identification of ASD subtypes[2]. Future case–control imaging investigations of individuals harboring Tsc2 mutations[76] might also crucially help test the validity of our predictions and their mechanistic significance in relevant patient populations.

In conclusion, our work establishes a mechanistic link between mTOR-related synaptic pathology and functional hyperconnectivity in ASD. Our findings provide direct evidence that aberrations in synaptic pruning can lead to behavioral disruption and ASD via alterations in neural connectivity, and define a unifying, multiscale translational framework that links developmental synaptopathy with functional circuit aberrations in ASD.

## Methods
### Mouse studies
*Ethical statement.* Animal studies were conducted in accordance with the Italian Law (DL 26/2014, EU 63/2010, Ministero della Sanità, Roma) and the recommendations in the Guide for the Care and Use of Laboratory Animals of the National Institutes of Health. Animal research protocols were reviewed and consented to by the animal care committee of the Istituto Italiano di Tecnologia and the Italian Ministry of Health. All surgical procedures were performed under anesthesia.

*Mice.* Male $Tsc2^{+/-}$ mice (Jackson Laboratories, Stock nr. 004686) backcrossed for more than ten generations with C57BL/6 J mice were used throughout the study. $Tsc2^{+/+}$ littermates were used as control, "wild-type" mice. Animals were group-housed in a 12:12 h light–dark cycle in individually ventilated cages with access to food and water ad libitum and with temperature maintained at $21 \pm 1\,°C$ and humidity at $60 \pm 10\%$. Food and water were provided ad libitum. We used a cross-sectional treatment protocol with four cohorts of $n = 20$ mice each: control $Tsc2^{+/+}$ mice treated with rapamycin (3 mg/kg i.p.); control $Tsc2^{+/+}$ mice treated with vehicle (2% DMSO, 30% PEG400, 5% Tween80); mutant $Tsc2^{+/-}$ mice treated with rapamycin; mutant $Tsc2^{+/-}$ mice treated with vehicle. Rapamycin or vehicle was administered every day for a consecutive week (p21–p28). rsfMRI and other tests were carried out in male mice at P29-P34 as in Tang et al.[7].

*Resting-state fMRI.* For resting-state functional MRI (rsfMRI) mice were anesthetized with isoflurane (5% induction), intubated, and artificially ventilated (2% maintenance)[35,77]. After surgery, isoflurane was discontinued and replaced with halothane (0.7%). Functional data acquisition started 45 min after isoflurane cessation. Possible genotype-dependent differences in anesthesia sensitivity were evaluated with minimal alveolar concentration, a readout previously shown to be correlated with anesthesia depth[28,78]. No intergroup differences were observed in minimal alveolar concentration ($Tsc2^{+/-}$ mutant mice: $1.51\% \pm 0.10$ and $Tsc2^{+/+}$ control mice: $1.55\% \pm 0.08$, $P = 0.35$). This observation argues against a significant confounding contribution of anesthesia to our functional measures. To further rule out a neurovascular origin of the mapped functional changes, we measured the hemodynamic response in the PFC, a brain region showing prominent hyperconnectivity in $Tsc2^{+/-}$ mice. Intergroup genotype-dependent comparisons revealed unimpaired hemodynamic response in $Tsc2^{+/-}$ mice (BOLD peak amplitude, $P = 0.75$; time to peak, $P = 0.52$; FWHM, $P = 0.75$, Supplementary Fig. S9). These results corroborate a neural (rather than neurovascular) origin of the rsfMRI changes in $Tsc2^{+/-}$ mice.

Functional images were acquired with a 7T MRI scanner (Bruker Biospin, Milan) with Bruker Paravision software (v6) and using a 72-mm birdcage transmit coil and a 4-channel solenoid coil for signal reception[79]. For each session, in vivo anatomical images were acquired with a fast spin-echo sequence (repetition time [TR] = 5500 ms, echo time [TE] = 60 ms, matrix $192 \times 192$, the field of view $2 \times 2$ cm, 24 coronal slices, slice thickness 500 μm). Co-centered single-shot BOLD rsfMRI timeseries were acquired using an echo-planar imaging (EPI) sequence with the following parameters: TR/TE = 1000/15 ms, flip angle 30°, matrix $100 \times 100$, the field of view $2.3 \times 2.3$ cm, 18 coronal slices, slice thickness 600 μm for 1920 volumes.

*Functional connectivity analyses.* RsfMRI timeseries preprocessing encompassed two sequential steps: core preprocessing and denoising[55,77]. Core preprocessing includes the following steps (in brackets the software and function employed). The initial 50 volumes of the timeseries were removed to allow for T1 and gradient thermal equilibration effects (AFNI v21, 3dTcat). BOLD timeseries were then despiked (AFNI, 3dDespike), motion-corrected (FSL v6, mcflirt), skull-stripped (FSL, bet), and spatially normalized with affine and diffeomorphic registration (ANTS v2, antsRegistration + antsApplyTransforms) to a skull-stripped reference BOLD template. The denoising pipeline included the following steps: motion traces of head realignment parameters (three translations + three rotations) and mean ventricular signal (corresponding to the averaged BOLD signal within a reference ventricular mask, FSL, fslmeants) were used as nuisance covariates and regressed out from each time course (FSL, fsl_regfilt). All rsfMRI timeseries also underwent band-pass filtering to a frequency window of 0.01–0.1 Hz (AFNI 3dBandpass) and spatial smoothing with a full width at half maximum of 0.6 mm (AFNI, 3dBlurInMask). To obtain a data-driven identification of the brain regions exhibiting genotype-dependent alterations in functional connectivity, we calculated voxelwise weighted global connectivity (GC) maps for all mice as in ref. [80] (here referred to as global connectivity). Global connectivity is also known as unthresholded weighted-degree centrality in graph-based metrics, is a measure of mean temporal correlation between a given voxel and all other voxels within the brain[80]. Global connectivity was calculated using Python (v3). Global connectivity is strongly biased toward long-range connections (defined here as >600 μm apart from a given voxel) as these comprise 99.77% voxels in the brain. In keeping with this notion, we found that restricting GC within a radius of 600 μm did not reveal any intergroup differences between wild-type and $Tsc2^{+/-}$ mutants treated with rapamycin or vehicle ($|t| > 2$, $P > 0.05$, FWER $P > 0.01$, all comparisons). For this reason, we have here used the terms global connectivity and long-range connectivity interchangeably. Voxelwise rsfMRI measurements allow for a spatially unbiased mapping of connectivity without the constraints of pre-imposed anatomical boundaries[43] and are amenable to direct cross-species translation[35]. Pearson's correlation scores were first transformed to $z$ scores using Fisher's $r$-to-$z$ transform and then averaged to yield the final connectivity scores. Target regions of long-range connectivity alterations in $Tsc2^{+/-}$ mice were mapped using seed-based analysis in volumes of interest

(VOIs). Specifically, VOIs were placed in the prefrontal cortex to map derangements in anteroposterior connectivity within the default-mode network, and bilaterally in the insula to probe impaired connectivity within the salience network. As for long-range connectivity analysis, $r$ scores were transformed to $z$ scores using Fisher's $r$-to-$z$ transform before statistics. Voxelwise intergroup differences in local and long-range connectivity as well as for seed-based mapping were assessed using a two-tailed Student's $t$ test ($|t| > 2$, $P < 0.05$) and family-wise error (FWER) cluster-corrected for multiple comparisons using a cluster threshold of $P < 0.01$ as implemented in FSL[81]. Anteroposterior default-mode network (DMN) and salience network (SN) connectivity alterations were quantified by computing temporal Pearson's correlation between seed regions and cubic volumes of interest ($3 \times 3 \times 1$ voxels) and plotted using scatterplots. VOI-based analysis was carried out using Prism GraphPad (v9.2). Seeds and VOI locations are depicted in Supplementary Fig. S10.

The statistical significance of intergroup changes was quantified using an unpaired two-tailed Student's $t$ test ($P < 0.05$). Given our specific hypotheses, we applied FDR (Benjamini-Hochberg) to correct for multiple comparisons our set of planned contrasts. Our contrast of interest were (a) the functional connectivity alterations produced by loss of $Tsc2$ (i.e., $Tsc2^{+/-}$ mutants vs. control littermates), (b) its rapamycin-induced rescue (i.e., $Tsc2^{+/-}$ mutants treated with rapamycin vs. vehicle-treated $Tsc2^{+/-}$ mutants), and (c) the degree of phenotypic normalization produced by rapamycin in $Tsc2^{+/-}$ mutants (i.e., vehicle-treated wild-type mice vs. rapamycin-treated $Tsc2^{+/-}$ mutants). We used the same FDR-correction strategy to correct intergroup comparisons of dendritic spine density and behavioral scores. To assess the network specificity of the rsfMRI changes observed in $Tsc2^{+/-}$ mice, we regionally quantified the spatial intersection between the DMN and salience network (SN) using Dice Similarity Coefficient (DSC)[82]. DCS quantifications yielded a score of 0.41, and 0.54 in control and $Tsc2^{+/-}$ mutants, respectively. A spatial comparison of SN-DMN overlap with SN and DMN hyperconnectivity maps in $Tsc2^{+/-}$ mutants (Fig. 3c, d, bottom row) revealed that 62% of the foci of hyperconnectivity were SN-specific and 55.7% were DMN-specific. Areas of shared abnormalities mostly encompassed striatal regions, the PFC, posterior cingulate, and insular areas, to recapitulate the pattern of increased long-range connectivity reported in Fig. 1b.

*Dendritic spine quantification.* To achieve sparse cortical labeling, 2 µl of AAV-hSyn-GFP $10^{10}$ IU/ml were injected bilaterally in the lateral ventricles of newborn (0–1 postnatal day) $Tsc2^{+/-}$ ($n = 5$ treated with vehicle and $n = 7$ treated with rapamycin) and $Tsc2^{+/+}$ littermates ($n = 7$ treated with vehicle and $n = 7$ treated with rapamycin). Rapamycin/vehicle intraperitoneal injections were performed daily from postnatal day 21 (p21) to p28. Mice were sacrificed at p29 by transcardial 4% paraformaldehyde-PBS perfusion, brains were sliced with a vibratome (Leica) at 100-µm thickness (coronal) and processed for free-floating immunofluorescence using chicken anti-green-fluorescent-protein (GFP) primary antibody (1:1000 Abcam, ab13970), and goat anti-chicken-alexa488 secondary antibody (1:500 Thermo fisher, A11039). High-resolution confocal images were acquired with a Nikon A1 confocal microscope, NIS element AR v4.20.03 software (Nikon) using a 60x plan-apo oil immersion objective. Spine density quantification was performed on basal dendrites of layer V pyramidal neurons both in secondary somatosensory cortex (SS2) as in ref. [7] and in the neighboring anterior insula, a core component of the mouse salience network[33]. Quantifications were carried out with Imaris-Microscopy Image Analysis Software by Bitplane (v7.2.3). The same measurements were replicated also in the retrosplenial cortex (wild-type mice, $n = 5$; $Tsc2^{+/-}$ mice, $n = 4$; wild-type mice treated with rapamycin, $n = 5$; $Tsc2^{+/-}$ mice treated with rapamycin, $n = 5$), a distal region showing similar rsfMRI hyperconnectivity in $Tsc2^{+/-}$ mice. The analysis was performed by an operator blind to the genotype and treatment. Data are expressed as spine count per µm of dendrite length. Statistical analysis was carried out with Prism GraphPad (v9.2).

*Diffusion MRI and white matter fractional anisotropy.* Postmortem MRI-based diffusion-weighted (DW) imaging was carried out in $Tsc2^{+/-}$ mutants and control wild-type littermates[55]. Images were acquired ex vivo in PFA-fixed specimens, a procedure used to obtain high-resolution images with negligible confounding contributions from physiological or motion artifacts. Brains were imaged inside intact skulls to avoid post-extraction deformations. Mice were deeply anesthetized with 5% isoflurane, and their brains were perfused in situ via cardiac perfusion[82,83]. The perfusion was performed with PBS followed by 4% PFA (100 ml, Sigma-Aldrich). Both perfusion solutions were added with a gadolinium chelate (Pro-Hance, Bracco) at a concentration of 10 and 5 mM, respectively, to shorten longitudinal relaxation times.

High-resolution DW images morphoanatomical T2-weighted MR imaging of mouse brains was performed using a 72-mm birdcage transmit coil, a custom-built saddle-shaped solenoid coil for signal reception. Each DW dataset was composed of 8 non-DW images and 81 different diffusion gradient-encoding directions with $b = 3000$ s/mm$^2$ ($\partial = 6$ ms, $\Delta = 13$ ms) acquired using an EPI sequence with the following parameters: TR/TE = 13500/27.6 ms, the field of view $1.68 \times 1.54$ cm, matrix $120 \times 110$, in-plane spatial resolution $140 \times 140$ µm, 54 coronal slices, slice thickness 280 µm, the number of averages 20 as recently described[55]. The DW datasets were first corrected for eddy current distortions and skull-stripped. The resulting individual brain masks were manually corrected using ITK-SNAP[84].

Tract-based spatial statistics (TBSS) analysis was implemented in FSL[27,85]. Fractional anisotropy (FA) maps from all subjects were nonlinearly registered to an in-house FA template with FLIRT and FNIRT and thinned using a FA threshold of 0.2 to create a skeleton of the white matter. Voxelwise intergroup differences between deletion and control mice were evaluated with permutation testing using 5000 permutations ($P < 0.05$). Genotype-dependent FA alterations were also regionally quantified in major white matter structures, including the corpus callosum, dorsal hippocampal commissure, anterior commissure, cingulate and internal capsule using VOIs of $3 \times 3 \times 1$ voxels. Genotype-dependent differences were assessed with an unpaired two-tailed Student's $t$ test ($P < 0.05$). For visualization purposes, white matter regions are depicted in red (Fig. 2a, top) on the Allen Mouse Brain Atlas (http://www.brain-map.org/). Statistical analysis was carried out with Prism GraphPad (v9.2).

*Virus production and injection.* For unpseudotyped recombinant SAD∆G-mCherry Rabies Virus production, B7GG packaging cells, which express the rabies envelope G protein, were infected with unpseudotyped SAD∆G-mCherry-RV[35,55]. Five to six days after infection, viral particles were collected, filtrated through 0.45-µm filter, and concentrated by two rounds of ultracentrifugation. The titer of the SAD∆G-mCherry-RV preparation was established by infecting Hek-293T cells (ATCC cat no. CRL-11268) with tenfold serial dilution of viral stock, counting mCherry expressing cells 3 days after infection. The titer was calculated as $2 \times 10^{11}$ Infective Units/ml (IU/ml), and the stock was therefore considered suitable for in vivo microinjection. Mice were anesthetized with isoflurane (4%) and firmly stabilized on a stereotaxic apparatus (Stoelting Inc.). A micro drill (Cellpoint Scientific Inc.) was used to drill holes through the skull. RV injections were performed with a Nanofil syringe mounted on an UltraMicroPump UMP3 with a four-channel Micro4 controller (World Precision Instruments), at a speed of 5 nl per seconds, followed by a 5–10 min waiting period, to avoid backflow of viral solution and unspecific labeling[86]. One microliter of viral stock solution was injected unilaterally in the primary cingulate cortex of $Tsc2^{+/-}$ and wild-type control mice. Coordinates for injections, in mm from Bregma: +1.42 from anterior to posterior, +0.3 lateral, −1.6 deep.

*Immunohistochemistry and image analysis.* Animals were transcardially perfused with 4% paraformaldehyde (PFA) under deep isoflurane anesthesia (5%), brains were dissected, post-fixed overnight at 4 °C and vibratome-cut (Leica Microsystems). RV infected cells were detected by means of immunohistochemistry performed on every other 100-µm thick coronal section, using rabbit anti-red-fluorescent-protein (RFP) primary antibody (1:500 Abcam), and goat anti-rabbit-HRP secondary antibody (1:500 Jackson Immunoresearch), followed by 3-3' diaminobenzidine tetrahydrochloride (DAB, Sigma-Aldrich) staining. Wide-field imaging was performed with MacroFluo microscope (Leica), Nis Elements F v3.2 software (Nikon) and RGB pictures where acquired at $1024 \times 1024$ pixel resolution. Labeled neuron identification was manually carried out in $Tsc2^{+/-}$ ($n = 4$) mutants and $Tsc2^{+/+}$ ($n = 4$) control littermates by an operator blind to the genotype, while the analysis was performed using custom made scripts to automatically register each brain section on the corresponding table on the Allen Mouse Brain Atlas and to count labeled neurons and assign them to their anatomical localization. Final regional cell population counts were expressed as a fraction of the total amount of labeled cells counted across the entire brain (including both hemispheres) for each animal. Statistical analysis was carried out with Prism GraphPad (v9.2).

*MBP immunohistochemistry and imaging analysis.* To rule out the presence of myelination deficits in our mice, we measured MBP and MBP coherency in $Tsc2^{+/-}$ and wild-type mice treated with and without rapamycin. Myelin basic protein (MBP) was stained using rat anti-MBP (Abcam ab7349). Briefly, 50-µm vibratome brain sections were incubated in blocking solution (5% horse serum in PBTriton 0.1%) for 1 h at room temperature and subsequently incubated overnight at 4 °C with the primary antibody rat anti-MBP (1:1000). The following day, four 1-h washes in PBTriton 0.1% were performed. Brain sections were then incubated overnight at 4 °C with a donkey anti-rat IgG (H + L) secondary antibody Alexa Fluor 488-conjugate (ThermoFisher A21208). Sections were rinsed in PBTriton 0.1% and mounted with AquaPolymount (Polyscience). Control brain sections were processed using the same protocol, omitting the primary antibody. Images of MBP immunostaining were acquired with a Nikon A1 confocal microscope, NIS element AR v4.20.03 software (Nikon) in the corpus callosum (CC), cingulate cortex (Cg), and hippocampus (HP). A ×20 plan-apochromat objective was used to image three sections at comparable anteroposterior anatomical levels ($n = 3$ animals for each experimental cohort, 0.7 µm Z-step, 17 steps).

Quantification of fluorescence intensity (optical density, OD) was performed with ImageJ Fiji software (v1.53c). To obtain a relative optical density value (ROD), the background value measured on control sections was subtracted to the OD value. From each section, three square $124 \times 124 \times 10$-µm regions (e.g., corpus callosum and cingulate cortex) or $93 \times 93 \times 10$ µm (layer lacunosum moleculare of hippocampal CA1) were analyzed ($n = 3$ sections per subject, $n = 3$ animals per cohort). Coherency of MBP immunostained fibers was computed as in[38] using the function measure of OrientationJ plugin of ImageJ software. Coherency was evaluated on three regions per section as described above ($n = 3$ sections per

subject, $n = 3$ animals per cohort). Statistical analysis was carried out with Prism GraphPad (v9.2).

*Whole-brain network modeling.* The in silico whole-brain model used in this study is a biological plausible Dynamic Mean Field (DMF) approximation[44] of multiple interconnected excitatory and inhibitory spiking neurons into reciprocally coupled excitatory ($E$) and inhibitory ($I$) pools of neurons[87]. Briefly, the coupled differential Eqs. (1–6) describe the dynamic for each type of pool ($E$, $I$) within each brain region $i$, where $I_i^{(E,I)}$ (in nA) defines the total input current, $r_i^{(E,I)}$ (in Hz) characterizes the firing rate and $S_i^{(E,I)}$ indicates the synaptic gating variable. The overall effective external input current $I_0 = 0.382$ nA is weighted differently depending on the type of neuronal pool, such as $W_E = 1$ and $W_I = 0.7$. Short- and long-range excitatory synaptic connections, are weighted by $J_{NMDA} = 0.15$ nA. However, the excitatory recurrent weight is defined by $w_+ = 1.4$. The input–output function $H^{(E,I)}$ converts the total incoming synaptic input currents to $E$ and $I$ pools into firing rates[88], where $a_E = 310$ nC$^{-1}$ and $a_I = 615$ nC$^{-1}$ refer to the gain factor, $b_E = 125$ (Hz) and $b_I = 177$ (Hz) denote the input current and $d_E = 0.16$ (s) and $d_I = 0.087$ (s) specify the noise factor that determines the shape of the curvature of $H$. The excitatory $S_i^{(E)}$ and inhibitory $S_i^{(I)}$ synaptic gating variables, are respectively mediated by the NMDA and GABA$_A$ receptors with the corresponding decay time constant of $\tau_{NMDA} = 0.1$ s (plus the kinetic parameter $\gamma = 0.641$), and $\tau_{GABA} = 0.01$ s. Both synaptic gates depend on an uncorrelated standard Gaussian noise $\nu_i$ with amplitude $\sigma = 0.01$ nA.

$$I_i^{(E)} = W_E I_0 + w_+ J_{NMDA} S_i^{(E)} + G J_{NMDA} \sum_j C_{ij} S_j^{(E)} - J_i S_i^{(I)} \quad (1)$$

$$I_i^{(I)} = W_I I_0 + J_{NMDA} S_i^{(E)} - S_i^{(I)} \quad (2)$$

$$r_i^{(E)} = H^{(E)}\left(I_i^{(E)}\right) = \frac{a_E I_i^{(E)} - b_E}{1 - \exp\left(-d_E\left(a_E I_i^{(E)} - b_E\right)\right)}, \quad (3)$$

$$r_i^{(I)} = H^{(I)}\left(I_i^{(I)}\right) = \frac{a_I I_i^{(I)} - b_I}{1 - \exp\left(-d_I\left(a_I I_i^{(I)} - b_I\right)\right)}, \quad (4)$$

$$\frac{dS_i^{(E)}(t)}{dt} = -\frac{S_i^{(E)}}{\tau_{NMDA}} + \left(1 - S_i^{(E)}\right)\gamma r_i^{(E)} + \sigma \nu_i(t) \quad (5)$$

$$\frac{dS_i^{(I)}(t)}{dt} = -\frac{S_i^{(I)}}{\tau_{GABA}} + r_i^{(I)} + \sigma \nu_i(t) \quad (6)$$

The parameters were defined and used as in ref. [87] to simulate the spontaneous neural activity such that the excitatory firing rate of a single disconnected node should be close to a low firing rate (~3Hz)[89–92] and fixed to it even when receiving excitatory input from coupled brain regions. This is done by adjusting the $J_i$ inhibitory weight for each area[44]. The mesoscale wiring between $ij$ regions, is established at the E-to-E level by the structural connectivity $C_{ij}$, from the Allen Mouse Brain Connectome[93], parcellated into 74 brain regions. Those excitatory couplings between brain areas were then equally scaled through the excitatory coupling strength (denoted by $G$). To compare experimental and simulated data, the modeled neural activity was transformed through the Balloon–Windkessel hemodynamic model into BOLD signal[94]. More details on the equations and parameters are discussed elsewhere[95]. In total, 50 simulations of 3000 timepoints were run with $G$ varying in a range of [0, 0.75] using steps of 0.005 and a time step of 1.2 ms. Intergroup rsfMRI functional connectivity comparisons between $Tsc2^{+/-}$ and $Tsc2^{+/+}$ mice were carried out at the best fit between empirical and simulated data based on the optimal excitatory coupling strength value (G). The minimum absolute difference between the mean upper triangular rsfMRI connectivity matrix from empirical and simulated data was used to estimate the optimal excitatory coupling strength that optimally explains the whole-brain functional connectivity experimentally measured in $Tsc2^{+/-}$ and $Tsc2^{+/+}$ mice.

*Slice preparation and electrophysiological recordings.* Mice were anesthetized with isoflurane and decapitated, and their brains were transferred to ice-cold dissecting modified-artificial cerebrospinal fluid (aCSF) containing 75 mM sucrose, 87 mM NaCl, 2.5 mM KCl, 1.25 mM NaH$_2$PO$_4$, 7 mM MgCl$_2$, 0.5 mM CaCl$_2$, 25 mM NaHCO$_3$, 25 mM D-glucose, saturated with 95% O$_2$ and 5% CO$_2$. Coronal sections (250-μm thick were cut using a Vibratome 1000 S (Leica, Wetzlar, Germany), then transferred to aCSF containing 115 mM NaCl, 3.5 mM KCl, 1.2 mM NaH$_2$PO$_4$, 1.3 mM MgCl$_2$, 2 mM CaCl$_2$, 25 mM NaHCO$_3$, and 25 mM D-glucose and aerated with 95% O$_2$ and 5% CO$_2$. Following 20 min incubation at 32 °C, slices were kept at 22–24 °C. During experiments, slices were continuously superfused with aCSF at a rate of 2 ml/min at 28 °C.

Whole-cell patch-clamp recordings were made on deep-layer cortical neurons or on striatal projection neurons (SPNs) of the dorsolateral striatum (DLS) in coronal slices. Excitatory postsynaptic currents (EPSCs) were evoked in the presence of the GABA$_A$ receptor antagonist gabazine (10 μM) by stimulation of the cortical layer II/III by using a theta glass electrode or by intrastriatal stimulation

using a concentric bipolar electrode (20 μsc–80 μsc, 0.02 mA–0.1 mA) connected to a constant-current isolation unit (Digitimer LTD, Model DS3) and acquired every 10 s. Voltage clamp experiments were performed on PFC deep-layer pyramidal neurons or on SPNs, using borosilicate patch pipettes (3–4 MΩ) filled with a solution containing (in mM): 135 CsMeSO$_3$, 5 CsCl, 5 NaCl, 2 MgCl$_2$, 0.1 EGTA, 10 HEPES, 0.05 CaCl$_2$, 2 Na2-ATP, 0.4 Na$_3$-GTP (pH 7.3, 280–290 mOsm/kg). Neurons were voltage-clamped at −70 mV and at +40 mV to evoke AMPA and NMDA receptor-mediated EPSCs respectively. AMPA and NMDA EPSCs were recorded before and after blocking AMPA-mediated currents by bath applying 20 μM NBQX disodium salt. Access resistance was monitored throughout the experiment. Signals were sampled at 10 kHz filtered at 2.4 kHz. Series resistance (range 10–20 MΩ) was monitored at regular intervals throughout the recording and presented minimal variations (≤ 20%) in the analyzed cells. Data are reported without corrections for liquid junction potentials. Data were acquired using a Multiclamp 700B amplifier controlled by pClamp 10 software (Molecular Device), with a Digidata 1322 (Molecular Device). AMPA/NMDA ratio of each neuron was calculated as the ratio between AMPA EPSC peak amplitude (pA) and the NMDA EPSC peak amplitude (pA) of the subtracted current upon NBQX application. Statistical analysis was carried out with Prism GraphPad (v9.2).

*High-resolution structural MRI.* To rule out the presence of major morphoanatomical changes in $Tsc2^{+/-}$ mice, we carried out ex vivo structural MRI on PFA-fixed specimens[82] on a cohort of $n = 5$ $Tsc2^{+/-}$ male and $n = 5$ control littermates. Brains were imaged inside intact skulls to avoid post-extraction deformations. Mice were deeply anesthetized with 5% isoflurane, and their brains were perfused in situ via cardiac perfusion. PFA intracardiac perfusion was performed with PBS followed by 4% PFA (100 ml, Sigma-Aldrich). Both perfusion solutions were doped with a gadolinium chelate (ProHance, Bracco) at a concentration of 10 and 5 mM, respectively, to shorten longitudinal relaxation times. High-resolution morphoanatomical T2-weighted MR imaging of mouse brains was performed using a 72-mm birdcage transmit coil and a custom-built saddle-shaped solenoid coil for signal reception. High-resolution morphoanatomical images were acquired with the following imaging parameters: FLASH 3D sequence with TR = 17 ms, TE = 10 ms, α = 30°, matrix size of 260 × 180 × 180, FOV of 1.83 × 1.26 × 1.26 cm, and voxel size of 70 μm (isotropic). Structural MRI confirmed the absence of cerebral tubers in our $Tsc2^{+/-}$ mouse line, confirming the previous reports[39].

*Behavioral tests.* Behavioral testing was carried out at P29 on the same mice that underwent the imaging sessions. Mice underwent a male–male social interaction test during the light phase[96,97]. Before behavioral testing, each mouse was placed in the test cage and left to habituate for one hour. Then, the unfamiliar mouse was placed into the testing cage for a 5-min test session. Scoring was conducted by an observer blind to mouse genotype and treatment, and multiple behavioral responses exhibited by the test mouse were measured, including anogenital sniffing (direct contact with the anogenital area), body sniffing (sniffing with the flank area), head sniffing (sniffing with the head area), following (time spent in following the unfamiliar mouse), self-grooming (self-cleaning, licking any part of its own body), wall rearing (rearing up against the wall of the home-cage), digging in the bedding and immobility. Behaviors were recorded by using the Any-Maze Video Tracking software (v6.34) (Stoelting Co.) connected to a Camera DMK 22AUC03 (Ugo Basile). Voxelwise correlation mapping between behavioral scores and seed-based functional connectivity maps of the prefrontal and insular cortex were performed with Pearson's correlation ($|t| > 2$, $P < 0.05$, and FWER cluster-corrected using a cluster threshold of $P < 0.01$). Statistical analysis was carried out with Prism GraphPad (v9.2).

**Human studies**

*Ethical statement.* Human rsfMRI and clinical data were downloaded from the ABIDE website (http://fcon_1000.projects.nitrc.org/indi/abide/abide_I.html). In accordance with HIPAA guidelines and 1000 Functional Connectomes Project/ INDI protocols, all ABIDE datasets have been anonymized, with no protected health information included. Ethical approval of the ABIDE study was obtained by the local Institutional Review Board of each laboratory[46].

*Resting-state fMRI data and preprocessing.* Resting-state fMRI timeseries of children with ASD ($n = 163$, 6–13 years old) and age-matched typically developing (TD) controls ($n = 168$) were downloaded from the Autism Brain Imaging Data Exchange[46] initiative. We analyzed data collected from eight independent laboratories including Kennedy Krieger Institute (KKI), New York University (NYU), Oregon Health and Science University (OHSU), Stanford University (SU), University of California Los Angeles (UCLA, samples 1 and 2), University of Michigan (UM, sample 1) and Yale Child Study Center (YALE). For details of scan parameters for each collection site, see at http://fcon_1000.projects.nitrc.org/indi/abide/ abide_I.html.

Preprocessing of the resting-state data was split into two components; core preprocessing and denoising. Core preprocessing was implemented with AFNI[98] (http://afni.nimh.nih.gov/) using the tool speedypp.py[99] (http://bit.ly/23u2vZp). This core preprocessing pipeline included the following steps: slice acquisition correction using heptic (7th order) Lagrange polynomial interpolation, rigid-body head movement correction to the first frame of data, using quintic (5th order)

polynomial interpolation to estimate the realignment parameters (three displacements and three rotations), obliquity transform to the structural image, affine co-registration to the skull-stripped structural image using a gray-matter mask, nonlinear warping to MNI space (MNsI152 template) with AFNI 3dQwarp, spatial smoothing (6 mm FWHM), and a within-run intensity normalization to a whole-brain median of 1000.

Core preprocessing was followed by denoising steps to further remove motion-related and other artifacts. Denoising steps included: wavelet timeseries despiking ("wavelet denoising"), confound signal regression including the six motion parameters, their first order temporal derivatives, and ventricular cerebrospinal fluid (CSF) signal (referred to as 13-parameter regression). To avoid the generation of artificial anti-correlation in connectivity measurements, the fMRI global signal was not regressed. The wavelet denoising method has been shown to mitigate substantial spatial and temporal heterogeneity in the motion-related artifact that manifests linearly or nonlinearly and can do so without the need for data scrubbing[100]. Wavelet denoising is implemented with the Brain Wavelet toolbox (http://www.brainwavelet.org) in MATLAB (R2019a). The 13-parameter regression of motion and CSF signals was achieved using AFNI 3dBandpass with the -ort argument. To further characterize motion and its impact on the data, we computed FD and DVARS[101]. To further denoise the resting-state fMRI data by using a more stringent pipeline, we also carried out a set of supplementary analyses where we additionally regressed out either mean white matter signal (WM), or WM principal component as calculated with CompCorr[102] and implemented in ANTsR (https://www.rdocumentation.org/packages/ANTsR/versions/1.0)". Regression of this set of covariates did not alter the direction and anatomical location of our connectivity findings (Supplementary Fig. S11), arguing against a significant contribution of motion-related artifacts and physiological noise to the observed rsfMRI hyperconnectivity. As these pipelines produced highly consistent results, and given the controversy around the regression of WM components[103–105], all the subsequent analyses were carried out without WM regression.

*Functional connectivity analyses.* Analogous to the functional connectivity analysis we carried out for the mouse study (see above), to obtain an unbiased mapping of intergroup connectivity differences we calculated individual maps of global connectivity (hereafter interchangeably referred to as "long-range rsfMRI connectivity"). To exclude the contribution of white matter and CSF, we limited our measurements to voxels within a 25% gray-matter probability mask[35]. To test the across-site reproducibility of our findings, we first calculated voxelwise intergroup comparisons of global connectivity maps for each dataset (i.e., site) separately by using Cohen's $d$[106]. We then produced a summary occurrence map, indicating the number of datasets presenting Cohen's $d > 0.2$. To identify the hotspots of long-range functional hyperconnectivity, we aggregated all the scans and carried out a meta-analysis of global connectivity. Prior to this, we used a mixed-model regression analysis to regress out the spurious contribution of intersite variability and harmonize functional connectivity maps. The same model was also used to regress out the contribution of age, sex, and IQ. Intergroup comparisons were then carried out by using unpaired two-tailed Student's $t$ test ($|t| > 2$, $P < 0.05$). Given the highly focal nature of the insular clusters mapped with global connectivity (Fig. 4b), the obtained map did not survive formal cluster correction. However, the exquisitely bilateral nature of the mapped effect and the presence of a clearly identifiable involvement of insular overconnectivity across all the examined ASD sites (Fig. 4a) as per our hypothesis, strongly support the biological plausibility of this result. We subsequently performed seed analyses, using as seed the bilateral foci showing altered global connectivity. Voxelwise intergroup differences for seed-based mapping were assessed using unpaired two-tailed Student's $t$ test ($|t| > 2$, $P < 0.05$) and family-wise error (FWER) cluster-corrected using a cluster threshold of $P < 0.01$ as implemented in FSL. VOI-based analysis was carried out using Prism GraphPad (v9.2).

Despite the use of wavelet despiking[100], intergroup comparisons revealed that mean framewise displacement (FD) was higher in ASD vs. TD children (0.26 ± 0.19 and 0.19 ± 0.12, respectively). To confirm that our results were not contaminated by in-scanner head motion we applied a stricter control for motion, by removing all timepoints with FD higher than 0.2 mm, and we repeated global connectivity and seed-based correlation mapping on the obtained censored timeseries. Intergroup comparisons largely confirmed the results obtained with unscrubbed BOLD timeseries of children with ASD, ruling out a possible spurious contribution of head motion in the mapped effects (Supplementary Fig. S1). Statistical analysis was carried out with Prism GraphPad (v9.2).

*Gene expression decoding and enrichment analysis.* To link rsfMRI functional hyperconnectivity in human patients with molecular mechanisms of relevance to mTOR and Tsc2, we carried out gene decoding and gene enrichment analysis using R (v4.1). The goal of this analysis is to test whether the list of genes exhibiting an expression pattern spatially correlated to our rsfMRI map is significantly enriched for genes belonging to the mTOR–Tsc2 interactome and dysregulated in ASD. As the two lists of genes are independently determined, this analysis does not suffer from statistical circularity. To carry out gene decoding analysis we used Neurosynth and NeuroVault[47] to identify genes whose spatial expression patterns are consistently similar across participants to our maps of connectivity differences.

This decoding analysis utilizes spatial gene expression data from the six donor brains from the Allen Institute Human Brain Gene Expression atlas[48,49]. This database contains gene expression maps for $n = 20,787$ genes. The analysis first utilizes a linear model to compute the similarity between a user-input unthresholded whole-brain imaging map and spatial patterns of gene expression for each of the six brains in the Allen Institute dataset (http://human.brain-map.org/). The slopes of these donor-specific linear models encode how similar each gene's spatial expression pattern is with user-input imaging map. Donor-specific slopes were then subjected to a one-sample $t$ test to identify genes whose spatial expression patterns are consistently of high similarity across the donor's brains to the user-input imaging map. The resulting list of genes is then thresholded for multiple comparisons and only the genes positive t-statistic values surviving FDR $q < 0.05$ are considered. In our application of these gene expression decoding analyses, the user-input imaging maps used were whole-brain unthresholded t-statistic maps of group differences in insular seed connectivity for ASD vs TD or ASD subgroup vs TD comparisons. The result of this first analysis is a list of $n = 2412$ genes (FDR corrected, $q < 0.05$) whose expression topography is statistically similar to our input fMRI hyperconnectivity map.

With a subset of fMRI-relevant genes isolated, we then asked whether this list significantly overlaps with genes that interact at the protein level with mTOR or Tsc2. To identify a broad network of mTOR-Tsc2 relevant genes, we carried out protein–protein interaction (PPI) analyses in STRING-DB[107] (https://string.db.org). Here, we used Tsc2 or mTOR as seed genes, and queried for up to 500 possible interactors at the default confidence level of "medium" (0.4). Regarding the settings for STRING-DB, we have not restricted the type of interactors because there was no a priori reason to prioritize on type of interactor over another. Also, the use of default settings of medium confidence was instrumental to our goal of generating a broad mTOR–Tsc2 interactome network. This initial broad network was then pruned to be more ASD-specific through the identification of interactors that are also members of dysregulated co-expression modules in ASD cortex from the study by Parikshak et al.[50]. In particular, we filtered for Tsc2 or mTOR interactors that are also members of differentially expressed genes in autism. As our analyses are based on interrogation of gene expression levels across the brain, we used gene co-expression modules from[50] as this list contains genes that have been shown to be differentially expressed in brain tissue from ASD individuals, i.e., it contains gene expression values that are directly relatable to our rsfMRI measurements of interest. This pruning step was crucial to narrow the broad set of interactors to only those with evidence of dysregulation in ASD postmortem cortical tissue, thus reducing the initial list of $n = 585$ interactors to a final set of $n = 88$ genes. The resulting $n = 88$ gene lists are thus composed of genes that interact with mTOR or Tsc2 at the protein level, and are dysregulated in ASD. Finally, to establish a possible link between mTOR–Tsc2 dysregulation and fMRI hyperconnectivity maps, we carried out a gene enrichment analysis. This step tests whether the list of $n = 2142$ genes that exhibit spatial correlation with our maps is significantly enriched for mTOR–Tsc2 ASD-relevant interactors as per our $n = 88$ list. The enrichment analysis was implemented with custom code that computes enrichment by using odds ratios and hypergeometric $P$ values https://github.com/mvlombardo/utils/blob/master/genelistOverlap.R.

To assess the specificity of our gene enrichment results, we conducted a control analysis by using CHD8 as a seed gene for the PPI analysis. This analysis was done to test whether any autism-relevant gene might result in similar types of enrichment results. Utilizing CHD8-interacting genes as a control analysis was done because of the similarly high relevance of CHD8 to autism and because CHD8 mutations similarly result in rsfMRI hyperconnectivity[108]. However, CHD8 is a regulator of transcription, via chromatin remodeling, rather than directly altering translational control, as is the case for Tsc2 and mTOR. Thus, this type of control analysis represents a very high level and stringent look at the specificity of the enrichment results for genes of relevance to mTOR–Tsc2 translational dysregulation. The results of this analysis were far from statistical significance (OR = 0.46, $P = 0.91$), corroborating the specificity of mTOR–Tsc2-enrichment results.

*ASD subtyping.* Subject-specific insular seed-based connectivity matrices were vectorized to create a matrix of participants x voxels. To isolate subgroups of ASD children with similar insular functionally connectivity patterns, we next computed subject-wise similarity matrices using Euclidean distance as the metric of similarity[82] and then applied agglomerative hierarchical clustering, as implemented in the R package "gplots" (http://cran.r-project.org/web/packages/gplots/index.html), to the subject-wise similarity matrices. To visualize the degree of similarity between individuals, we used dendrograms to reorder participants by similarity and to demarcate clusters. To determine the optimal number of clusters, we used NbClust[109] and we searched up to ten possible cluster solutions. The indices calculated by NbClust revealed the presence of either 4 (6/16) or 2 (5/16) as the top solutions for the optimal number of clusters. Using a majority vote rule, we retained 4 as the optimal cluster solution for the following enrichment analysis. To rule out the possibility that the clustering process could be partially affected by head motion then resulting in clusters characterized by different levels of motion, we carried out post hoc intercluster comparisons of mean FD by using unpaired two-tailed Student's $t$ test. This analysis yielded non-significant results ($P > 0.24$, all

clusters) and confirmed that none of the clusters had significantly higher head motion compared to others. Next, to identify how these ASD subgroup clusters differ from TD in functional connectivity, we compared the seed-based maps of each ASD subgroup against those of TDs by using unpaired two-tailed Student's $t$ test ($|t| > 2$, $P < 0.05$). Finally, to identify the ASD subtype enrichments with Tsc2/mTOR interactome genes, we carried out a similar gene expression decoding and enrichment analysis (see above) for each cluster separately, and we then applied FDR correction ($q < 0.05$)[110] to all enrichment $P$ values from the multiple comparisons.

**Reporting summary.** Further information on research design is available in the Nature Research Reporting Summary linked to this article.

## Data availability

The authors declare that mouse rsfMRI scans are publicly available at OpenNeuro in BIDS format, https://doi.org/10.18112/openneuro.ds003804.v1.0.0. De-identified human MRI and phenotypic data are publicly available at the following public repository: http://fcon_1000.projects.nitrc.org/indi/abide/abide_I.html. The authors also declare that gene expression data are publicly available on the web portal of Allen Brain Institute (https://human.brain-map.org/static/download). Source data are provided with this paper.

## Code availability

The code used for preprocessing mouse rsfMRI data is available at https://github.com/functional-neuroimaging/rsfMRI-preprocessing. The code employed for preprocessing human rsfMRI data is available at http://bit.ly/23u2vZp. Code for mapping global connectivity in mice and humans is available at https://github.com/functional-neuroimaging/rsfMRI-global-local-connectivity. The code employed for gene enrichment analysis is available at https://github.com/functional-neuroimaging/gene_decoding_and_enrichment.

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

## Acknowledgements
This work was supported by Simons Foundation Grants (SFARI 400101) to A. Gozzi. A. Gozzi was also supported by Brain and Behavior Foundation 2017 (NARSAD—National Alliance for Research on Schizophrenia and Depression) and the European Research Council (ERC—DISCONN, GA802371. M. Pagani was supported by the European Union's Horizon 2020 research and innovation programme under the European Union's Horizon 2020 research and innovation programme under grant agreement No. 845065 (Marie Sklodowska-Curie Global Fellowship - CANSAS). M.V. Lombardo was funded during this period of work by the European Research Council (ERC) under the European Union's Horizon 2020 research and innovation programme under grant agreement No 755816. V.M and K.S. were supported by grants from the Simons Foundation (SFARI 308939) and the NIH (MH084164). G. Deco was supported by AWAKENING Using whole-brain models perturbational approaches for predicting external stimulation to force transitions between different brain states (ref. PID2019-105772GB-I00, AEI FEDER EU) funded by the Spanish Ministry of Science, Innovation and Universities (MCIU), State Research Agency (AEI) and European Regional Development Funds (FEDER), HBP SGA3 Human Brain Project Specific Grant Agreement 3 (Grant Agreement No. 945539), funded by the EU H2020 FET Flagship program, and SGR Research Support Group support (ref. 2017 SGR 1545), funded by the Catalan Agency for Management of University and Research Grants (AGAUR). L. Ulysse was supported by BRAIN-CON-NECTS: Brain Connectivity during Stroke Recovery and Rehabilitation (Id. 201725.33), Funded by the Fundacio La Marato de TV3 2017.

## Author contributions
A.Go conceived and supervised the study, with input from M.Pas. and M.V.L. A.Ga, M.Pag performed mouse brain imaging. M.Pag carried out functional connectivity mapping and FA quantification in mice. N.B., A.B., and M.Pas carried out spine counting, viral tracing, and immunostaining. S.T. and M.V.L. preprocessed human neuroimaging data. L.U. and G.D. performed whole-brain mathematical modeling. A.L., I.M., and R.T. carried out electro-physiology measurements. A.D.F. and C.C. performed pharmacological treatments in mice. A.D.F. carried out mouse behavioral assessment. M.Pag, K.S., and V.M. carried out functional connectivity mapping in humans. M.V.L. and M.Pag performed gene enrichment analysis. A.Go and M.Pag wrote the manuscript, with inputs from M.V.L., M.Pas, and V.M. All authors read and approved the final version of the manuscript.

## Competing interests
The authors declare no competing interests.
