## [Peer Review File · Nature Communications]

Reviewers' Comments:

Reviewer #1:

Remarks to the Author:

This is a very well-done study using rsfMRI, electrophysiology, and in silico modelling in Tsc2 haplo-insufficient mice, demonstrating that mTOR-dependent increases in neuronal spine density is associated with autism-like stereotypies and cortico-striatal hyperconnectivity. Deficits, however, are completely reversed through the pharmacological inhibition of mTOR.

Additionally, the authors show further that children diagnosed with idiopathic autism present parallel cortical-striatal hyperconnectivity, noting that such a "connectivity fingerprint" is supplemented for autism-dysregulated genes interacting with mTOR or TSC2.

Finally, the study illustrates that the transcriptomic profile is expressed in a subset of children diagnosed with autism - potentially defining an identifiable subtype of autism spectrum disorder.

The findings of these authors indicate a potentially causal link involving mTOR-related synaptic pathology and large-scale alterations in functional brain networks. The major take away from this study is the potential for a unifying, multi-scale framework which may mechanistically reconcile developmental synaptopathy and functional hyperconnectivity in children diagnosed as being on the autism spectrum.

This study involving experimental animals as well as involving human fMRI is unique and has the potential to be one which can change the direction of autism research. The theory of hyperconnectivity in the brain of children diagnosed with autism has attracted much attention. But the absence to date of a clear mechanism producing such an effect has limited understanding of the disorder and how it, thus, might be treated. mTOR-dependent increases in neuronal spine, as demonstrated in the mouse, is a unique model for so doing. That idiopathic autism presents a similar increased level of cortical-striatal connectivity, as measured using resting-state fMRI, and noting the relationship to genes associated with mTOR. Such a relationship offers an intriguing mechanism to understanding autism, its genetic contributions, the role of mTOR's function as a serine/threonine protein kinase regulating cell growth, spine density, proliferation, motility, survival, protein synthesis, autophagy, and transcription and where these processes are dysfunctional in autism.

Apart from a few minor spelling errors, this is an excellent manuscript describing a study of much interest to the autism spectrum disorders research community. The examination using both mice as well as human evidence is compelling. It is highly appropriate for Nature Communications. I enjoyed reading it and I am confident that others will also.

Several additional articles the authors may wish to cite include:

- Winden, KD, Ebrahimi-Fakhari, D, Sahin, M. "Abnormal mTOR Activation in Autism". *Annu Rev Neurosci.* 2018;41:1-23. doi: 10.1146/annurev-neuro-080317-061747. PubMed PMID: 29490194.
- Sato, A. "mTOR, a Potential Target to Treat Autism Spectrum Disorder". *CNS Neurol Disord Drug Targets.* 2016;15(5):533-43. doi: 10.2174/1871527315666160413120638. PubMed PMID: 27071790; PMCID: PMC5070418.
- Lee, DY. "Roles of mTOR Signaling in Brain Development". *Exp Neurobiol.* 2015;24(3):177-85. doi: 10.5607/en.2015.24.3.177. PubMed PMID: 26412966; PMCID: PMC4580744.
- Hull, JV, Dokovna, LB, Jacokes, ZJ, Torgerson, CM, Irimia, A, Van Horn, JD. "Resting-State Functional Connectivity in Autism Spectrum Disorders: A Review". *Front Psychiatry.* 2016;7:205. doi: 10.3389/fpsy.2016.00205. PubMed PMID: 28101064; PMCID: PMC5209637.

Reviewer #2:

Remarks to the Author:

The paper by Pagani et al. reports data on both mice and humans suggesting a link between

mTOR-related synaptic pathology and functional hyperconnectivity in autism. This is a very well written article and the results are interesting supporting previous hypotheses on atypical connectivity in autism. I have no major criticisms but several suggestions:

The impact of the loss of Tsc2 on dendrites is not novel and I am surprised that the authors did not quote the article by Tavazoie et al. Regulation of neuronal morphology and function by the tumor suppressors Tsc1 and Tsc2, Nature Neuroscience 2005. This paper is to my knowledge the first to report that loss of Tsc1 or Tsc2 triggered enlargement of somas and dendritic spines and altered the properties of glutamatergic synapses.

Regarding the brain MRI data, there is no reference on the neuroanatomical abnormalities observed in patients carrying deleterious TSC2 mutations. A recent review is Russo et al. Neuroimaging in tuberous sclerosis complex Child's Nervous System, 2020.

The choice of the default mode network (DMN) and salience network (SN) is indeed relevant to autism, but the author could indicate in more details the overlap between these two networks. And whether the changes they observed in the Tsc2 mice concern the overlapping brain regions shared by the networks or some regions that are more specific to each network. There are some discussion about this in the current version, but more emphasis on the shared abnormalities could be interesting.

it would be also interesting to compare the diffusion tensor imaging (DTI) results obtained here with those obtained using the same approach by Hsieh et al. (Detection of endophenotypes associated with neuropsychiatric deficiencies in a mouse model of tuberous sclerosis complex using diffusion tensor imaging Brain Pathol, 2020).

It is indicated that the rs fMRI « were preprocessed as previously described ». the authors could provide more information and justify why they chose the voxel-wise approach.

#. It is not clear if the authors used GSR (Global signal regression). This is currently advised to run the analyses with or without to have more robust results.

The authors suggest that "the lack of major structural anatomical alterations predictive of the observed rsfMRI hyperconnectivity points at a possible mechanistic link between mTOR-related spine surplus and increased rsfMRI coupling in Tsc2+/- mutants". I would like to know if this is possible to observe tubers in the brain of the Tsc2+/- mutant mice? It is unlikely that they that could alter specific networks, but since these tubers are observed in patients, it would be important to have the information and eventually to rule out this possibility.

For the Neurosynth and NeuroVault, as indicated by the authors, the dataset is based on only 6 brains from the Allen Brain. This is a concern, but this is to date the only data available. Since the authors used STRING to identify the interactors of TSC2, it seems that they selected all kind of interactors, not only binding partners, but also co-expressed genes. This could be circular to choose a set of co-expressed genes and to find that they are co-expressed in the brain regions of interest. It would be informative to test different confidence levels and types of interactions. A supplementary figure on this aspect would be nice. I really appreciated the gene enrichment analyses using CHD8 as a control for the specificity of the TSC2 network.

Difference to anesthesia sensitivity between mutant and wild-type mice has been tested, but since functional connectivity is very sensitive to anesthesia, the authors might discuss the fact that fMRI humans studies are performed in majority without anesthesia in contrast to their protocol.

I appreciate that the authors usually indicate that their results might be true only for a subset of patients. I would therefore suggest to indicate this in the abstract " a subset of" before "children with autism". By the way, several MRI studies supporting the hypothesis were performed in adults so "individuals" might be more correct than "children".

Reviewer #3:

Remarks to the Author:

In their study, Pagani et al conduct a series of analyses across micro- and macro-scales in the Tsc2 mouse model of Tuberous sclerosis and idiopathic ASD in humans with the goal of establishing a causal mechanistic link between mTOR-synaptic alterations and functional connectivity deficits in ASD.

The clear strength of this study is the multidisciplinary tools across mouse and human to assess phenotypes from the level of the synapse to whole brain connectivity. The application of these multilevel tools represents a level of novelty, revealing new ways of looking at Tsc+/- phenotypes with putative links to humans. The manuscript is also very well written. The connectivity findings using rsfMRI and associated rescue of this phenotype in the Tsc+/- mouse model is the most novel finding, and also most convincing. This finding in particular will be of high value to the field.

Tempering these strengths are notable weaknesses. The first is the above described novelty is somewhat overshadowed by the fact the the mTOR pathway and many associated readouts presented, including the pharmacological rescue experiments, have been well explored in previous studies. Beyond this, and most importantly, in an number of experimental instances (described below in more detail), the data is only cursorily presented, reducing confidence that the experiments have the needed rigor to reach the conclusions presented. This is especially critical since the authors strongly claim a mechanistic link between micro-scale synaptic deficits and macro-scale connectivity deficits, a link for which other mechanisms have not been fully ruled out.

The addition of human subjects is also a strength of the study, but considering the tremendous heterogeneity in connectivity findings from a number of studies, there are inherent limitations of how to extrapolate the findings to: 1) linking directly to experiments presented in Tsc+/- mice and 2) how widely they are applicable across idiopathic ASD. In this instance, if possible, it would have been more applicable to study humans with TSC.

More specifics regarding concerns are:

1) While in general the experiments appear to be well executed, the link between spine abnormalities and more large scale connectivity and behavioral deficits needs more experimental evidence. For example, other deficits that additionally or more likely may be causal include axonal placement/survival (including interneurons), axonal integrity or myelination, all of which have been observed in Tsc+/- mice and cannot be ruled out. While the authors begin to address the status of myelin and axonal pathways in Figure 2, the data presented are insufficient to support their claims of lack of myelin or axonal deficits. Myelin deficits can be directly assessed by myelin stains and retrograde axonal tracing data should be shown in a separate figure with consisting of visualization of axon pathways, and appropriate quantification and controls for size of infection/ number of starter cells infected

2) The rapamycin rescue of connectivity deficits as assessed by rsfMRI in Figure 3 is robust and impressive. While the authors show spine abnormality rescue by mTOR inhibition, the connectivity deficits can also be explained by putative rescue of above described phenotypes, which are unexplored

3) The effort to link regional gene alterations in ASD as shown in Figures 4 & 5 is a worthwhile effort, though the input ASD gene list is of critical importance. Here it appears that the Parikshak et al, 2016 gene list was used. The rationale for the choice of this gene set is not clear. It is suggested that the authors instead use the Satterstrom et al 2020 gene set which represents the largest cohort to date, and perhaps also the SFARI gene set (though this is not without its caveats

Reviewer #4:

Remarks to the Author:

General comments:

Postmortem studies from several independent groups have collectively revealed increased density

of excitatory synapses in the brains of individuals with autism. Neuroimaging studies reported aberrant hyperconnectivity in large scale networks. This study investigates potential causal link between excess of synapses and aberrant functional connectivity in a mice model and in human subjects. Using rsfMRI, electrophysiology and in silico modelling in *Tsc2* haploinsufficient mice, the authors claim that mTOR-dependent increased spine density is associated with autism-like stereotypies and cortico-striatal hyperconnectivity; and that these deficits are rescued by pharmacological inhibition of mTOR. The authors performed similar neuroimaging analysis in children with idiopathic autism, and found analogous cortical-striatal hyperconnectivity. They further showed that transcriptomic signature was predominantly expressed in a subset of children with autism, thereby defining a segregable autism subtype.

This is potentially an important paper. Being able to establish a causal link between non-invasive macroscopic measure (aberrant functional MRI network) in ASD patients and cellular mechanism and transcriptomic signature, and making cross-species link could have profound implications in early diagnosis and treatment prognosis. However, some of the data presented in this manuscript do not appear to be convincing. I am afraid substantial work is needed to make a more convincing case.

1. Technically, there is a lack of clarity in fMRI data analysis, in both human and animal fMRI data. There appears to be some confusion in terminology as well. On page 22, it was stated "..... we calculated voxel-wise long-range connectivity maps for all mice. Long-range connectivity is a graph-based metric also known as unthresholded weighted degree centrality and defines connectivity as the mean temporal correlation between a given voxel and all other voxels within the brain (Cole et al., 2010)." The connectivity reported in Cole's paper was "global connectivity", which is basically the mean temporal correlation between a given voxel and all other voxels within the brain. Additionally, although conceptually related, it is different from degree of centrality as defined in graph theory. Furthermore, it is unclear how seed-based analysis was conducted. It was stated "VOIs were placed in the prefrontal cortex to map derangements ..." It was later stated "Antero-posterior default mode network (DMN) and salience network (SN) connectivity alterations were probed by computing seed-to-VOI correlations." As a result, the reviewer can only guess what information Fig. 1B-D conveys.

2. Spatial specificity of synaptic alterations in *Tsc2*+/- mutants. Figure 1A shows spine density counting. But only in insular cortex. There is no mention whether alterations in spine density was region-specific or a global phenomenon. Considering the fMRI measures appear to implicate multiple brain networks, it would seem particularly relevant in order to determine whether "mTOR-dependent synaptic pathology is associated with a specific brain-wide signature of functional dysconnectivity", as stated by the authors.

3. Figure 3 shows the effect of pharmacological mTOR inhibition on synaptic surplus, functional hyperconnectivity and autism-like behavior in *Tsc2*+/- mice. There are 4 cohort of animals, 2 animal groups \times 2 treatment (rapamycin vs vehicle). It appears that t-stats were applied to analyze the data. Would ANOVA followed by post-hoc analysis be more appropriate? Also, unless I have missed, nowhere in the manuscript mentioned how "rescue" stats (blue contrast) was generated. Also, through out this manuscript, statistical threshold was set at 2-tailed t-test at $P < 0.05$, this is relatively liberal. More specifics should be give about multiple comparisons.

4. Human fMRI data analysis. Motion artifacts are a practical problem in fMRI with ASD patients. The author clearly recognized this problem, and tried to deal with it. However, the subtypes showing Fig. 5C, especially in subtype 1, and subtype 3, the maps do not appear to have anatomical specificity. The reviewer wonders whether residual motion played a role. To remove non-neuronal artefacts in the resting-state fMRI data, the covariates in confound signal regression of human fMRI data typically will include white matter signal in addition to CSF signal, and may include not just the average WM and CSF signal, but also their additional variations projected to higher-order principal components. I suggest inclusion of WM signals as covariates. On page 31, the authors performed data scrubbing to remove time points using stringent FD threshold. It's not clear after scrubbing, the statistics of remaining number of time points for each group, e.g. how many volumes remain after scrubbing for each group (range, mean +/- standard deviation etc) and whether there is group difference.

There is a reasonable literature on human insular function connectivity (e.g. Deen B et al., cerebral cortex, 2011; Di Martino A et al. American Journal of Psychiatry. 2009) . The anterior, middle and posterior insular have well established connectivity profile. For sanity check, it would be more convincing if the authors can present insular connectivity maps, and cross check with literature.

5. Perhaps the weakest point of this manuscript is in-silico modelling that the authors claim to predict mTOR-related functional hyper-connectivity through increased interareal synaptic coupling. While the hypothesis is fascinating, the data presented is nothing more than mathematical exercise. There are numerous assumptions from synaptic excitability, to membrane potential fluctuation, to field potential and neuronal firing, and finally through neurovascular coupling (a black box) to BOLD fluctuation. Even with simultaneously acquired LFP/spiking and BOLD data, the neuroimaging field cannot agree on how to relate BOLD fluctuation to spontaneous neuronal activity. I feel it is exceedingly speculative to make such a claim.

We would like to thank the Editor and Reviewers for their positive evaluations, constructive comments, and for the opportunity to submit a revised manuscript. We believe the resulting paper is greatly improved following their suggestions. Please find below a point-by-point response to the reviewers' comments (our response in **blue** typeface). We have also added below the text edits we have introduced in the manuscript in response to the reviewers' comments (**yellow highlight**).

In response to the editorial note, we have also added a data and code availability statement as follows:

Data and code availability statement: The authors declare that human MRI and phenotypic data is available in the following public repository http://fcon_1000.projects.nitrc.org/indi/abide/, and that mouse fMRI scans are available from the corresponding author upon reasonable request. All the other data supporting the findings of this study are available within the paper and its supplementary information files. The authors also declare that all the employed code is either publicly available (FSL, ANTs and AFNI), or can be obtained from the corresponding author upon reasonable request.

Reviewer #1 (Remarks to the Author):

This is a very well-done study using rsfMRI, electrophysiology, and in silico modelling in Tsc2 haplo-insufficient mice, demonstrating that mTOR-dependent increases in neuronal spine density is associated with autism-like stereotypies and cortico-striatal hyperconnectivity. Deficits, however, are completely reversed through the pharmacological inhibition of mTOR. Additionally, the authors show further that children diagnosed with idiopathic autism present parallel cortical-striatal hyperconnectivity, noting that such a "connectivity fingerprint" is supplemented for autism-dysregulated genes interacting with mTOR or TSC2. Finally, the study illustrates that the transcriptomic profile is expressed in a subset of children diagnosed with autism - potentially defining an identifiable subtype of autism spectrum disorder.

The findings of these authors indicate a potentially causal link involving mTOR-related synaptic pathology and large-scale alterations in functional brain networks. The major take away from this study is the potential for a unifying, multi-scale framework which may mechanistically reconcile developmental synaptopathy and functional hyperconnectivity in children diagnosed as being on the autism spectrum.

This study involving experimental animals as well as involving human fMRI is unique and has the potential to be one which can change the direction of autism research. The theory of hyperconnectivity in the brain of children diagnosed with autism has attracted much attention. But the absence to date of a clear mechanism producing such an effect has limited understanding of the disorder and how it, thus, might be treated. mTOR-dependent increases in neuronal spine, as demonstrated in the mouse, is a unique model for so doing. That idiopathic autism presents a similar increased level of cortical-striatal connectivity, as measured using resting-state fMRI, and noting the relationship to genes associated with mTOR. Such a relationship offers an intriguing mechanism to understanding autism, its genetic contributions, the role of mTOR's function as a serine/threonine protein kinase regulating cell growth, spine density, proliferation, motility, survival, protein synthesis, autophagy, and transcription and where these processes are dysfunctional in autism.

Apart from a few minor spelling errors, this is an excellent manuscript describing a study of much interest to the autism spectrum disorders research community. The examination using both mice as well as human evidence is compelling. It is highly appropriate for Nature Communications. I enjoyed reading it and I am confident that others will also. Several additional articles the authors may wish to cite include:

- Winden, KD, Ebrahimi-Fakhari, D, Sahin, M. "Abnormal mTOR Activation in Autism". *Annu Rev Neurosci*. 2018;41:1-23. doi: 10.1146/annurev-neuro-080317-061747. PubMed PMID: 29490194.
- Sato, A. "mTOR, a Potential Target to Treat Autism Spectrum Disorder". *CNS Neurol Disord Drug Targets*. 2016;15(5):533-43. doi: 10.2174/1871527315666160413120638. PubMed PMID: 27071790; PMCID: PMC5070418.
- Lee, DY. "Roles of mTOR Signaling in Brain Development". *Exp Neurobiol*. 2015;24(3):177-85. doi: 10.5607/en.2015.24.3.177. PubMed PMID: 26412966; PMCID: PMC4580744.
- Hull, JV, Dokovna, LB, Jacokes, ZJ, Torgerson, CM, Irimia, A, Van Horn, JD. "Resting-State Functional Connectivity in Autism Spectrum Disorders: A Review". *Front Psychiatry*. 2016;7:205. doi: 10.3389/fpsy.2016.00205. PubMed PMID: 28101064; PMCID: PMC5209637.

We thank the reviewer for the words of appreciation. We have carefully read the manuscript in all of its parts and identified a few spelling errors and typos, which we have amended in the revised version. Furthermore, we have updated our reference list based on the suggested articles as follows:

Introduction: "mTOR is a key regulator of synaptic protein synthesis (Sato, 2016), and aberrations in mTOR signaling have been linked to synaptic and neuroanatomical abnormalities (Tavazoie et al., 2005; Russo et al., 2020) that are associated with both syndromic (e.g. tuberous sclerosis, Sahin and Sur, 2015; Winden et al., 2018) and idiopathic ASD (Lee, 2015; Gazestani et al., 2019; Rosina et al., 2019). Further mechanistic investigations by..."

Introduction: "At the macroscopic level, several studies have highlighted the presence of aberrant functional connectivity in ASD as measured with resting state fMRI (rsfMRI) (Supekar et al., 2013b; Uddin et al., 2013; Hull et al., 2017)"

Reviewer #2 (Remarks to the Author):

The paper by Pagani et al. reports data on both mice and humans suggesting a link between mTOR-related synaptic pathology and functional hyperconnectivity in autism. This is a very well written article and the results are interesting supporting previous hypotheses on atypical connectivity in autism.

We thank the reviewer for the positive comment and for the detailed list of constructive suggestions.

I have no major criticisms but several suggestions:

The impact of the loss of Tsc2 on dendrites is not novel and I am surprised that the authors did not quote the article by Tavazoie et al. Regulation of neuronal morphology and function by the tumor suppressors Tsc1 and Tsc2, Nature Neuroscience 2005. This paper is to my knowledge the first to report that loss of Tsc1 or Tsc2 triggered enlargement of somas and dendritic spines and altered the properties of glutamatergic synapses.

We thank the reviewer for pointing out this omission. We have added this work to the reference list in the introduction section as follows:

Introduction: “mTOR is a key regulator of synaptic protein synthesis (Sato, 2016), and aberrations in mTOR signaling have been linked to synaptic and neuroanatomical abnormalities (Tavazoie et al., 2005; Russo et al., 2020) that are associated with both syndromic (e.g. tuberous sclerosis, Sahin and Sur, 2015; Winden et al., 2018) and idiopathic ASD (Lee, 2015; Gazestani et al., 2019; Rosina et al., 2019). Further mechanistic investigations by...”

Regarding the brain MRI data, there is no reference on the neuroanatomical abnormalities observed in patients carrying deleterious TSC2 mutations. A recent review is Russo et al. Neuroimaging in tuberous sclerosis complex Child's Nervous System, 2020.

We added this work to the reference list in the introduction section as follows:

Introduction: “mTOR is a key regulator of synaptic protein synthesis (Sato, 2016), and aberrations in mTOR signaling have been linked to synaptic and neuroanatomical abnormalities (Tavazoie et al., 2005; Russo et al., 2020) that are associated with both syndromic (e.g. tuberous sclerosis, Sahin and Sur, 2015; Winden et al., 2018) and idiopathic ASD (Lee, 2015; Gazestani et al., 2019; Rosina et al., 2019). Further mechanistic investigations by...”

The choice of the default mode network (DMN) and salience network (SN) is indeed relevant to autism, but the author could indicate in more details the overlap between these two networks. And whether the changes they observed in the Tsc2 mice concern the overlapping brain regions shared by the networks or some regions that are more specific to each network. There is some discussion about this in the current version, but more emphasis on the shared abnormalities could be interesting.

We thank the reviewer for this comment, which gives us the possibility to clarify this important point. Default mode network (DMN) and salience network (SN) are two evolutionarily-conserved brain systems that are segregable in multiple mammalian species via rsfMRI connectivity mapping (Menon, 2011). Some anatomical intersection between the two systems is appreciable in fronto-limbic and striatal areas in rodents (Lu et al., 2012; Gozzi and Schwarz, 2016; Tsai et al., 2020), and, to a lower extent, also in non-

human primates (Mantini et al., 2011; Touroutoglou et al., 2016) and humans (Sripada et al., 2012; Goulden et al., 2014). Such overlap is to be expected given that the anterior cingulate cortex (defined as medial prefrontal cortex – PFC, in our manuscript) is a hub-like core component of both the SN and DMN in rodents (Grandjean et al., 2020; Whitesell et al., 2021), and a relay-station connecting striatal circuits to higher order association cortices (Coletta et al., 2020). The larger overlap in rodents is also consistent with the lower extension and reduced regional differentiation of the cortical mantle in this species with respect to primates and human (Buckner and Krienen, 2013).

To assess the network specificity (and shared abnormalities) of the functional connectivity changes observed in *Tsc2*^{+/-} mice, we regionally compared and quantified the spatial intersection between the DMN and SN using Dice Similarity Coefficient (DSC). DSC provides a voxel-wise estimate of spatial overlap between parametric maps (Pagani et al., 2016b), yielding a similarity value that ranges between 0 (when there is no overlap) and 1 (when the two maps are identical). We next extended this analysis to quantify the proportion of voxels in between-group over-connectivity maps that exhibit network-specific location, with respect to those affecting common neuroanatomical substrates.

DCS quantifications of DMN and SN yielded a DSC score of 0.41, with regions of overlap being most prominently located in anterior cingulate, insular and dorso-striatal areas (as appreciable in seed maps in Fig. 1C, D). DMN-SN overlap was slightly broader in *Tsc2*^{+/-} mutants (DSC = 0.54), a result consistent with the stronger and more extended PFC-striatal connectivity, and a larger extension of the DMN in these mice. A spatial comparison of SN-DMN overlap with SN and DMN hyper-connectivity maps in *Tsc2* mutants (Fig. 3C-D, bottom row) revealed that 62% of the foci of hyper-connectivity were SN-specific and 55.7% were instead DMN-specific. Areas of shared abnormalities mostly encompassed striatal regions, the PFC, retrosplenial cortex and insular areas, to broadly recapitulate the pattern of increased long-range connectivity reported in Fig (1B). These findings corroborate the presence of core fronto-cortico-striatal connectivity pathology in *Tsc2* mutants and human individuals with ASD, suggesting that aberrant coupling between these substrates can differentially affect the topographical organization of salience and default mode networks.

In the revised manuscript we have added a brief account of these analyses as follows:

Methods: “The statistical significance of intergroup changes was quantified using an unpaired 2-tailed Student’s t-test ($p < 0.05$). To assess the network specificity of the rsfMRI changes observed in *Tsc2*^{+/-} mice, we regionally quantified the spatial intersection between the DMN and salience network (SN) using Dice Similarity Coefficient (DSC) (Pagani et al., 2016b). DCS quantifications yielded a score of 0.41, and 0.54 in control and *Tsc2*^{+/-} mutants, respectively. A spatial comparison of SN-DMN overlap with SN and DMN hyper-connectivity maps in *Tsc2* mutants (Figure 3C-D, bottom row) revealed that 62% of the foci of hyper-connectivity were SN-specific and 55.7% were DMN-specific. Areas of shared abnormalities mostly encompassed striatal regions, the PFC, retrosplenial cortex and insular areas, to recapitulate the pattern of increased long-range connectivity reported in Figure 1B.”

Discussion: “...Taken together, these findings advance our understanding of mTOR-related ASD pathology in humans, and identify a segregable subtype of ASD characterized by increased fronto-insular-striatal connectivity.

An interesting result of our investigation is the presence of hyper-connectivity in shared substrates of the salience and default-mode network, such as the medial prefrontal cortex. While these two networks are clearly segregable with rsfMRI connectivity mapping, a degree of anatomical intersection between the two systems is apparent in rodents (Gozzi and Schwarz, 2016; Whitesell et al., 2021) and, to a lower extent, in humans (Goulden et al., 2014). The larger anatomical overlap in rodents is consistent with the lower

extension and reduced regional differentiation of the cortical mantle in this species (Buckner and Krienen, 2013), and with the anatomical configuration of the rodent anterior cingulate as a relay-station linking striatal circuits to other higher-order association cortices (Coletta et al., 2020). In keeping with this notion, rsfMRI connectivity abnormalities in *Tsc2* mutants mostly encompassed striatal regions, the PFC, the retrosplenial cortex and insular areas (Figure 1B). These results delineate fronto-cortico-striatal hyperconnectivity as a core feature of mTOR-related connectopathy, and suggest that aberrant coupling between these substrates can differentially affect the topographical organization of ASD-relevant networks such as the salience and default mode networks.

Our findings are broadly consistent with EEG data in children harboring *Tsc2* mutations...”

It would be also interesting to compare the diffusion tensor imaging (DTI) results obtained here with those obtained using the same approach by Hsieh et al. (Detection of endophenotypes associated with neuropsychiatric deficiencies in a mouse model of tuberous sclerosis complex using diffusion tensor imaging *Brain Pathol*, 2020).

We thank the reviewer for pointing out this study. In keeping with our results, Hsieh and colleagues did not find any fractional anisotropy (FA) alterations in white matter tracts of *Tsc2*^{+/-} mice as assessed with *in vivo* DTI. This finding was corroborated by Myelin Basic Protein (MBP) densitometric staining, which did not reveal any change in MBP signal across genotypes. These results are in agreement with our investigations, arguing against the presence of gross macroscale white matter integrity alterations in this mouse line. The same authors however described foci of altered FA and mean diffusivity in gray matter areas such as the cingulate cortex, or dorsal hippocampus. The authors further reported that in these regions the coherency of myelinated axons appeared to be increased in *Tsc2* mutants, an effect that was interpreted as putative loss of neurite complexity in this mouse line.

Prompted by these findings, we have now followed the procedure described by Hsieh et al. (2021), and assessed myelin integrity and coherency in a major inter-callosal white matter bundles (corpus callosum) as well as in gray matter regions (e.g. cingulate cortex and dorsal hippocampus) of *Tsc2* mutants with and without rapamycin treatment. We report the results of these new investigations below (Supplementary Figure S2 in revised manuscript). Corroborating the notion of largely preserved myelin integrity and macroscale white matter structure in these mice, our new investigations did not reveal any genotype-dependent densitometric changes in MBP in any of the areas examined. Similarly, but in contrast to the results from Hsieh and associates, our assessments did not reveal any genotype-dependent MBP coherence alterations in any of the examined regions. Taken together, these findings corroborate our prior claims of a lack of major white matter alterations or reorganization in *Tsc2* mutants.

A number of technical considerations could account for the discrepant FA findings in our work with respect to the results described by Hsieh et al. First, while FA is extensively used to assess the integrity of major white matter bundles, the interpretation and significance of this parameter in gray matter areas (like in Hsieh et al.) is non-customary, and highly debated. It is well known that, owing to the sharp contrast between FA in white and gray matter, minor image co-registration issues can result in partial volume effects that can critically contaminate low-FA gray matter signal (Smith et al., 2006a; Schwarz et al., 2014). To avoid this problem, we employed state-of-the-art skeletonization registration via tract-based spatial statistics – TBSS (Smith et al., 2006b), a procedure that retains and registers high intensity FA-white matter voxels, resulting in a better alignment of high FA tracts, considerably reducing partial-volume and co-registration issues. Moreover, the substantially coarser spatial and tensor resolution of the FA assessment employed by Hsieh et al. (400 x 400 x 400 vs. 128 x 128 x 256 μm^3 ; and 10 vs 80 directions, respectively) make their measurements intrinsically more susceptible to partial volume contributions, an effect that

might be compounded by different statistical power of the two studies (N=11 vs N=20, respectively). Notwithstanding these technical differences, in both studies MBP staining of major white matter tracts did not reveal signs of altered white matter integrity and structure, corroborating our original claims of an overall preservation of white matter organization in Tsc2 mutants.

The significance of discrepant findings in gray matter MBP coherence between the two studies is unclear. Our analyses did not highlight any sign of such alterations, both in areas of high axonal coherence (corpus callosum), as well as in gray matter areas with less organized axonal structure (cingulate and hippocampus), arguing against a contribution of this putative axonal phenotype to our rsfMRI imaging findings. Supporting this notion, it should also be noted that Hsieh et al reported putatively altered MBP coherence only in the dorsal hippocampus of Tsc2 mice, i.e. a brain regions that does not exhibit rsfMRI connectivity alterations in our Tsc2 mutants. Taken together, these considerations argue against a substantial contribution of macro- and micro-scale white matter alterations to our imaging results.

Supplementary Figure S2. Myelin basic protein staining. A) Representative sections showing myelin basic protein (MBP) staining in different brain regions. B) Myelin basic protein densitometry in the corpus callosum (CC, top panel), cingulate cortex (Cg, middle panel) and hippocampus (HPC, bottom panel) in control and Tsc2^{+/-} mice treated with vehicle or rapamycin (rapa). C) MBP coherence in CC, Cg, and HPC in control or Tsc2^{+/-} mice treated with vehicle or rapamycin. ROD: relative optical density value.

In the revised manuscript we have reported these new data in Supplementary Figure S2 and briefly discussed these results as follows.

Results: “Voxel-wise and regional assessments of fractional anisotropy (FA), a parameter sensitive to **microscale** white matter integrity (Bertero et al., 2018), revealed the presence of largely preserved microstructure in all the major fiber tracts of Tsc2^{+/-} mutants ($q > 0.24$, FDR-corrected, Figure 2A). The lack of regional FA differences also argues against the presence of major alterations in whole-brain white matter topography, as these would be appreciable in the form of large regional FA differences (Dodero et al., 2013). **To rule out the presence of more subtle myelination deficits in Tsc2^{+/-} mice, we carried out measurements of myelin basic protein (MBP) density and coherence (Hsieh et al., 2021) using immunostaining methods. In keeping with our FA measurements, MBP staining did not show any**

genotype-dependent alterations in the corpus callosum or in gray matter regions (PFC, hippocampus) as assessed with densitometric quantifications and MBP coherency mapping (Supplementary Figure S2.) These results support the lack of gross micro- and micro-structural WM reorganization in these mice.

Methods: *MBP immunohistochemistry and imaging analysis*

To rule out the presence of myelination deficits in our mice, we measured MBP and MBP coherency in *Tsc2*^{-/-} and wild-type mice treated with and without rapamycin. Myelin basic protein (MBP) was stained using Rat anti MBP (Abcam ab7349). Briefly, 50 µm vibratome brain sections were incubated in blocking solution (5% Horse serum in PB-Triton 0.1%) for 1h at room temperature, and subsequently incubated overnight at 4° C with the primary antibody rat anti MBP (1:1000). The following day, four 1-hour washes in PB-Triton 0.1% were performed. Brain sections were then incubated overnight at 4° C with a donkey anti-Rat IgG (H+L) secondary antibody Alexa Fluor 488 conjugate (ThermoFisher A21208). Sections were rinsed in PB-Triton 0.1% and mounted with AquaPolymount (Polyscience). Control brain sections were processed using the same protocol, omitting primary antibody. Images of MBP immunostaining were acquired with a Nikon A1 confocal microscope in corpus callosum (CC), cingulate cortex (Cg) and hippocampus (HP). A 20x plan-apochromat objective was used to image three sections at comparable antero-posterior anatomical level (n = 3 animals for each experimental cohort, 0.7 µm Z-step, 17 steps).

Quantification of fluorescence intensity (optical density, OD) was performed using ImageJ software. To obtain relative optical density value (ROD), background value measured on control sections was subtracted to the OD value. From each section, three square 124x124x10 µm regions (e.g. corpus callosum and cingulate cortex) or 93 x 93 x 10 µm (layer lacunosum moleculare of hippocampal CA1) were analyzed (n = 3 sections per subject, n = 3 animals per cohort). Coherency of MBP immunostained fibers were computed as in (Hsieh et al., 2021) using the function measure of OrientationJ plugin of ImageJ software. Coherency was evaluated on three regions per section as described above (n=3 sections per subject, n=3 animals per cohort).

It is indicated that the rsfMRI « were preprocessed as previously described ». The authors could provide more information and justify why they chose the voxel-wise approach.

In the revised manuscript we have expanded the methodological description of all the preprocessing steps employed for our mouse rsfMRI connectivity analysis. To better outline each step of our preprocessing pipeline, we have also specified the neuroimaging tool we employed for each image processing step as follows:

Methods: “Raw rsfMRI timeseries were preprocessed as previously described (Sforazzini et al., 2014; Pagani et al., 2019). The procedure employed consists of two sequential components: core preprocessing and denoising. Core preprocessing includes the following steps (in brackets the software and function employed). The initial 50 volumes of the time series were removed to allow for T1 and gradient thermal equilibration effects (AFNI, 3dTcat). BOLD time-series were then despiked (AFNI, 3dDespike), motion corrected (FSL, mcflirt), skull-stripped (FSL, bet) and spatially normalized with affine and diffeomorphic registration (ANTS, antsRegistration + antsApplyTransforms) to a skull-stripped reference BOLD template. The denoising pipeline included the following steps: motion traces of head realignment parameters (3 translations + 3 rotations) and mean ventricular signal (corresponding to the averaged BOLD signal within a reference ventricular mask, FSL, fslmeants) were used as nuisance covariates and regressed out from each time course (FSL, fsl_regfilt). All rsfMRI time series also underwent band-pass filtering to a frequency window of 0.01 - 0.1 Hz (AFNI 3dBandpass) and spatial smoothing with a full width at half maximum of 0.6 mm (AFNI, 3dBlurInMask).”

The reasons for our use of voxel-wise measurements are three-fold: first, the use of voxelwise measurements allows for a spatially unbiased mapping of connectivity without the constraints of pre-imposed anatomical boundaries (Coletta et al., 2020). The lack of neuroanatomical constraints is an important strength of our study, as it permits us to maximize the spatial resolution of functional mapping and quantify topographical alterations in rsfMRI connectivity at a scale that would not be achievable with region-wise quantifications. Second, voxelwise measurements do not rely on the arbitrary choice of which and how many regions to use to parcel the mouse brain, reducing resolution bias in functional mapping (Wang et al., 2009). Third, we previously showed that voxelwise measurements and the resulting global connectivity mapping (Cole et al., 2010) allow for a smooth transition of findings across species (Bertero et al., 2018). It should also be emphasized that all the voxelwise assessments of our manuscript were complemented by quantifications in regions of interest to quantify global connectivity measurements (e.g. Figure 1B and Figure 3B) and seed-based measurements (e.g. plots in Figure 1CD, 3CD and 4A-B-C). These region-wise quantifications allowed us to describe the distribution of connectivity across subjects, providing a visual estimate of the corresponding effect sizes.

In the revised text, we have added the following short paragraph to the method section to better highlight the advantages of measuring functional connectivity at the voxel level.

Methods: "... mean temporal correlation between a given voxel and all other voxels within the brain (Cole et al., 2010). Voxelwise rsfMRI measurements allow for a spatially unbiased mapping of connectivity without the constraints of pre-imposed anatomical boundaries (Coletta et al., 2020) and are amenable to direct cross-species translation (Bertero et al., 2018)."

#. It is not clear if the authors used GSR (Global signal regression). This is currently advised to run the analyses with or without to have more robust results.

We thank the reviewer for giving us the possibility to clarify this important point. We are fully aware of the imperative need to rigorously control for spurious contributions and motion-related artefacts in human connectivity studies. For this reason, we regressed out CSF signal, 12 head motion parameters and employed wavelet despiking, i.e. a strategy that has been shown to robustly mitigate artefactual fMRI signal spikes and spurious correlations associated with head motion (Patel et al., 2014). Moreover, we also showed that all of our findings hold also when we use strict censoring of fMRI volumes contaminated by head motion (FD > 0.2mm, Figure S2, original manuscript). As suggested by the reviewer, GSR has also been previously used as an aggressive mitigation strategy for motion-related artefacts and physiological noise (Ciric et al., 2017). However, GSR does artefactually alter rsfMRI signal distribution in a way that would have strongly biased the cross-species approach and the result of our analyses we employed here. For this reason, we did not add GSR to our analyses, and we argue that this procedure should not be used to control for motion-related artefacts in rodent and human rsfMRI imaging. We explain in greater detail below why GSR would have detrimental biased the results of our analyses and their interpretability:

1) A major goal of our study was to extrapolate the rsfMRI hyper-connectivity signature identified in *Tsc2*^{+/-} mice to corresponding clinical populations. To this aim, it is of paramount importance that processing steps across species are comparable. There is now wide consensus in the preclinical imaging community that in rodents, where fMRI timeseries are virtually devoid of motion, GSR substantially worsens the spatial specificity of mouse functional networks, resulting in greatly impaired long-range connectivity (Grandjean et al., 2020). Accordingly, prior work by us and others have shown that the mouse pipeline we employed provides exquisite correspondence between the topography of fMRI and axonal connectivity networks (Grandjean et al., 2017; Coletta et al., 2020), ensuring close correspondence between

electrophysiological signals and interareal rsfMRI connectivity (Canella et al., 2020). Based on these considerations, we did not use GSR in rodent preprocessing, and by extension, in human rsfMRI scans, such as to maximize alignment and cross-species comparability of our findings.

2) GSR mathematically centers the distribution of inter-areal functional connectivity around zero, hence introducing artificial anti-correlations in connectivity measurements (Murphy et al., 2009). This aspect is particularly critical in studies where functional connectivity is compared across groups and conditions, as GSR-dependent anti-correlation makes the ensuing connectivity differences physiologically uninterpretable (Saad et al., 2012). Of equal importance, the use of whole brain aggregative measures such as “global connectivity” in GSR fMRI timeseries would result in zero-shifted mean functional connectivity (Supekar et al., 2013b), hence preventing the possibility of identifying physiologically meaningful between-group differences in insular regions.

3) A third important argument against the use of GSR regression is the emerging view that the GS, when devoid of artefactual motion-contamination, reflects important neural contributions. This has been recently demonstrated in a number of studies showing that fMRI global signal is tightly coupled to calcium signaling in mice (Matsui et al., 2016), local field potentials in monkeys (Schölvinck et al., 2010) and EEG activity in humans (Wen and Liu, 2016). Further primate research has shown that fMRI GS is related to spontaneous fluctuations in arousal-related systems (Chang et al., 2016; Turchi et al., 2018). Importantly, our own work further revealed that global signal fluctuations serve as a “master-clock” for the emergence of dynamic rsfMRI state fluctuations (Gutierrez-Barragan et al., 2019). We report below for the reviewer’s perusal a recent extension of this relationship for the human brain, as part of an ongoing study we are carrying out in control and autistic patients (Reviewer Figure R1). The reported results clearly show that, like in our mouse investigation (Gutierrez-Barragan et al., 2019), recurring human rsfMRI brain states (assessed with co-activation patterns - CAPs) are tightly phased-locked with global signal fluctuations. These results are consistent with a neural origin for GS and highlight a role of global fluctuations in timing spontaneous rsfMRI dynamics.

Based on these considerations, we did not use fMRI GSR in our work.

Reviewer Figure R1. Global signal as a master clock for the reconfiguration of brain states in humans. A) Spatial topographies of brain states in humans as measured by mapping co-activation patterns (CAPs) as in (Gutierrez-Barragan et al., 2019). B) Radar plots depicting the temporal occurrence of each CAP with respect to corresponding global signal fluctuation phase (that occurs at 0π). Temporal occurrence of all CAPs is phased locked to global signal, pointing to a key role of global signal in governing brain dynamics. (Pagani et al, in preparation).

In the revised version we have edited the methods by specifying that no GS regression was applied to our preprocessing:

Methods: "...Denoising steps included: wavelet time series despiking ('wavelet denoising'), confound signal regression including the 6 motion parameters, their first order temporal derivatives and ventricular cerebrospinal fluid (CSF) signal (referred to as 13-parameter regression). To avoid the generation of artificial anti-correlation in connectivity measurements, the fMRI global signal was not regressed. The wavelet denoising method has been shown to mitigate substantial spatial and temporal heterogeneity in motion-related artefacts....

The authors suggest that "the lack of major structural anatomical alterations predictive of the observed rsfMRI hyperconnectivity points at a possible mechanistic link between mTOR-related spine surplus and increased rsfMRI coupling in Tsc2+/- mutants". I would like to know if this is possible to observe tubers in the brain of the Tsc2^{+/-} mutant mice? It is unlikely that they that could alter specific networks, but since these tubers are observed in patients, it would be important to have the information and eventually to rule out this possibility.

The mouse line we employed (Onda et al., 1999) is the same line previously used by Tang et al. (2014), and it is characterized by complete lack of cerebral tubers (Sato et al., 2012). To confirm this finding, we have now used post-mortem morphoanatomical imaging (Pagani et al., 2016b) to obtain a high resolution (70 x 70 x 70 μm) T2-weighted mapping of the brain of Tsc2 mice (Supplementary Figure S4). An inspection of the obtained images confirmed the absence of cerebral tuberi in Tsc2 mutants. This finding rules out a possible confounding contribution from these anatomical formations to our imaging results.

Supplementary Figure S4: High-resolution morpho-anatomical imaging confirmed the absence of tubers in tsc2^{+/-} mice. The figure depicts a set of coronal cuts from a representative Tsc2^{+/-} mouse brain.

In the revised text, we have added Supplementary Figure S4 and described the lack of tubers in Tsc2 mutants:

Results: "...Furthermore, in keeping with prior investigations (Sato et al., 2012), high resolution anatomical imaging did not reveal any major morphological changes in the brain of Tsc2 mutants (Supplementary Figure S4). To probe axonal mesoscale structure in Tsc2+/- mutants..."

Methods: *High-resolution structural MRI.*

To rule out the presence of major morpho-anatomical changes in *Tsc2*^{+/-} mice we carried out ex-vivo structural MRI on PFA-fixed specimens (Pagani et al., 2016a) on a cohort of n = 5 *Tsc2*^{+/-} male and n = 5 control littermates. Brains were imaged inside intact skulls to avoid post-extraction deformations. Mice were deeply anesthetized with 5% isoflurane, and their brains were perfused *in situ* via cardiac perfusion. PFA intracardiac perfusion was performed with PBS followed by 4% PFA (100 ml, Sigma-Aldrich). Both perfusion solutions were doped with a gadolinium chelate (ProHance, Bracco) at a concentration of 10 and 5 mM, respectively, to shorten longitudinal relaxation times. High-resolution morpho-anatomical T2-weighted MR imaging of mouse brains was performed using a 72 mm birdcage transmit coil and a custom-built saddle-shaped solenoid coil for signal reception. High-resolution morpho-anatomical images were acquired with the following imaging parameters: FLASH 3D sequence with TR = 17 ms, TE = 10 ms, $\alpha = 30^\circ$, matrix size of 260 × 180 × 180, FOV of 1.83 × 1.26 × 1.26 cm, and voxel size of 70 μm (isotropic). Structural MRI confirmed the absence of cerebral tubers in our *Tsc2* mouse line, confirming previous reports (Sato et al., 2012).

For the Neurosynth and NeuroVault, as indicated by the authors, the dataset is based on only 6 brains from the Allen Brain. This is a concern, but this is to date the only data available. Since the authors used STRING to identify the interactors of TSC2, it seems that they selected all kind of interactors, not only binding partners, but also co-expressed genes. This could be circular to choose a set of co-expressed genes and to find that they are co-expressed in the brain regions of interest. It would be informative to test different confidence levels and types of interactions. A supplementary figure on this aspect would be nice. I really appreciated the gene enrichment analyses using CHD8 as a control for the specificity of the TSC2 network.

We thank the reviewer for giving us the opportunity to clarify this point. Our analysis does not probe whether mTOR-Tsc2 network genes are co-expressed in brain regions of interest, but it tests instead whether the list of genes exhibiting an expression pattern spatially correlated to our rsfMRI maps (i.e. a list of n = 2,412 genes out of the n = 20,787 gene probed; FDR-corrected at, $q < 0.05$) is significantly enriched for genes belonging to the mTOR-tsc2 network and dysregulated in ASD (n=88 genes, STRING analysis). As the two lists of genes we compared are independently determined, our analysis does not suffer from statistical circularity. The non-circularity of our analysis is also empirically supported by our control analysis where we show that the ChD8 interactome, which we also built by using co-expressed interactors, did not reveal any statistically significant enrichment. This would be not the case if our approach was intrinsically circular.

We have not restricted the type of interactors in STRING because there was no *a priori* reason for why a specific type of interactor would be relatively more important to focus on than others. Moreover, the use of default settings of medium confidence was instrumental to our goal of generating a broad mTOR-Tsc2 interactome network. This strategy also avoids the huge space of results that would occur if these parameters were systematically changed. This initial broad network was then pruned to be more ASD-specific through identification of interactors that are also members of dysregulated co-expression modules in ASD cortex from the study by Parikshak et al., (2016). This pruning step is very important since the broad set of interactors are then constrained to only those with evidence of dysregulation in ASD post-mortem cortical tissue, reducing the corresponding gene lists from n = 585 to n = 88. Given that the pruning step helps make the lists more ASD-specific, we are not of the opinion that it would be beneficial to make the initial list of interactors extremely specific by being conservative with confidence levels, or by introducing restrictions to specific kinds of interactor types.

To illustrate the problems that would occur if we changed the parameters to highest level of confidence and only physical interactors, we did this for mTOR as the seed gene. This PPI results in only 27 interactor genes and of those, only 5 overlap with Parikshak et al.,'s dysregulated co-expression modules. This $n = 5$ gene list is not large enough to represent the multifactorial genetics of autism and it is too limited for a highly powered enrichment test against imaging-relevant genes from the gene expression decoding analysis. Additionally, the addition of highly conservative parameters may possibly introduce biases relevant to the specific experimental evidence the PPI network currently uses. For example, if the physical interactor evidence is restricted to specific models organisms, tissues or milieus (e.g., yeast, cancer), then such a restrictive set of parameters will not necessarily yield brain-relevance for other contexts. Because of this, it is useful to leave the initial parameters at relaxed levels and then to increase the specificity through the added pruning by ASD-dysregulated co-expression module genes.

In the revision text, we have added more statements to clarify what our initial ideas were regarding leaving the STRING parameters at their default settings and then using the Parikshak et al., gene lists to prune the interactome network to ASD-relevant genes.

Methods: “To link rsfMRI functional hyper-connectivity in human patients with molecular mechanisms of relevance to mTOR and TSC2, we carried out gene decoding and gene enrichment analysis. The goal of this analysis is to test whether the list of genes exhibiting an expression pattern spatially correlated to our rsfMRI map is significantly enriched for genes belonging to the mTOR-Tsc2 interactome and dysregulated in ASD. As the two lists of genes are independently determined, this analysis does not suffer from statistical circularity. To carry out gene decoding analysis we used Neurosynth and NeuroVault (Gorgolewski et al., 2014) to identify genes whose spatial expression patterns are consistently similar across subjects to our maps of connectivity differences. This decoding analysis utilizes spatial gene expression data from the six donor brains from the Allen Institute Human Brain Gene Expression atlas (Hawrylycz et al., 2012; Hawrylycz et al., 2015). This database contains gene expression maps for $n = 20,787$ genes. The analysis first utilizes a linear model to compute similarity between a user-input unthresholded whole-brain imaging map and spatial patterns of gene expression for each of the six brains in the Allen Institute dataset. The slopes of these donor-specific linear models encode how similar each gene's spatial expression pattern is with user-input imaging map. Donor-specific slopes were then subjected to a one-sample t-test to identify genes whose spatial expression patterns are consistently of high similarity across the donor brains to the user-input imaging map. The resulting list of genes is then thresholded for multiple comparisons and only the genes positive t-statistic values surviving FDR $q < 0.05$ are considered. In our application of these gene expression decoding analyses, the user-input imaging maps used were whole-brain unthresholded t-statistic maps of group-differences in insular seed connectivity for ASD vs TD or ASD subgroup vs TD comparisons. The result of this first analysis is a list of $n = 2,412$ genes (FDR corrected, $q < 0.05$) whose expression topography is statistically similar to our input fMRI hyper-connectivity map.

With a subset of fMRI-relevant genes isolated, we then asked whether this list significantly overlaps with genes that interact at the protein level with mTOR or TSC2. To identify a broad network of mTOR-TSC2 relevant genes, we carried out protein-protein interaction (PPI) analyses in STRING-DB (Szklarczyk et al., 2019) (<https://string-db.org>). Here we used TSC2 or mTOR as seed genes, and queried for up to 500 possible interactors at the default confidence level of 'medium' (0.4). Regarding the settings for STRING-DB, we have not restricted the type of interactors because there was no *a priori* reason to prioritize on type of interactor over another. Also, the use of default settings of medium confidence was instrumental to our goal of generating a broad mTOR-Tsc2 interactome network. This initial broad network was then pruned to be more ASD-specific through the identification of interactors that are also members of dysregulated co-expression modules in ASD cortex from the study by Parikshak et al., (2016). This pruning step was crucial to narrow the broad set of interactors to only those with evidence of dysregulation in

ASD post-mortem cortical tissue, thus reducing the initial list of $n = 585$ interactors to a final set of $n = 88$ genes. The resulting $n = 88$ gene lists is thus composed of genes that interact with mTOR or Tsc2 at the protein level, and are dysregulated in ASD. Finally, to establish a possible link between mTOR-Tsc2 dysregulation and fMRI hyper-connectivity maps we carried out gene enrichment analysis. This step tests whether the list of $n = 2,142$ genes that exhibit spatial correlation with our maps is significantly enriched for mTOR-Tsc2 ASD-relevant interactors as per our $n = 88$ list. The enrichment analysis was implemented with custom code that computes enrichment by using odds ratios and hypergeometric p-values <https://github.com/mvlombardo/utis/blob/master/genelistOverlap.R>

Difference to anesthesia sensitivity between mutant and wild-type mice has been tested, but since functional connectivity is very sensitive to anesthesia, the authors might discuss the fact that fMRI humans studies are performed in majority without anesthesia in contrast to their protocol.

Rodent fMRI connectivity mapping is typically carried out under light sedation to ensure immobility and avoid restraint-induced stress. This was obviously not the case of human ASD scans, which were acquired in awake subjects (Di Martino et al., 2014). Notwithstanding these differences, we were able to reliably translate a signature of fronto-striatal-insular hyperconnectivity across species. This result is not surprising, as a large body of evidence indicates that light anesthesia in rodents preserves brain network organization (Gozzi and Schwarz, 2016; Coletta et al., 2020), arguing against a major contribution of brain state differences to our findings.

Great care was also taken in ruling out genotype-dependent changes in anesthesia sensitivity. In addition to our previous measurements of minimal alveolar concentration, we have now included a new analysis showing that control and Tsc2 mutants exhibit a comparable hemodynamic response function in the PFC, a brain region showing prominent hyper-connectivity in Tsc2^{+/-} mice (BOLD peak amplitude, $p = 0.75$; time to peak, $p = 0.52$; FWHM, $p = 0.75$). These results further corroborate a neural (rather than neurovascular) origin of the rsfMRI changes in Tsc2^{+/-} mice (Supplementary Figure S9).

Supplementary Figure 9. Hemodynamic response function exhibit canonical features in Tsc2^{+/-} mice. A) Hemodynamic response in PFC in Tsc2^{+/-} mutants and +/+ control mice. B) BOLD peak amplitude, time to peak and FWHM of the hemodynamic response in Tsc2^{+/-} mice are comparable to the corresponding parameters observed in control mice.

In the revised manuscript we have added these measurements as follows:

Methods: "... This observation argues against a significant confounding contribution of anesthesia to our functional measures. To further rule out a neurovascular origin of the mapped functional changes, we measured the hemodynamic response in the PFC, a brain region showing prominent hyper-connectivity in Tsc2^{+/-} mice. Intergroup genotype-dependent comparisons revealed unimpaired hemodynamic

response in *Tsc2*^{+/-} mice (BOLD peak amplitude, $p = 0.75$; time to peak, $p = 0.52$; FWHM, $p = 0.75$, Supplementary Figure S9). These results corroborate a neural (rather than neurovascular) origin of the rsfMRI changes in *Tsc2*^{+/-} mice.”

In the revised manuscript we have also introduced the following sentence to highlight the different brain state of our cross-species connectivity mapping.

Discussion: “Our rsfMRI results in mice are broadly consistent with EEG data in children harboring *Tsc2* mutations, in which broadly increased inter-regional synchronization has been recently reported (Davis et al., 2019). These parallel findings are important, given that our cross-species measurements were carried out under different brain states, e.g. in lightly anesthetized rodents and awake ASD patients.”

I appreciate that the authors usually indicate that their results might be true only for a subset of patients. I would therefore suggest to indicate this in the abstract “a subset of” before “children with autism”. By the way, several MRI studies supporting the hypothesis were performed in adults so “individuals” might be more correct than “children”.

Thanks for the comment. Throughout our abstract and manuscript we referred to children with autism to account for the fact that our analyses were restricted to individuals from ABIDE-I between 6 and 13 years old. In our abstract we referred to children with autism when describing the results of our initial case-control group-level analyses (e.g. Figure 4), which revealed hyperconnectivity and mTOR-relevant transcriptional changes across the entire ABIDE-cohort, and to our extension of these investigations to ASD subtypes (e.g. Figure 5), where we specifically mentioned our results to be driven by a “subset of children” with ASD.

We agree with the reviewer that a consistent number of studies of functional dysconnectivity have been carried out in adult autistic populations, for a review see (Hull et al., 2017). However, evidence of functional connectivity alterations is also well documented and replicated across pediatric cohorts (Di Martino et al., 2011; Abrams et al., 2013; Supekar et al., 2013a; Uddin et al., 2013; You et al., 2013; Spera et al., 2019). To make this point clearer we added references to studies reporting aberrant rsfMRI connectivity in pediatric populations in the introduction as follows:

“At the macroscopic level, several studies have highlighted the presence of aberrant functional connectivity in ASD as measured with resting state fMRI (rsfMRI) (Di Martino et al., 2011; Abrams et al., 2013; Supekar et al., 2013a; Uddin et al., 2013; You et al., 2013; Spera et al., 2019)”

Reviewer #3 (Remarks to the Author):

In their study, Pagani et al conduct a series of analyses across micro- and macro-scales in the Tsc2 mouse model of Tuberous sclerosis and idiopathic ASD in humans with the goal of establishing a causal mechanistic link between mTOR-synaptic alterations and functional connectivity deficits in ASD. The clear strength of this study is the multidisciplinary tools across mouse and human to assess phenotypes from the level of the synapse to whole brain connectivity. The application of these multilevel tools represents a level of novelty, revealing new ways of looking at Tsc^{+/-} phenotypes with putative links to humans. The manuscript is also very well written. The connectivity findings using rsfMRI and associated rescue of this phenotype in the Tsc^{+/-} mouse model is the most novel finding, and also most convincing. This finding in particular will be of high value to the field.

We thank the reviewer for the appreciation for our work.

Tempering these strengths are notable weaknesses. The first is the above described novelty is somewhat overshadowed by the fact the the mTOR pathway and many associated readouts presented, including the pharmacological rescue experiments, have been well explored in previous studies. Beyond this, and most importantly, in a number of experimental instances (described below in more detail), the data is only cursorily presented, reducing confidence that the experiments have the needed rigor to reach the conclusions presented. This is especially critical since the authors strongly claim a mechanistic link between micro-scale synaptic deficits and macro-scale connectivity deficits, a link for which other mechanisms have not been fully ruled out.

We thank the reviewer for the constructive appraisal of our work. In response to the reviewer's comments, we have now added new experimental evidence ruling out major micro- and mesoscale anatomical changes in Tsc2 mice, further corroborating a possible mechanistic link between synaptic pathology and macro-scale hyper-connectivity as per our original claims. We describe these new experimental results in greater detail below.

Regarding the originality of our findings, the major strength of our work lies in the identification of a previously-unidentified link between mTOR-related synaptic pathology and large-scale network aberrations, and the translation of this connectivity and transcriptomic signatures to human autism. We believe this is a novel and important contribution to the field, as it reconciles two major theories of autism across scales of inquiry. In this respect, it should be emphasized that all the Tsc2 studies we carried out have been solely and specifically designed to mechanistically support our hypothesis, and when based on prior work, we have openly and extensively acknowledged the original references (e.g. Tang et al., 2014) throughout the text.

The addition of human subjects is also a strength of the study, but considering the tremendous heterogeneity in connectivity findings from a number of studies, there are inherent limitations of how to extrapolate the findings to: 1) linking directly to experiments presented in Tsc^{+/-} mice and 2) how widely they are applicable across idiopathic ASD. In this instance, if possible, it would have been more applicable to study humans with TSC.

While a direct comparison of mouse connectivity findings with analogous rsfMRI measurements in TSC2 would be of interest, to the best of our knowledge no such a dataset is freely available to the imaging community. We nonetheless think our work is important as it permits to link a commonly observed neural trait associated with autism (i.e. increased dendritic spine density) to a specific connectivity signature in human ASD. In this respect, the relatively isolated synaptic pathology in Tsc2 mice, together with the

possibility of rescuing it with rapamycin, allowed us to mechanistically link these two pathological mechanisms with a high degree of confidence.

We also would like to point out that it was not our intention to suggest that the observed hyper-connectivity fingerprint, and its corresponding transcriptional signature, are representative of idiopathic autism as a whole. This statement is supported by our analyses depicted in Figure 5, which were motivated by the assumption that idiopathic autism cannot be exclusively or prominently ascribed to mTOR-related pathology. This was explicitly acknowledged in our original statement introducing the analyses of Figure 5: *“While previous work has implicated aberrant mTOR signaling in idiopathic ASD (Tang et al., 2014; Gazestani et al., 2019), the daunting etiological heterogeneity underlying these disorders implies that only a fraction of ASD individuals within a multisite dataset like ABIDE would be expected to be affected by mTOR-related dysfunction.”*

We are indeed fully aware of the extraordinary heterogeneity of the disorder, and we recently addressed this issues by showing that connectivity changes across 16 mouse models of autism exhibit distinct, and at times diverging, connectivity signatures (Zerbi et al., 2020). In the same work, we also showed that this connectional heterogeneity can however be classified into segregable connectivity neuro-subtypes. Extending this approach to human autism, in the present work we show that idiopathic ASD can similarly be deconstructed into distinct subtypes characterized by specific patterns of atypical fronto-striato-insular functional connectivity (Figure 5). Importantly, only one of the identified subtypes exhibited a transcriptional signature that we could relate to the mTOR pathway via transcriptomic analysis, in full agreement with the reviewer’s point that mTOR pathology is only one of the many etiological contributors of autism.

In order to comply with editorial requirements about manuscript length, we have not been able to introduce a discussion of a number of relevant comments, including this specific point.

More specifics regarding concerns are:

1) While in general the experiments appear to be well executed, the link between spine abnormalities and more large-scale connectivity and behavioral deficits needs more experimental evidence. For example, other deficits that additionally or more likely may be causal include axonal placement/survival (including interneurons), axonal integrity or myelination, all of which have been observed in *Tsc^{+/-}* mice and cannot be ruled out. While the authors begin to address the status of myelin and axonal pathways in Figure 2, the data presented are insufficient to support their claims of lack of myelin or axonal deficits. Myelin deficits can be directly assessed by myelin stains and retrograde axonal tracing data should be shown in a separate figure with consisting of visualization of axon pathways, and appropriate quantification and controls for size of infection/number of starter cells infected

We thank the reviewer for this comment. We agree that a further substantiation of our findings was warranted. To provide additional evidence supporting a possible link between increased spine density and rsfMRI hyper-connectivity, we have now expanded our anatomical investigations as detailed below.

First, we have carried out new measurements of dendritic spine density in the retrosplenial cortex, a cortical region of the mouse DMN exhibiting robust rsfMRI hyperconnectivity in *Tsc2* mice (Supplementary Figure S1). In line with our prior findings in the insular cortex, this new experiment revealed robustly increased spine density in *Tsc2* mice, and complete rescue of this phenotype by developmental treatment with rapamycin. Because the retrosplenial cortex in *Tsc2* mice shows robust hyperconnectivity that is

rescued by rapamycin, this novel set of results further corroborates a possible link between mTOR-dependent synaptic surplus and functional hyperconnectivity.

Retrosplenial cortex

Supplementary Figure S1. Increased dendritic spine density in the retrosplenial cortex of *Tsc2*^{+/-} mice is rescued by rapamycin. In keeping with our prior measurements, we found increased spine density in the retrosplenial cortex of *Tsc2*^{+/-} mice (orange circles, $q < 0.001$, FDR-corrected). This alteration was completely rescued by pharmacological treatment with rapamycin (blue circles, $q < 0.001$, FDR-corrected).

We have now added this new result in Supplementary Figure S1 and included these additional statements in Methods and Results sections.

Methods: "... a core component of the mouse salience network (Gozzi and Schwarz, 2016), using Imaris software. The same measurements were replicated also in the retrosplenial cortex (Wild-type mice, N=5; *Tsc2*^{+/-} mice, N=4; wild-type mice treated with rapamycin, N=5; *Tsc2*^{+/-} mice treated with rapamycin, N=5), a distal region showing similar rsfMRI hyperconnectivity in *Tsc2*^{+/-} mice. The analysis was performed by an operator blind to the genotype and treatment..."

Results: "...density compared to control littermates ($p = 0.021$, Figure 1A and $p < 0.001$, Supplementary Figure S1). We next used rsfMRI to map brain..."

Results: "...comparable to *Tsc2*^{+/+} control mice (Figure 3A and Supplementary Figure S1). Remarkably, rsfMRI mapping..."

Second, to probe a possible contribution of myelin deficits to the observed rsfMRI hyperconnectivity, we also quantified myelin basic protein (MBP) in the cingulate cortex, corpus callosum and hippocampus of *Tsc2* mutants. Our measurements revealed largely unimpaired MBP in *Tsc2*^{+/-} mice in all the examined brain regions. Similarly, treatment with rapamycin did not affect MBP levels in control or mutant mice (Supplementary Figure S2). The lack of MBP density alterations in *Tsc2* mice replicates the results of similar investigations by Hsieh et al. (Hsieh et al., 2021), arguing against a contribution of microscale white matter integrity alterations to the hyperconnectivity we observed in this mouse line.

Supplementary Figure S2. Myelin basic protein staining. A) Representative sections showing myelin basic protein (MBP) staining in different brain regions. B) Myelin basic protein densitometry in the corpus callosum (CC, top panel), cingulate cortex (Cg, middle panel) and hippocampus (HPC, bottom panel) in control and *Tsc2*^{+/-} mice treated with vehicle or rapamycin (rapa). C) MBP coherency in CC, Cg, and HPC in control or *Tsc2*^{+/-} mice treated with vehicle or rapamycin. ROD: relative optical density value.

Because the same authors described the presence of putatively altered coherency of myelinated axons in the hippocampus of *Tsc2* mutants (Hsieh et al., 2021), we also carried out this analysis on our MBP-stained samples. Our measurements covered inter-callosal white matter bundles (corpus callosum) as well as two gray matter regions, i.e. the cingulate cortex and dorsal hippocampus. We report the results of these new assessments in Supplementary Figure S2. In contrast to what reported by Hsieh et al., 2021, our analyses did not observe any genotype-dependent MBP coherence difference in any of three examined regions. These new findings rule out gross myelin-related alterations in *Tsc2* mutants, and are consistent with a synaptic origin for the connectivity changes we mapped.

Finally, in response to the reviewer's comment about our mesoscale rabies tracing, we have complemented our original circular plot visualizations of cell projection density to PFC with an expanded presentation of retrograde axonal tracing data as requested (Supplementary Figure S3). Specifically, we report now corresponding regional quantifications of normalized infected cell counts as requested, and show the distribution of pre-synaptically traced projecting neurons in two representative brain slices, denoting a broadly similar distribution of cell bodies (and therefore comparable long-range axonal projection pathways) across genotypes. We also report two representative images depicting area of viral injection of the anterior cingulate in the two genotypes. These analyses substantiate our prior claims of a lack of gross mesoscale axonal rewiring in *Tsc2* mutants.

Supplementary Figure S3. Recombinant rabies virus tracing of cell projections to PFC. A) Regional quantifications revealed unimpaired cell projection density in $Tsc2^{+/-}$ mice vs. control littermates ($q > 0.76$, FDR-corrected, all regions). To avoid viral batch effects and operator-dependent biases our measurements have been normalized for total number of marked cells as in (Bertero et al., 2018; Pagani et al., 2019). B) Distribution of cells projecting to PFC as mapped with viral tracing in representative coronal slices (top panel, Bregma = -1.255, and bottom panel, Bregma = -1.855) in $Tsc2^{+/-}$ mice (green dots) and control mice (red dots). C) Extension of site of infection in representative control (top panel) and $Tsc2^{+/-}$ (bottom panel) mice.

We have now added these novel analyses and quantifications to the manuscript as Supplementary Figure S2 and Supplementary Figure S3, and expanded the corresponding description of these results as follows:

Results: “The lack of regional FA differences also argues against the presence of major alterations in whole-brain white matter topography, as these would be appreciable in the form of large regional FA differences (Dodero et al., 2013). To rule out the presence of more subtle myelination deficits in $Tsc2^{+/-}$ mice, we carried out measurements of myelin basic protein (MBP) density and coherency (Hsieh et al., 2021), using immunostaining methods. In keeping with our FA measurements, MBP staining did not show any MBO-related alterations in the corpus callosum and in gray matter regions (PFC, hippocampus) of $Tsc2^{+/-}$ mutants (Supplementary Figure S2). These results corroborate the lack of gross macro- and micro-structural WM reorganization in these mice. To probe axonal mesoscale structure in $Tsc2^{+/-}$ mutants, we next carried out retrograde axonal tracing using recombinant rabies virus (Liska et al., 2018) in the medial prefrontal cortex, a region exhibiting prominent rsfMRI hyperconnectivity in $Tsc2^{+/-}$ mutants. Quantification of retrogradely-labelled cells in neocortical, striatal and sub-thalamic areas revealed largely comparable projection frequency in all the examined regions across genotypes ($q > 0.77$, FDR-corrected, Figure 2B and Supplementary Figure S3).

2) The rapamycin rescue of connectivity deficits as assessed by rsfMRI in Figure 3 is robust and impressive. While the authors show spine abnormality rescue by mTOR inhibition, the connectivity deficits can also be explained by putative rescue of above described phenotypes, which are unexplored

As described above in greater detail, MBP and rabies-infected projection densities were broadly unaltered in $Tsc2^{+/-}$ mice, ruling out the possibility that they could have a significant role in accounting for the rsfMRI hyper-connectivity mapped in our $Tsc2^{+/-}$ mice.

3) The effort to link regional gene alterations in ASD as shown in Figures 4 & 5 is a worthwhile effort, though the input ASD gene list is of critical importance. Here it appears that the Parikshak et al, 2016 gene

list was used. The rationale for the choice of this gene set is not clear. It is suggested that the authors instead use the Satterstrom et al 2020 gene set which represents the largest cohort to date, and perhaps also the SFARI gene set (though this is not without its caveats).

We thank the reviewer for bringing up this point, as it allows us to better articulate what our goals were by using the Parikshak et al., 2016 gene list. Our gene decoding and enrichment analysis was aimed to probe whether we could identify an mTOR-tsc2 transcriptomic signature in the cortico-striatal rsfMRI hyperconnectivity mapped in our clinical population of ASD individuals. To this aim, we leveraged gene expression maps of the Allen's Brain Institute to test whether our case-control fMRI contrast maps could be topographically linked to spatial expression maps of genes that are mTOR-Tsc2 interactors *and* differentially expressed in the brain of autistic individuals. As discussed more extensively above in response to the comments of reviewer #2, this list was then pruned to restrict mTOR-Tsc2 interactors to those that have been shown to be transcriptionally-dysregulated in ASD. Because our analyses are based on interrogation of gene expression levels across the brain, the gene list published by Parikshak (Parikshak et al., 2016) is the most suited to our goal, as it contains genes that have been shown to be differentially expressed in brain tissue from ASD individuals, i.e. it contains gene expression values that are directly relatable to our measurements of interest. Because our goal was to assess the overlap between imaging-relevant genes, whose spatial expression pattern is similar to the imaging maps, it is most-relevant to assess this enrichment with genes defined based on relevance at the level of the transcriptome. By contrast, Satterstrom's gene list and the SFARI ASD gene list are lists of genes that provide evidence of association with autism at the level of DNA mutations (not transcriptomic dysregulation) and as such they do not assess the level of interest (e.g., the transcriptome), given our use of spatial gene expression decoding with the Allen Institute Human Brain Atlas dataset.

In the revised manuscript, we have now better articulated this point regarding how the goal of the analysis was to match up imaging-relevant genes to genes of relevance to transcriptomic dysregulation in ASD.

Methods: "...This initial broad network was then pruned to be more ASD-specific through the identification of interactors that are also members of dysregulated co-expression modules in ASD cortex from the study by Parikshak et al., (2016). In particular we filtered for TSC2 or mTOR interactors that are also members of differentially expressed genes in autism. As our analyses are based on interrogation of gene expression levels across the brain, we used gene co-expression modules from (Parikshak et al., 2016) as this list contains genes that have been shown to be differentially expressed in brain tissue from ASD individuals, i.e. it contains gene expression values that are directly relatable to our rsfMRI measurements of interest. This pruning step was crucial to narrow the broad set of interactors to only those with evidence of dysregulation in ..."

Reviewer #4 (Remarks to the Author):

General comments:

Postmortem studies from several independent groups have collectively revealed increased density of excitatory synapses in the brains of individuals with autism. Neuroimaging studies reported aberrant hyperconnectivity in large scale networks. This study investigates potential causal link between excess of synapses and aberrant functional connectivity in a mice model and in human subjects. Using rsfMRI, electrophysiology and in silico modelling in Tsc2 haploinsufficient mice, the authors claim that mTOR-dependent increased spine density is associated with autism-like stereotypies and cortico-striatal hyperconnectivity; and that these deficits are rescued by pharmacological inhibition of mTOR. The authors performed similar neuroimaging analysis in children with idiopathic autism, and found analogous cortical-striatal hyperconnectivity. They further showed that transcriptomic signature was predominantly expressed in a subset of children with autism, thereby defining a segregable autism subtype.

This is potentially an important paper. Being able to establish a causal link between non-invasive macroscopic measure (aberrant functional MRI network) in ASD patients and cellular mechanism and transcriptomic signature, and making cross-species link could have profound implications in early diagnosis and treatment prognosis. However, some of the data presented in this manuscript do not appear to be convincing. I am afraid substantial work is needed to make a more convincing case.

We thank the reviewers for the words of appreciation and for the constructive suggestions. In the revised manuscript we have edited the methods and result sections to more clearly describe the rsfMRI connectivity analysis we carried out. We have also strongly corroborated the link between regional synaptic deficits and altered rsfMRI connectivity deficits by counting spine density in a separate brain region as suggested by the reviewer. We have further corrected all our statistical comparisons for post-hoc multiple comparisons, and re-calculated global connectivity and seed based mapping in our human sample by using white matter regression and CompCorr, two stringent denoising methods for rsfMRI time-series. Finally, we carried out additional experimental measurements and in-silico manipulations to validate our whole-brain modelling.

1. Technically, there is a lack of clarity in fMRI data analysis, in both human and animal fMRI data. There appears to be some confusion in terminology as well. On page 22, it was stated “..... we calculated voxel-wise long-range connectivity maps for all mice. Long-range connectivity is a graph-based metric also known as unthresholded weighted degree centrality and defines connectivity as the mean temporal correlation between a given voxel and all other voxels within the brain (Cole et al., 2010).” The connectivity reported in Cole’s paper was “global connectivity”, which is basically the mean temporal correlation between a given voxel and all other voxels within the brain. Additionally, although conceptually related, it is different from degree of centrality as defined in graph theory.

We agree with the reviewer that our prior description of fMRI data analyses was not clear enough, and we apologize for the confusion. We have now better explained our terminology and made it consistent throughout the revised manuscript.

To obtain spatially-unbiased maps of intergroup differences in functional connectivity (Figure 1B and Figure 3B), we used (weighted) global connectivity as implemented in Cole et al 2010. “Global connectivity” is a measure of the mean temporal correlation between a given voxel and all other voxels in the brain. To avoid the use of arbitrary rsfMRI correlation thresholds, our global connectivity measurements were carried out on functional connectivity maps without applying any arbitrary

correlation intensity threshold (hence our use of the adjective “*unthresholded*”). In graph theory, global connectivity is mathematically equivalent to (and as such also known as) weighted degree centrality (Rubinov and Sporns, 2011). It should also be noted that global connectivity predominantly reflects long-range functional connectivity, as by definition most of the voxels (i.e. 99.77%) are located more than 600 μm apart from the given voxel (i.e. a distance comparable to average cortical thickness in the mouse). For this reasons we previously equated global connectivity to long-range connectivity (Liska et al., 2015; Bertero et al., 2018; Pagani et al., 2019). This notion is corroborated by a measurement of local connectivity that we have now carried out by restricting global connectivity mapping within a radius of 600 μm . As predicted, this measurement failed to recapitulate our global connectivity result, as no intergroup voxel-wise difference between wild-type and Tsc2 mutants with or without treatment with rapamycin or vehicle was found ($|t| > 2$, $p > 0.05$, FWER $p > 0.01$, all comparisons). This results supports the notion that global connectivity is predominantly a measurement of long-range fMRI interactions. For these reasons, in our original manuscript we used the terms global connectivity, weighted degree centrality and long-range connectivity interchangeably.

In the revised manuscript we have now removed the terms weighted degree centrality (as well our reference to “*unthresholded*” GC) and retained only the terms global connectivity and long-range connectivity. We have also explained in greater detail the relationship and meaning of these measurements as follows:

Methods (mouse): “To obtain a data driven identification of the brain regions exhibiting genotype-dependent alterations in functional connectivity, we calculated voxel-wise weighted global connectivity maps for all mice as in (Cole et al., 2010) (here referred to as global connectivity). Global connectivity, known as weighted degree centrality in graph-based metrics, is a measure of mean temporal correlation between a given voxel and all other voxels within the brain (Cole et al., 2010). Global connectivity is strongly biased towards long range connections (defined here as $> 600 \mu\text{m}$ apart from a given voxel) as these comprise 99.77% voxels in the brain. In keeping with this notion, we found that restricting GC within a radius of 600 μm did not reveal any intergroup differences between wild-type and Tsc2 mutants treated with rapamycin or vehicle ($|t| > 2$, $p > 0.05$, FWER $p > 0.01$, all comparisons). For this reason we have here used the terms global connectivity and long-range connectivity interchangeably.”

Methods (human): “Analogous to the functional connectivity analysis we carried out for the mouse study (see above), to obtain an unbiased mapping of inter-group connectivity differences we calculated individual maps of global connectivity (hereafter interchangeably referred to as “long-range rsfMRI connectivity”). To exclude the contribution of white matter and CSF, we limited our measurements to voxels within a 25% grey-matter probability mask (Bertero et al., 2018). To test the across-site reproducibility of our findings, we first calculated voxel-wise intergroup comparisons global connectivity maps for each dataset (i.e. site) separately by using Cohen’s d (Sullivan and Feinn, 2012). We then produced a summary occurrence map, indicating the number of datasets presenting Cohen’s d > 0.2 . To identify the hotspots of long-range functional hyperconnectivity, we aggregated all the scans and we carried out a mega-analysis of global connectivity. Prior to this, we used a mixed model regression analysis to regress out the spurious contribution of inter-site variability and harmonize functional connectivity maps. The same model was also used to regress out the contribution of age, sex and IQ. Inter-group comparisons were then carried out by using unpaired 2-tailed Student’s t-test ($|t| > 2$, $p < 0.05$). We subsequently performed seed analyses, using as seed the bilateral foci showing altered global connectivity in individuals with ASD. Voxel-wise intergroup differences for seed based mapping were assessed using unpaired 2-tailed Student’s t-test ($|t| > 2$, $p < 0.05$) and family-wise error (FWER) cluster-corrected using a cluster threshold of $p < 0.01$ as implemented in FSL. Despite the use of wavelet despiking (Patel et al., 2014), inter-group comparisons revealed that mean framewise displacement (FD) was higher in ASD vs.

TD children (0.26 ± 0.19 and 0.19 ± 0.12 , respectively). To confirm that our results were not contaminated by in-scanner head motion we applied a stricter control for motion, by removing all time-points with FD higher than 0.2 mm, and we repeated **global connectivity** measurements and seed based correlation mapping on the obtained censored timeseries. Inter-group comparisons largely confirmed the results obtained with unscrubbed BOLD time-series of children with ASD, ruling out a possible spurious contribution of head motion in the mapped effects (Supplementary Figure S1).

Furthermore, it is unclear how seed-based analysis was conducted. It was stated “VOIs were placed in the prefrontal cortex to map derangements ...” It was later stated “Antero-posterior default mode network (DMN) and salience network (SN) connectivity alterations were probed by computing seed-to-VOI correlations.” As a result, the reviewer can only guess what information Figure 1B-D conveys.

Figure 1B depicts inter-group genotype-dependent difference maps of global connectivity. In subsequent seed-based correlation analyses, we mapped functional connectivity of the anterior cingulate (default mode network) and insular cortex (salience network) in *Tsc2* mutants and control mice. We also mapped the corresponding between-group voxelwise connectivity difference maps (Figure 1C-D). The anatomical location of the $3 \times 3 \times 1$ voxel seeds used for network probing is now reported in Supplementary Figure S10. Seed placement was based on prior anatomical characterization of these networks (Sforazzini et al., 2014; Whitesell et al., 2020). To quantify rsfMRI connectivity scores and provide a description of the distribution of connectivity across subjects, we next computed Pearson’s correlation between the seed and specific volumes of interest for both the DMN (Figure 1C) and SN (Figure 1D) in *Tsc2*^{+/−} and control mice (VOIs are depicted in green in Supplementary Figure S10). VOI locations were chosen based on the topography of between-group voxel-wise connectivity difference maps.

Supplementary Figure S10. Anatomical location of seed regions and volumes of interest. Seed based functional connectivity was quantified by computing correlation between the seed ($3 \times 3 \times 1$ voxels, in blue) and network-specific volumes of interests ($3 \times 3 \times 1$ voxels, in green).

In the revised manuscript, we have added Supplementary Figure S10 and we reworded the corresponding text as follows:

Methods: “... Antero-posterior default mode network (DMN) and salience network (SN) connectivity alterations were quantified by computing temporal Pearson’s correlation between seed regions and cubic volumes of interest ($3 \times 3 \times 1$ voxels) and plotted using scatterplots. Seeds and VOI locations are depicted in Supplementary Figure S10.”

Results: “Global connectivity analysis revealed prominent foci of increased connectivity in the PFC and insular cortex of $Tsc2^{+/-}$ mice (Figure 1B). The regions affected are core nodes of the mouse default-mode and salience networks, two evolutionarily-conserved systems (Gozzi and Schwarz, 2016) that have been widely implicated in ASD (Uddin et al., 2013; Gogolla et al., 2014). To identify the target regions associated with these connectivity changes, we next carried out network probing of PFC and in insular cortex by using seed based mapping. This analysis revealed that $Tsc2^{+/-}$ mice exhibit functional...”

Q3. Spatial specificity of synaptic alterations in $Tsc2^{+/-}$ mutants. Figure 1A shows spine density counting. But only in insular cortex. There is no mention whether alterations in spine density was region-specific or a global phenomenon. Considering the fMRI measures appear to implicate multiple brain networks, it would seem particularly relevant in order to determine whether “mTOR-dependent synaptic pathology is associated with a specific brain-wide signature of functional dysconnectivity”, as stated by the authors.

To probe the spatial extension of dendritic spine pathology in $Tsc2$ mutants, and further support a link between increased spine density and rsfMRI hyper-connectivity, we have now extended our dendritic spine investigations to the retrosplenial cortex, a brain region showing marked rsfMRI hyper-connectivity in $Tsc2$ mice. In line with measurements carried out in the insular cortex, we found increased spine density in the retrosplenial cortex of $Tsc2$ mice compared to control littermates. Notably, developmental treatment with rapamycin rescued retrosplenial dendritic spine surplus in $Tsc2$ mutants to levels found in wild-type untreated mice (Supplementary Figure S1). This novel results further support our hypothesis of a possible mechanistic link between mTOR-dependent synaptic surplus and functional hyperconnectivity in these mice. As previous studies reported increased dendritic spine density in auditory regions of $Tsc2$ mice (Tang et al.) we speculate that dendritic spine density pathology in $Tsc2$ might broadly affect large cortical territories. This notion would be consistent with the spatial over-extension of DMN connectivity observed in $Tsc2$ mutants (i.e. Figure 1C), in which this network extends to robustly incorporate components that are peripheral to this system, including portions of the auditory and somato-motor cortices.

Retrosplenial cortex

Supplementary Figure S1. Increased dendritic spine density in the retrosplenial cortex of $Tsc2^{+/-}$ mice is rescued by rapamycin. In keeping with our prior measurements, we found increased spine density in the retrosplenial cortex of $Tsc2^{+/-}$ mice (orange circles, $q < 0.001$, FDR-corrected). This alteration was completely rescued by pharmacological treatment with rapamycin (blue circles, $q < 0.001$, FDR-corrected).

In the revised manuscript we have included these new assessments as Supplementary Figure S1 (see reviewer #3, page 20 of this rebuttal) and have added a brief discussion of this point as follows:

Discussion: "... a subtype of cells that serve as key orchestrators of network-wide functional connectivity (Harris et al., 2018; Whitesell et al., 2020). The identification by us and others (Tang et al., 2014) of increased spine density in multiple cortical regions of Tsc2 mice suggests that mTOR-dependent spine pathology may affect large cortical territories. This observation is consistent with the spatial over-extension of DMN network connectivity observed in Tsc2 mutants to include peripheral motor-sensory districts of this systems (Figure 1C). Future investigations are warranted to assess whether this relationship can be generalized to other brain systems."

Q4. Figure 3 shows the effect of pharmacological mTOR inhibition on synaptic surplus, functional hyperconnectivity and autism-like behavior in Tsc2^{+/-} mice. There are 4 cohort of animals, 2 animal groups × 2 treatment (rapamycin vs vehicle). It appears that t-stats were applied to analyze the data. Would ANOVA followed by post-hoc analysis be more appropriate?

We thank the reviewer for this comment, which gives the opportunity to clarify the strategy employed for statistical testing. Given our specific hypotheses, we opted for planned contrasts and applied false discovery (FDR) rate correction (Benjamini-Hochberg) to correct for multiple comparisons. We felt this was a more direct approach than testing the interaction, only to later have to follow up with specific contrasts with directional predictions. Specifically, the planned contrast of interest were (a) the effect of the genetic alteration (i.e. Tsc2^{+/-} vs. control littermates), (b) its rescue by rapamycin (i.e. Tsc2^{+/-} mutant mice treated with rapamycin vs. vehicle-treated Tsc2^{+/-} mutants), and (c) and the degree of phenotypic normalization produced by rapamycin in Tsc2 mutants (i.e. vehicle-treated wild-type mice vs. rapamycin-treated Tsc2^{+/-} mutants). This is the specific core set of contrasts we used for mTOR-dependent spine surplus, rsfMRI over-connectivity and grooming behaviors. In the revised manuscript we have now expanded description of our statistical testing, by explicitly mentioning the planned comparisons, and by reporting that the significance levels in Figure 3 were all FDR corrected, a piece of information that we did not include in our original manuscript.

Caption of Figure 3: "... the striatum and PFC, respectively. *q < 0.05, **q < 0.01, FDR-corrected. Error bars represent SEM."

Methods: "... seed-to-VOI correlations. The statistical significance of intergroup changes was quantified using an unpaired 2-tailed Student's t-test (p < 0.05). Given our specific hypotheses, we applied FDR (Benjamini-Hochberg) to correct for multiple comparisons our set of planned contrasts. Our contrast of interest were (a) the functional connectivity alterations produced by loss of Tsc2 (i.e. Tsc2^{+/-} mutants vs. control littermates), (b) its rapamycin-induced rescue (i.e. Tsc2^{+/-} mutants treated with rapamycin vs. vehicle-treated Tsc2^{+/-} mutants), and (c) the degree of phenotypic normalization produced by rapamycin in Tsc2 mutants (i.e. vehicle-treated wild type mice vs. rapamycin-treated Tsc2^{+/-} mutants). We used the same FDR-correction strategy to correct intergroup comparisons of dendritic spine density and behavioral scores."

Q5. Also, unless I have missed, nowhere in the manuscript mentioned how "rescue" stats (blue contrast) was generated.

Thanks for the comment. We apologize for this omission. We used blue coloring to depict brain regions showing significantly decreased connectivity when comparing rapamycin-treated vs. vehicle-treated Tsc2^{+/-} mice (Figure 3B). We call this effect "rescue" as the corresponding map exhibits decreased rsfMRI

connectivity in the same cortico-limbic-striatal regions that are overconnected in *Tsc2* mutants. These regions are depicted in red in the same panel, and were obtained by comparing connectivity in control mice and *Tsc2* mutants. Similarly, the “rescue” maps obtained in seed-based connectivity analyses illustrate the connectivity difference between rapamycin-treated and vehicle-treated *Tsc2*^{+/-} mice. We have now edited the caption of Figure 3 of the revised manuscript as follows:

Caption Figure 3: “(B) Voxel-wise rsfMRI mapping showed a pattern of decreased long-range functional connectivity in rapamycin treated *Tsc2*^{+/-} mice (blue), recapitulating the anatomical regions showing increased rsfMRI connectivity in *Tsc2*^{+/-} mice (red-yellow coloring, left). Rescue here denotes the effect of rapamycin in *Tsc2* mutants i.e. areas of decreased rsfMRI connectivity in rapamycin-treated vs. vehicle-treated *Tsc2*^{+/-} mice. Global histogram analysis confirmed rescue of long-range hyper-connectivity in prefrontal and insular regions of rapamycin-treated mutants (right). (C) Spatial extension of the mouse default mode network in vehicle (top) and rapamycin (bottom) treated *Tsc2*^{+/-} mice. Between-group connectivity mapping and regional quantifications revealed that treatment with rapamycin rescued prefrontal rsfMRI hyperconnectivity in *Tsc2*^{+/-} mice to levels comparable to control mice (right). (D) Spatial extension of the salience network in vehicle (top) and rapamycin (bottom) treated *Tsc2*^{+/-} mice. Rapamycin completely rescued salience network hyperconnectivity in rapamycin treated *Tsc2*^{+/-} mice. Rescue in C and D denotes the effect of rapamycin in *Tsc2* mutant i.e. areas of decreased rsfMRI connectivity in rapamycin-treated vs. vehicle-treated *Tsc2*^{+/-} mice. “

Q6. Also, throughout this manuscript, statistical threshold was set at 2-tailed t-test at $P < 0.05$, this is relatively liberal. More specifics should be given about multiple comparisons.

We thank the reviewer for this comment, which allows us to clarify this aspect. As a matter of fact, all voxel-wise network maps and intergroup comparisons of our manuscript, in both mouse and human, were corrected for multiple comparisons using family-wise error (FWER) cluster correction at $p < 0.01$ as implemented in FSL, with the sole exception of the case-control global connectivity map of Figure 4B. This information was originally reported in our methods section (p. 23, 29 and 31, original manuscript). We however concur that our prior description might be confusing or not explicit enough. We have now more explicitly mentioned both in the text and in the corresponding figure captions that all maps were FWER corrected, and also the lack of such correction in Figure 4B. Given the highly focal nature of the insular clusters mapped in Figure 4B, the fact that that the obtained map did not survive formal cluster correction is not surprising. It should however be noted that a quantification of the corresponding global connectivity difference across subjects resulted in a significance level $p = 0.0015$ (Figure 4B), that is, a value greatly exceeding the nominal statistical threshold used for voxel-wise mapping ($|t| > 2$, $p < 0.05$). This observation, together with the exquisitely bilateral nature of the mapped effect, and the presence of a clearly identifiable involvement of insular over-connectivity across all the examined ASD sites (Figure 4A) as per our hypothesis greatly increase the biological plausibility of this result. We note that all the subsequent investigations, which led to the identification of mTOR-related transcriptional and connectivity signature, were cluster corrected. We now have edited the method section to make more explicit the multiple comparisons correction used as follows:

Methods (mouse): “Voxel-wise intergroup differences in local and long-range connectivity as well as for seed-based mapping were assessed using a 2-tailed Student's t-test ($|t| > 2$, $p < 0.05$) and family-wise error (FWER) cluster-corrected for multiple comparisons using a cluster threshold of $p < 0.01$ as implemented in FSL (Worsley et al., 1992).”

Methods (humans): “... age, sex and IQ. Inter-group comparisons were then carried out by using unpaired 2-tailed Student's t-test ($|t| > 2$, $p < 0.05$). Given the highly focal nature of the insular clusters mapped

with global connectivity (Figure 4B), the obtained map did not survive formal cluster correction. However, the exquisitely bilateral nature of the mapped effect and the presence of a clearly identifiable involvement of insular over-connectivity across all the examined ASD sites (Figure 4A) as per our hypothesis strongly support the biological plausibility of this result.”

Q7. Human fMRI data analysis. Motion artifacts are a practical problem in fMRI with ASD patients. The author clearly recognized this problem, and tried to deal with it. However, the subtypes showing Fig. 5C, especially in subtype 1, and subtype 3, the maps do not appear to have anatomical specificity. The reviewer wonders whether residual motion played a role.

Thanks for the comment. To rule out a possible confounding contribution of head motion artefacts to our human imaging findings, in our work we performed fMRI volume realignment based on motion correction, we regressed out 12 motion parameters and we carried out wavelet despiking as in (Patel et al., 2014). In an additional set of analyses, (Figure S2, original manuscript), we also performed strict volume scrubbing based on strict frame-wise displacement ($FD > 0.2\text{mm}$), and showed that insular hyper-connectivity in ASD was specifically identifiable also after this additional step. The results of these rigorous controls argue against a residual contribution of motion in our subtype mapping. This notion is further corroborated by a novel subject-wise quantification of frame-wise displacement in each subtype, which we designed to test whether one or more clusters could be dominated by head motion. Here we reasoned that, if residual motion played a role in the lack of specificity in subtypes 1 and 3, the corresponding scans in these two groups, but not in subtype 2 and 4, should exhibit greatly increased motion-flagged frames. However, intergroup comparisons of FD revealed that displacement is equally distributed across subtypes ($F = 0.1662$, $p = 0.9190$, one way ANOVA), suggesting that no specific cluster is dominated by head motion.

The relatively-low regional specificity of functional connectivity alterations in subtype 1 and 3 is instead the consequence of our choice to show connectivity maps of all subtypes using the same t-threshold, in the face of apparent cluster-specific differences in effect size. This decision was made to maximize consistency of cross-subtype thresholding at the cost of showing relatively spatially-undifferentiated difference maps. As reported in Figure S8 RXXX, by arbitrarily increasing thresholding to $|t| > 5$ (FWER cluster corrected) in subtype 1 and $|t| > 3$ (FWER cluster corrected) in subtype 3, it is possible to identify cluster-specific connectivity features. We have now added Supplementary Figure S8 to the revised manuscript and edited the corresponding text as follows:

Results: “... (n = 102) presented weak foci of frontal hyper-connectivity (Figure 5C and Supplementary Figure S8) ...”

Supplementary Figure S8. Subtype dependent regional specificity of insular hyper- and hypo-connectivity in ASD. Regional specificity of target regions of hyperconnectivity in subtype 1 and hypoconnectivity of subtype 3 is not apparent when a single t -threshold is used across clusters (top, Figure 5C), reflecting different effect size for the different subtypes. Subtype-specific connectional features are however apparent when an arbitrary subtype-dependent threshold is applied to the less spatially undifferentiated subtype 1 ($|t| > 5$, FWER cluster corrected) and subtype 3 ($|t| > 3$, FWER cluster corrected) (bottom).

Q8. To remove non-neuronal artefacts in the resting-state fMRI data, the covariates in confound signal regression of human fMRI data typically will include white matter signal in addition to CSF signal, and may include not just the average WM and CSF signal, but also their additional variations projected to higher-order principal components. I suggest inclusion of WM signals as covariates.

We thank the reviewer for the comment. While motion control is of paramount importance in rsfMRI preprocessing, as discussed above in response to a suggestion from reviewer #2, we believe regression of parenchymal signals should be carried out with caution, as increasing evidence points at a contribution of BOLD signal in white matter (Yarkoni et al., 2009; Grajauskas et al., 2019; Wang et al., 2020), and large signal regression can alter the sign and distribution of brain connectivity changes (Murphy et al., 2009). The effect of BOLD WM signal is also robust to the most state-of-the-art denoising strategies. For example, methods like multi-echo ICA (ME-ICA) (Kundu et al., 2017) are extremely good at identifying non-BOLD signal components based on their lack of decay across echoes, and then removing that signal from the data. Such methods remarkably increase signal to noise ratio, reliability, statistical power, etc (e.g., Lynch et al., 2020; Lombardo et al., 2016). In every single ME-ICA dataset we have encountered (e.g. co-author Lombardo has extensive experience in this area), all denoised datasets representing highly prominent BOLD-like signals still have at least one or two white matter components after non-BOLD signal was removed from the datasets. This experience as well as the cited literature above indicates that WM signal regression is a practice that is commonly done but also poorly understood and likely results in removal of true BOLD-like signals. Since WM is not a tissue compartment that would be of interest in whole-brain analysis of GM, the practice of WM-signal regression does not help to identify signal in that tissue. Rather, the practice simply risks removal of a poorly characterized, but likely BOLD-like signal from the data and could inadvertently affect the GM signals we care about. Thus, while WM regression seems to be a

practice that many have used in the past, our interpretation is that this practice persists without many in the field properly considering the detrimental effects it likely has on GM signal of interest.

To take the comment into account, we have nonetheless regressed white matter as well as the principal components of white matter using CompCorr (Behzadi et al., 2007) as suggested by the reviewer, and generated the corresponding global connectivity and seed-based analyses maps as in Figure 4. Regression of this set of covariates did not substantially alter the direction and overall anatomical location of our connectivity findings with respect to what we originally showed with our original pipeline (Supplementary Figure S11A). Together, these new analyses argue against a significant contribution of motion-related artefacts and physiological noise to the observed hyper-connectivity. We now report this set of control analyses as Supplementary Figure S11 in the revised manuscript.

Supplementary Figure S11. rsfMRI connectivity mapping upon mean white matter and white matter CompCorr regression. (A) Global connectivity mapping and (B) seed based correlation analysis carried out following the original denoising pipeline of our work, i.e. wavelet despiking, 12 motion parameters and mean CSF regression as in Patel et al., 2014. These are the same results we report in Figure 4. (C) Global connectivity mapping and (D) seed-based correlation analysis carried out following addition of mean white matter regression. (E) Global connectivity mapping and (F) seed based correlation analysis carried out following regressions of principal components of the white matter signal as implemented in ANTsR (<https://www.rdocumentation.org/packages/ANTsR/versions/1.0>). All the denoising strategies produced similar foci of hyper-connectivity in insular cortices (A-C-E) and fronto-insulo-striatal hyper-connectivity in ASD scans (B-D-E) arguing against a spurious origin for the hyper-connectivity we mapped in children with ASD.

Methods: "... To characterize motion and its impact on the timeseries, we computed FD and DVARS (Power et al., 2012). To further denoise the resting-state fMRI data by using a more stringent pipeline, we also carried out a set of supplementary analysis where we additionally regressed out either mean white matter signal (WM), or WM principal component as calculated with CompCorr (Behzadi et al., 2007) and implemented in ANTsR (<https://www.rdocumentation.org/packages/ANTsR/versions/1.0>).” Regression of this set of covariates did not alter the direction and anatomical location of our connectivity findings (Supplementary Figure S11), arguing against a significant contribution of motion-related artefacts and physiological noise to the observed rsfMRI hyper-connectivity. As these pipelines produced highly

consistent results, and given the controversy around the regression of WM components (Yarkoni et al., 2009; Grajauskas et al., 2019; Wang et al., 2020), all the subsequent analyses were carried out without WM regression.

Q9: On page 31, the authors performed data scrubbing to remove time points using stringent FD threshold. It's not clear after scrubbing, the statistics of remaining number of time points for each group, e.g. how many volumes remain after scrubbing for each group (range, mean \pm standard deviation etc) and whether there is group difference.

We thank the reviewer for the comment, as it gives us the opportunity to explain more in detail our scrubbing strategy. To strictly control for motion, we carried data volume scrubbing at a very stringent framewise-displacement threshold ($FD > 0.2$ mm). Before scrubbing, time-series length in the ASD and control groups were comparable (ASD: mean number of volumes = 186, $sd = 64$, range = 82-330; CTRL: mean number of volumes = 182, $SD = 63$, range = 82-330). Importantly, upon volume censoring, we retained only time-series with at least 50% of the original number of volumes as suggested by others (Tang et al., 2020). This resulted in the “a posteriori” exclusion of OHSU rsfMRI scans, a result of the low sample size (ASD, $n = 9$ and CTR, $n = 15$) and limited number of time-points ($n = 82$) in the time-series of this specific site. This procedure resulted in a final sample of $n = 108$ individuals with ASD and $n = 128$ CTR subjects. Predictably, the number of scrubbed volumes was higher in ASD than control subjects (ASD, mean volumes = 133, $SD = 45$, range = 58-289; CTR, mean volumes = 157, $SD = 52$, range = 58-294). Nonetheless, intergroup comparisons of global connectivity, as well as seed-based correlation mapping in motion scrubbed time-series, broadly replicated patterns of rsfMRI hyper-connectivity in children with ASD (Supplementary Figure S7 C). These results were confirmed also upon using the number of volumes as a nuisance covariate in our statistical testing of functional connectivity (Supplementary Figure S7D). Together, these investigations argue against a spurious contribution of head-motion to the functional changes mapped in our work.

We have now added this set of supplementary analysis to Supplementary Figure S7 (formerly Supplementary Figure S2) and described the resulting in the caption of the same figure as follows:

Supplementary Figure S7. Insular hyper-connectivity in children with ASD is identifiable also after strict control of in-scanner head motion with data scrubbing. A) In-scanner head motion during fMRI acquisition as measured with mean frame-wise displacement. Mean frame-wise displacement (FD) is higher in brain scans of children with ASD acquired at KKI ($t = 3.21$, $p = 0.002$), NYU ($t = 3.29$, $p = 0.001$) and UM ($t = 2.86$, $p = 0.006$) as compared to site-matched TDs. B) Percentage of volumes flagged with $FD > 0.2$ mm. C) Upon removal of volumes showing $FD > 0.2$ mm, we retained only time-series with at least 50% of the original number of volumes (Tang et al., 2020) for further functional connectivity analysis. Due to the low sample size (ASD, $n=9$ and CTR, $n=15$) and the limited number of time-points (volumes, $n=82$) of the OHSU site, subjects scanned at OHSU were removed from subsequent analyses. In total, for this confirmatory analysis we retained $n = 108$ individuals with ASD and $n = 128$ CTR subjects. Long-range connectivity mapping confirmed bilateral hyperconnectivity in anterior insular regions in children with ASD. Seed based mapping carried out on the same scrubbed data corroborated long-range hyperconnectivity between anterior insular areas and cortico-striatal targets. D) As length of the time-series of ASD subjects (number of volumes, mean = 133, $sd = 45$, range = 58-289) was shorter than that of CTR subjects (number of volumes, mean = 157, $sd = 52$, range = 58-294), we also included the number of volumes as a nuisance covariate in our statistical testing of functional connectivity. This set of analysis once again confirmed insular functional hyper-connectivity in the ASD cohort. $**p < 0.01$, $***p < 0.001$

Q10: There is a reasonable literature on human insular function connectivity (e.g. Deen B et al., cerebral cortex, 2011; Di Martino A et al. American Journal of Psychiatry. 2009). The anterior, middle and posterior insular have well established connectivity profile. For sanity check, it would be more convincing if the authors can present insular connectivity maps, and cross check with literature.

We thank the reviewer for the comment. To complement inter-group comparisons of seed based insular connectivity maps (Figure 4C), we report below insular network of TDs and ASD cohort separately (Supplementary Figure S5). Seed placement (depicted in green) is based on the anatomical localization of inter-group differences mapped with global connectivity (Figure 4B). This location largely overlaps with anterior and middle insula (Deen et al., 2011). In keeping with previous literature (Deen et al., 2011), connectivity probing of the salience network by seed based analysis of the insular cortex revealed robust functional connectivity to the dorsal anterior cingulate cortex (dACC) in TDs. A similar topographical network was found in ASD, with evidence of increased connectivity with the dACC and pregenual ACC (Palomero-Gallagher et al., 2019), another brain region that has been reported to be functionally connected to anterior insula (Di Martino et al., 2009), as depicted with intergroup differences in Figure 4C.

We now report this new analysis in Supplementary Figure S5. We have also edited the manuscript as follows:

Results: "...identified area of insular hyper-connectivity as a seed region (Supplementary Figure S5). The obtained case-control difference..."

Supplementary Figure S5. Functional connectivity of insular cortex as measured with seed based mapping. A) Seed location (in green). B) Seed based network of insular cortex in TDs and individuals with ASD.

5. Perhaps the weakest point of this manuscript is in-silico modelling that the authors claim to predict mTOR-related functional hyper-connectivity through increased interareal synaptic coupling. While the hypothesis is fascinating, the data presented is nothing more than mathematical exercise. There are numerous assumptions from synaptic excitability, to membrane potential fluctuation, to field potential and neuronal firing, and finally through neurovascular coupling (a black box) to BOLD fluctuation. Even with simultaneously acquired LFP/spiking and BOLD data, the neuroimaging field cannot agree on how to relate BOLD fluctuation to spontaneous neuronal activity. I feel it is exceedingly speculative to make such a claim.

Thanks for this comment. We agree that our claim that in-silico modelling is “predictive” of mTOR-related functional hyper-connectivity was too strong, and possibly misleading. We however believe our *in silico* modelling is not a simple mathematical exercise, but a significant result that helps support a putative mechanistic link between spine pathology and overconnectivity in our data. The overall goal of this *in silico* analysis was actually to test whether state-of-the-art whole brain models of rsfMRI dynamics could (at least theoretically) explain the rsfMRI connectivity values experimentally measured in *Tsc2^{+/-}* mice. Our excitement about these measurements reflects these two non-trivial observations:

- (i) Using a voxel-wise model of the mouse connectome, we could reliably simulate pattern of whole-brain functional connectivity that robustly converge with those empirically measured in both group of mice.
- (ii) The best empirical fit in *Tsc2* mutants was obtained by varying a single parameter (i.e. synaptic coupling strength - G) that can be conceptually linked to the synaptic pathology we observed in *Tsc2* mice. Importantly, (and once again, non-trivially) the synaptic coupling strength that more accurately reproduced the hyper-connectivity patterns observed in *Tsc2* mice is significantly higher than the corresponding value in control mice. Given the well characterized non-linear dynamics of these in silico models (Deco et al., 2014), these results are not a trivial consequence of the employed mathematical modelling, but instead they represent solid computational work suggesting that (at least theoretically), the observed hyper-connectivity could be explained by the increased spine density we experimentally measured in our *Tsc2* mice.

To demonstrate that this result is non-trivially affected by any linear variation of any parameter of the model affecting neuronal coupling, we have now run the same simulation by manipulating the overall effective external input current I_0 , a parameter linearly related to membrane excitability, and we then re-

computed our modelling. We show below for the reviewers only the results of this analysis, which show that no convergence between simulated and measured functional connectivity values can be achieved by uniquely manipulating this parameter across groups (Reviewer Figure R2).

Reviewer Figure R2. Whole brain computational modelling. No optimal fitting of functional connectivity values can be achieved by varying IO within a physiological range [0.3 - 0.4] (Deco et al., 2014).

In the revised manuscript we have now reworded the corresponding paragraph as follows:

Title of paragraph: mTOR-related functional hyper-connectivity can be modelled in silico by increasing synaptic coupling

Results: "...inputs in distributed cortical circuits (Yuste, 2011). The recent development of whole-brain computational models of rsfMRI connectivity (Deco et al., 2014) allowed us to test the theoretical validity of this assumption via in silico manipulations of macro-scale interareal coupling strength. To this aim, we.."

Results: "... This finding supports a putative association between mTOR-related synaptic over-abundance and rsfMRI hyperconnectivity, suggesting that rsfMRI over-synchronization in high connection density components of the DMN and salience networks (Coletta et al., 2020) may emerge out of a generalized increase in interareal coupling strength ..."

In the revised version we have also probed the presence of genotype-dependent differences in hemodynamic response, another key parameter of our model, by extrapolating the hemodynamic response function in our experimental data using the method described by (Wu et al., 2013). This analysis revealed broadly comparable BOLD peak amplitude ($t = 0.20$; $p = 0.75$), time to peak ($t = 0.63$; $p = 0.52$) and FWHM ($t = 0.31$; $p = 0.75$) across genotypes (Supplementary Figure S9). These results suggest that the hemodynamic response is unimpaired in $Tsc2^{+/-}$ mice, arguing against a possible contribution of this factor in the emergence of the observed phenotype, and ruling out further explorations of this parameter in our whole brain modelling. While the reviewer is correct in stating that the wide space of parameters constituting this model could not be systematically explored, and that other contributions could be involved in the hyper-connectivity phenotype of $Tsc2$ mice, these findings increase our confidence in the putative mechanistic value of our simulations, which were valuable in hinting at a possible mechanism to be tested in real-world rescue studies with rapamycin.

In the revised manuscript we have added these measurements as follows:

Methods: “... This observation argues against a significant confounding contribution of anesthesia to our functional measures. To further rule out a neurovascular origin of the mapped functional changes, we measured the hemodynamic response in the PFC, a brain region showing prominent hyper-connectivity in *Tsc2*^{+/-} mice. Intergroup genotype-dependent comparisons revealed unimpaired hemodynamic response in *Tsc2*^{+/-} mice (BOLD peak amplitude, *p* = 0.75; time to peak, *p* = 0.52; FWHM, *p* = 0.75, Supplementary Figure 9). These results corroborate a neural (rather than neurovascular) origin of the rsfMRI changes in *Tsc2*^{+/-} mice”.

Supplementary Figure S9. Hemodynamic response function exhibits canonical features in *Tsc2*^{+/-} mice. A) Hemodynamic response in PFC in *Tsc2*^{+/-} mutants and +/+ control mice. B) BOLD peak amplitude, time to peak and FWHM of the hemodynamic response in *tsc2*^{+/-} mice are comparable to the corresponding parameters observed in control mice.

References

- Abrams DA, Lynch CJ, Cheng KM, Phillips J, Supekar K, Ryali S, Uddin LQ, Menon V (2013) Underconnectivity between voice-selective cortex and reward circuitry in children with autism. *Proceedings of the National Academy of Sciences* 110:12060-12065.
- Behzadi Y, Restom K, Liu J, Liu TT (2007) A component based noise correction method (CompCor) for BOLD and perfusion based fMRI. *NeuroImage* 37:90-101.
- Bertero A, Liska A, Pagani M, Parolisi R, Masferrer ME, Gritti M, Pedrazzoli M, Galbusera A, Sarica A, Cerasa A, Buffelli M, Tonini R, Buffo A, Gross C, Pasqualetti M, Gozzi A (2018) Autism-associated 16p11.2 microdeletion impairs prefrontal functional connectivity in mouse and human BRAIN 141:2055–2065.
- Buckner RL, Krienen FM (2013) The evolution of distributed association networks in the human brain. *Trends in Cognitive Sciences* 17:648-665.
- Canella C, Rocchi F, Noei S, Gutierrez-Barragan D, Coletta L, Galbusera A, Vassanelli S, Pasqualetti M, Iurilli G, Panzeri S (2020) Cortical silencing results in paradoxical fMRI overconnectivity. *bioRxiv*.
- Chang C, Leopold DA, Schölvinck ML, Mandelkow H, Picchioni D, Liu X, Ye FQ, Turchi JN, Duyn JH (2016) Tracking brain arousal fluctuations with fMRI. *Proceedings of the National Academy of Sciences* 113:4518.
- Ciric R, Wolf DH, Power JD, Roalf DR, Baum GL, Ruparel K, Shinohara RT, Elliott MA, Eickhoff SB, Davatzikos C, Gur RC, Gur RE, Bassett DS, Satterthwaite TD (2017) Benchmarking of participant-level confound regression strategies for the control of motion artifact in studies of functional connectivity. *NeuroImage* 154:174-187.
- Cole MW, Pathak S, Schneider W (2010) Identifying the brain's most globally connected regions. *NeuroImage* 49:3132-3148.
- Coletta L, Pagani M, Whitesell JD, Harris JA, Bernhardt B, Gozzi A (2020) Network structure of the mouse brain connectome with voxel resolution. *Science Advances* 6:eabb7187.
- Deco G, Ponce-Alvarez A, Hagmann P, Romani GL, Mantini D, Corbetta M (2014) How local excitation–inhibition ratio impacts the whole brain dynamics. *Journal of Neuroscience* 34:7886-7898.
- Deen B, Pitskel NB, Pelphrey KA (2011) Three systems of insular functional connectivity identified with cluster analysis. *Cerebral cortex* 21:1498-1506.
- Di Martino A, Kelly C, Grzadzinski R, Zuo X-N, Mennes M, Mairena MA, Lord C, Castellanos FX, Milham MP (2011) Aberrant striatal functional connectivity in children with autism. *Biological psychiatry* 69:847-856.
- Di Martino A, Shehzad Z, Kelly C, Roy AK, Gee DG, Uddin LQ, Gotimer K, Klein DF, Castellanos FX, Milham MP (2009) Relationship between cingulo-insular functional connectivity and autistic traits in neurotypical adults. *American Journal of Psychiatry* 166:891-899.
- Di Martino A et al. (2014) The autism brain imaging data exchange: towards a large-scale evaluation of the intrinsic brain architecture in autism. *Mol Psychiatry* 19:659-667.
- Gazestani VH, Pramparo T, Nalabolu S, Kellman BP, Murray S, Lopez L, Pierce K, Courchesne E, Lewis NE (2019) A perturbed gene network containing PI3K–AKT, RAS–ERK and WNT– β -catenin pathways in leukocytes is linked to ASD genetics and symptom severity. *Nature neuroscience* 22:1624-1634.
- Goulden N, Khusnulina A, Davis NJ, Bracewell RM, Bokde AL, McNulty JP, Mullins PG (2014) The salience network is responsible for switching between the default mode network and the central executive network: replication from DCM. *Neuroimage* 99:180-190.
- Gozzi A, Schwarz AJ (2016) Large-scale functional connectivity networks in the rodent brain. *Neuroimage* 127:496-509.

- Grandjean J, Zerbi V, Balsters JH, Wenderoth N, Rudin M (2017) Structural Basis of Large-Scale Functional Connectivity in the Mouse. *The Journal of Neuroscience* 37:8092-8101.
- Grandjean J, Canella C, Anckaerts C, Ayranci G, Bougacha S, Bienert T, Buehlmann D, Coletta L, Gallino D, Gass N (2020) Common functional networks in the mouse brain revealed by multi-centre resting-state fMRI analysis. *Neuroimage* 205:116278.
- Gutierrez-Barragan D, Basson MA, Panzeri S, Gozzi A (2019) Infralow State Fluctuations Govern Spontaneous fMRI Network Dynamics. *Current Biology* 29:2295-2306.e2295.
- Hsieh CCj, Lo YC, Li SJ, Lin TC, Chang CW, Chen TC, Yang SH, Lee YC, Chen YY (2021) Detection of endophenotypes associated with neuropsychiatric deficiencies in a mouse model of tuberous sclerosis complex using diffusion tensor imaging. *Brain Pathology* 31:4-19.
- Hull JV, Jacokes ZJ, Torgerson CM, Irimia A, Van Horn JD (2017) Resting-State Functional Connectivity in Autism Spectrum Disorders: A Review. *Frontiers in Psychiatry* 7.
- Lee DY (2015) Roles of mTOR signaling in brain development. *Experimental neurobiology* 24:177.
- Liska A, Galbusera A, Schwarz AJ, Gozzi A (2015) Functional connectivity hubs of the mouse brain. *Neuroimage* 115:281-291.
- Lu H, Zou Q, Gu H, Raichle ME, Stein EA, Yang Y (2012) Rat brains also have a default mode network. *Proceedings of the National Academy of Sciences* 109:3979-3984.
- Mantini D, Gerits A, Nelissen K, Durand J-B, Joly O, Simone L, Sawamura H, Wardak C, Orban GA, Buckner RL (2011) Default mode of brain function in monkeys. *Journal of Neuroscience* 31:12954-12962.
- Matsui T, Murakami T, Ohki K (2016) Transient neuronal coactivations embedded in globally propagating waves underlie resting-state functional connectivity. *Proceedings of the National Academy of Sciences* 113:6556-6561.
- Menon V (2011) Large-scale brain networks and psychopathology: a unifying triple network model. *Trends in Cognitive Sciences* 15:483-506.
- Murphy K, Birn RM, Handwerker DA, Jones TB, Bandettini PA (2009) The impact of global signal regression on resting state correlations: are anti-correlated networks introduced? *Neuroimage* 44:893-905.
- Onda H, Lueck A, Marks PW, Warren HB, Kwiatkowski DJ (1999) Tsc2+/-mice develop tumors in multiple sites that express gelsolin and are influenced by genetic background. *The Journal of clinical investigation* 104:687-695.
- Pagani M, Bifone A, Gozzi A (2016a) Structural covariance networks in the mouse brain. *Neuroimage* 129:55-63.
- Pagani M, Damiano M, Galbusera A, Tsafaris SA, Gozzi A (2016b) Semi-automated registration-based anatomical labelling, voxel based morphometry and cortical thickness mapping of the mouse brain. *J Neurosci Methods* 267:62-73.
- Pagani M, Bertero A, Liska A, Galbusera A, Sabbioni M, Barsotti N, Colenbier N, Marinazzo D, Scattoni ML, Pasqualetti M, Gozzi A (2019) Deletion of autism risk gene Shank3 disrupts prefrontal connectivity. *The Journal of Neuroscience*:2529-2518.
- Palomero-Gallagher N, Hoffstaedter F, Mohlberg H, Eickhoff SB, Amunts K, Zilles K (2019) Human pregenual anterior cingulate cortex: structural, functional, and connective heterogeneity. *Cerebral Cortex* 29:2552-2574.
- Parikhshak NN, Swarup V, Belgard TG, Irimia M, Ramaswami G, Gandal MJ, Hartl C, Leppa V, Ubieta LdIT, Huang J, Lowe JK, Blencowe BJ, Horvath S, Geschwind DH (2016) Genome-wide changes in lncRNA, splicing, and regional gene expression patterns in autism. *Nature* 540:423.
- Patel AX, Kundu P, Rubinov M, Jones PS, Vértes PE, Ersche KD, Suckling J, Bullmore ET (2014) A wavelet method for modeling and despiking motion artifacts from resting-state fMRI time series. *Neuroimage* 95:287-304.

- Rosina E, Battan B, Siracusano M, Di Criscio L, Hollis F, Pacini L, Curatolo P, Bagni C (2019) Disruption of mTOR and MAPK pathways correlates with severity in idiopathic autism. *Translational psychiatry* 9:1-10.
- Rubinov M, Sporns O (2011) Weight-conserving characterization of complex functional brain networks. *NeuroImage* 56:2068-2079.
- Russo C, Nastro A, Cicala D, De Liso M, Covelli EM, Cinalli G (2020) Neuroimaging in tuberous sclerosis complex. *Child's Nervous System* 36:2497-2509.
- Saad ZS, Gotts SJ, Murphy K, Chen G, Jo HJ, Martin A, Cox RW (2012) Trouble at rest: how correlation patterns and group differences become distorted after global signal regression. *Brain connectivity* 2:25-32.
- Sahin M, Sur M (2015) Genes, circuits, and precision therapies for autism and related neurodevelopmental disorders. *Science* 350.
- Sato A (2016) mTOR, a potential target to treat autism spectrum disorder. *CNS & Neurological Disorders- Drug Targets (Formerly Current Drug Targets-CNS & Neurological Disorders)* 15:533-543.
- Sato A, Kasai S, Kobayashi T, Takamatsu Y, Hino O, Ikeda K, Mizuguchi M (2012) Rapamycin reverses impaired social interaction in mouse models of tuberous sclerosis complex. *Nature communications* 3:1-9.
- Schölvinck ML, Maier A, Frank QY, Duyn JH, Leopold DA (2010) Neural basis of global resting-state fMRI activity. *Proceedings of the National Academy of Sciences* 107:10238-10243.
- Schwarz CG, Reid RI, Gunter JL, Senjem ML, Przybelski SA, Zuk SM, Whitwell JL, Vemuri P, Josephs KA, Kantarci K, Thompson PM, Petersen RC, Jack CR (2014) Improved DTI registration allows voxel-based analysis that outperforms Tract-Based Spatial Statistics. *NeuroImage* 94:65-78.
- Sforazzini F, Schwarz AJ, Galbusera A, Bifone A, Gozzi A (2014) Distributed BOLD and CBV-weighted resting-state networks in the mouse brain. *Neuroimage* 87:403-415.
- Smith SM, Jenkinson M, Johansen-Berg H, Rueckert D, Nichols TE, Mackay CE, Watkins KE, Ciccarelli O, Cader MZ, Matthews PM (2006a) Tract-based spatial statistics: voxelwise analysis of multi-subject diffusion data. *Neuroimage* 31:1487-1505.
- Smith SM, Jenkinson M, Johansen-Berg H, Rueckert D, Nichols TE, Mackay CE, Watkins KE, Ciccarelli O, Cader MZ, Matthews PM, Behrens TEJ (2006b) Tract-based spatial statistics: Voxelwise analysis of multi-subject diffusion data. *NeuroImage* 31:1487-1505.
- Spera G, Retico A, Bosco P, Ferrari E, Palumbo L, Oliva P, Muratori F, Calderoni S (2019) Evaluation of altered functional connections in male children with autism spectrum disorders on multiple-site data optimized with machine learning. *Frontiers in psychiatry* 10:620.
- Sripada RK, King AP, Welsh RC, Garfinkel SN, Wang X, Sripada CS, Liberzon I (2012) Neural dysregulation in posttraumatic stress disorder: evidence for disrupted equilibrium between salience and default mode brain networks. *Psychosomatic medicine* 74:904.
- Supekar K, Uddin L-á, Khouzam A, Phillips J, Gaillard W-á, Kenworthy L-á, Yerys B-á, Vaidya C-á, Menon V (2013a) Brain Hyperconnectivity in Children with Autism and its Links to Social Deficits. *Cell Reports* 5:738-747.
- Supekar K, Uddin LQ, Khouzam A, Phillips J, Gaillard WD, Kenworthy LE, Yerys BE, Vaidya CJ, Menon V (2013b) Brain hyperconnectivity in children with autism and its links to social deficits. *Cell reports* 5:738-747.
- Tang S, Sun N, Floris DL, Zhang X, Di Martino A, Yeo BT (2020) Reconciling dimensional and categorical models of autism heterogeneity: a brain connectomics and behavioral study. *Biological psychiatry* 87:1071-1082.
- Tavazoie SF, Alvarez VA, Ridenour DA, Kwiatkowski DJ, Sabatini BL (2005) Regulation of neuronal morphology and function by the tumor suppressors Tsc1 and Tsc2. *Nature neuroscience* 8:1727-1734.

- Touroutoglou A, Bliss-Moreau E, Zhang J, Mantini D, Vanduffel W, Dickerson BC, Barrett LF (2016) A ventral salience network in the macaque brain. *Neuroimage* 132:190-197.
- Tsai P-J, Keeley RJ, Carmack SA, Vendruscolo JC, Lu H, Gu H, Vendruscolo LF, Koob GF, Lin C-P, Stein EA (2020) Converging structural and functional evidence for a rat salience network. *Biological Psychiatry* 88:867-878.
- Turchi J, Chang C, Ye FQ, Russ BE, Yu DK, Cortes CR, Monosov IE, Duyn JH, Leopold DA (2018) The Basal Forebrain Regulates Global Resting-State fMRI Fluctuations. *Neuron* 97:940-952.e944.
- Uddin LQ, Supekar K, Lynch CJ, Khouzam A, Phillips J, Feinstein C, Ryali S, Menon V (2013) Salience network–based classification and prediction of symptom severity in children with autism. *JAMA psychiatry* 70:869-879.
- Wang J, Wang L, Zang Y, Yang H, Tang H, Gong Q, Chen Z, Zhu C, He Y (2009) Parcellation-dependent small-world brain functional networks: A resting-state fMRI study. *Human brain mapping* 30:1511-1523.
- Wen H, Liu Z (2016) Broadband electrophysiological dynamics contribute to global resting-state fMRI signal. *Journal of Neuroscience* 36:6030-6040.
- Whitesell JD, Liska A, Coletta L, Hirokawa KE, Bohn P, Williford A, Groblewski PA, Graddis N, Kuan L, Knox JE (2021) Regional, Layer, and Cell-Type-Specific Connectivity of the Mouse Default Mode Network. *Neuron* 109:545-559. e548.
- Whitesell JD et al. (2020) Regional, layer, and cell-class specific connectivity of the mouse default mode network. *Neuron*:2020.2005.2013.094458.
- Winden KD, Ebrahimi-Fakhari D, Sahin M (2018) Abnormal mTOR activation in autism. *Annual review of neuroscience* 41:1-23.
- Worsley KJ, Evans AC, Marrett S, Neelin P (1992) A three-dimensional statistical analysis for CBF activation studies in human brain. *Journal of Cerebral Blood Flow & Metabolism* 12:900-918.
- Wu G-R, Liao W, Stramaglia S, Ding J-R, Chen H, Marinazzo D (2013) A blind deconvolution approach to recover effective connectivity brain networks from resting state fMRI data. *Medical image analysis* 17:365-374.
- You X, Norr M, Murphy E, Kuschner E, Bal E, Gaillard W, Kenworthy L, Vaidya CJ (2013) Atypical modulation of distant functional connectivity by cognitive state in children with Autism Spectrum Disorders. *Frontiers in human neuroscience* 7:482.
- Yuste R (2011) Dendritic Spines and Distributed Circuits. *Neuron* 71:772-781.
- Zerbi V, Pagani M, Markicevic M, Matteoli M, Pozzi D, Fagiolini M, Bozzi Y, Galbusera A, Scattoni ML, Provenzano G (2020) Brain mapping across 16 autism mouse models reveals a spectrum of functional connectivity subtypes. *bioRxiv*.

Reviewers' Comments:

Reviewer #1:

Remarks to the Author:

Thank you. I have no additional comments.

Reviewer #2:

Remarks to the Author:

This is an outstanding study that will help provide a mechanistic understanding of important developmental features underlying many instances of autism. Importantly, the work corroborates that appropriate timing of mTOR inhibition by rapamycin corrects ASD related deficits. The study also provides foundational insights on parallels between the TSC mouse model and genuine ASD.

There are a few critiques and comments

1. In addition to the important report of increased network activity due to lack of net synaptic pruning during a specific developmental period, the authors ought to mention that there can be cell intrinsic maturational changes that will regulate connectivity that may be corrected by mTOR inhibition.

PMID: 31913125 reports that the primary neurons in the striatum rely on mTOR/ macroautophagy (see PMID: 32296308) to develop a stable resting state at the same developmental time that the authors are using rapamycin in the present study.

Thus, in their experiments, mTOR inhibition is likely exerting effects in addition to correcting synaptic pruning, and the authors should mention that such additional mechanisms, importantly including cell intrinsic properties, may also underlie the normalization of network activity, and cortico-striatal loops may also be involved in the ASD network phenotypes.

2. Figure 1A shows a decrease in dendritic spine density in insular cortex, and an example from Layer 5, but does not clearly state in the legend or results which layers contribute to the data in the bar graph, please be explicit. Note the results say basal dendrites of layer V somatosensory but do not explicitly state the regions in the insula.

3) The authors identified frontal-insular-striatal functional hyperconnectivity as an endophenotype for a subset of autism patients. This endophenotype can be recapitulated in *Tsc2*^{+/-} mouse models, and normalized by mTOR inhibition, suggesting a mechanistic link between mTOR hyperactivation and functional hyperconnectivity, a point the authors can make in the Discussion.

4) The Discussion should be clear that the data suggest but do not provide strong evidence to support the contribution of excessive spines to functional hyperconnectivity. Although the authors have excluded the influence of brain structural connectivity (measured by DTI and MBP staining, Lines 135-160) and physiological changes (see methods), it is difficult to determine the causal relationship between the dendritic changes in cortical projection neurons and the functional hyperconnectivity between cortical and subcortical regions the mice with somatic *Tsc2*^{+/-} mutation. The increased synchrony between prefrontal cortex, insular cortex and striatum could be attributed to enhanced local neural activities in each of these three regions, and mTOR hyperactivation may contribute to neural hyperactivities in different neuronal types within each brain region through different mechanisms, e.g., membrane excitability, synaptic changes and metabolic activities, all of which may impact the fMRI BOLD signal.

5) The increase in mature dendritic spine density could either enhance or impair (PMID: 26929363) synaptic plasticity.

6) How sensitive are the fractional anisotropy and retrograde tracing in finding changes in synaptic coupling between regions? Do the authors think that the excess spines are synaptically connected to distal branches of axons, in which case, this would not show up in these techniques.

7) I'm surprised that the authors did not look for rsfMRI data on TSC patients with autistic

behaviors, since they use a mouse that's a model for TSC. Perhaps TSC patients would show similarities to "Group 2" of the ASD group. This would further strengthen the correlation. Were all of the ASD patients tested for TSC? Or did any have TSC-like stigmata?

8) References need to be updated. For example, "Zerbi et al., 2020 (Line 414)"± and "Davis et al., 2019 (Line 405)"± could not be located.

9) As a reader, I am unfamiliar with the use of the terms "dyadic" and "non-dyadic" in the context used and would appreciate a definition.

10) "Synopsis" is misspelled in at least a couple of instances.

11) Figure 1 – I can't see the +/- spine figure in A

12) The authors do rsfMRI imaging of mice at P28, a good time for this study. In future work, it would be interesting to do it again after the normal period of spine pruning to see if the connectivity normalizes or persists.

Reviewer #3:

Remarks to the Author:

The authors have addressed all of my comments. This is a very nice cross-species study that will be an important contribution to the field. Only one minor comment: Spelling error line 370: 'synapsis' should be 'synapses'

Reviewer #4:

Remarks to the Author:

The authors did a good job in addressing my concerns. In particular, additional fMRI data analyses are performed that incorporate different strategies in handling motion artifacts, a practical problem that is likely inherent in imaging ASD patients, and recapitulate the fMRI finding, which is reassuring. I have no further comments.

We would like to thank the Editor and Reviewers for their positive evaluations, constructive comments, and for the opportunity to submit a revised manuscript. Please find below a point-by-point response to the reviewers' comments (our response in **blue** typeface). We have also added below the text edits we have introduced in the manuscript in response to the reviewers' comments (yellow highlight).

REVIEWERS' COMMENTS

Reviewer #1 (Remarks to the Author):

Thank you. I have no additional comments.

We thank the reviewer for the positive appraisal of our work and for recommending its publication.

Reviewer #2 (Remarks to the Author):
(Replacement reviewer)

This is an outstanding study that will help provide a mechanistic understanding of important developmental features underlying many instances of autism. Importantly, the work corroborates that appropriate timing of mTOR inhibition by rapamycin corrects ASD related deficits. The study also provides foundational insights on parallels between the TSC mouse model and genuine ASD.

We thank the new reviewer for the words of appreciation.

There are a few critiques and comments

1. In addition to the important report of increased network activity due to lack of net synaptic pruning during a specific developmental period, the authors ought to mention that there can be cell intrinsic maturational changes that will regulate connectivity that may be corrected by mTOR inhibition.

PMID: 31913125 reports that the primary neurons in the striatum rely on mTOR/macroautophagy (see PMID: 32296308) to develop a stable resting state at the same developmental time that the authors are using rapamycin in the present study.

Thus, in their experiments, mTOR inhibition is likely exerting effects in addition to correcting synaptic pruning, and the authors should mention that such additional mechanisms, importantly including cell intrinsic properties, may also underlie the normalization of network activity, and cortico-striatal loops may also be involved in the ASD network phenotypes.

We thank the reviewer for bringing to our attention these recent studies linking mTOR activity and autophagy to maturational changes that may affect neuronal excitability (Lieberman et al., 2020a; Lieberman et al., 2020b). In the revised manuscript, we have now mentioned the two abovementioned studies as part of expanded discussion of non-synaptic mechanisms that could contribute, in addition to increased dendritic spine density, to the observed hyper-connectivity phenotype in Tsc2 mutants.

Discussion: “Collectively, these observations support a view in which the density and functionality of dendritic spines can crucially contribute to etiologically-relevant functional dysconnectivity in autism. However, the direction and location of the ensuing dysfunctional coupling are likely to be strongly biased by additional pathophysiological components within the ASD spectrum, including maladaptive microcircuit homeostasis (i.e. E/I imbalance (Filipello et al., 2018; Trakoshis et al., 2020)), developmental miswiring (Liska et al., 2018) or alterations in modulatory activity governing brain-wide coupling (Gutierrez-Barragan et al., 2019). Moreover, more subtle maturational mechanisms could conceivably compound synaptic abnormalities, or independently contribute to the generation of the aberrant functional coupling. These might include mTOR-related maturational alterations in striatal excitability (Lieberman et al., 2020a; Lieberman et al., 2020b) or maladaptive synaptic plasticity and metabolic activity (Li et al., 2016), all of which may impact rsfMRI BOLD signal. ”

2. Figure 1A shows a decrease in dendritic spine density in insular cortex, and an example from Layer 5, but does not clearly state in the legend or results which layers contribute to the data in the bar graph, please be explicit. Note the results say basal dendrites of layer V somatosensory but do not explicitly state the regions in the insula.

We thank the reviewer for pointing out this omission. Our measurements were performed in layer V of the insular cortex as well. In the revised manuscript we have reported this information as follows.

Methods: “... Spine density quantification was performed on basal dendrites of layer V pyramidal neurons both in secondary somatosensory cortex (SS2) as in (Tang et al., 2014), and in neighboring anterior insula...”

3) The authors identified frontal-insular-striatal functional hyperconnectivity as an endophenotype for a subset of autism patients. This endophenotype can be recapitulated in Tsc2^{+/-} mouse models, and normalized by mTOR inhibition, suggesting a mechanistic link between mTOR hyperactivation and functional hyperconnectivity, a point the authors can make in the Discussion.

We thank the reviewer for the suggestion to more explicitly emphasize a possible link between mTOR hyper-activation and functional hyperconnectivity in humans. To this end, we have briefly edited the discussion as following:

Discussion: "... The observation of strikingly conserved cross-species circuitual dysfunction and its enrichment in ASD-dysregulated mTOR-interacting genes support the validity of our approach, establishing a possible mechanistic link between mTOR pathway hyperactivation and fronto-insular-striatal hyper-connectivity endophenotypes in idiopathic autism. These findings advance our understanding of ASD pathology in humans, revealing a neural circuitry that is sensitive to mTOR-related maturational derailment in segregable subpopulations of patients.

4) The Discussion should be clear that the data suggest but do not provide strong evidence to support the contribution of excessive spines to functional hyperconnectivity. Although the authors have excluded the influence of brain structural connectivity (measured by DTI and MBP staining, Lines 135-160) and physiological changes (see methods), it is difficult to determine the causal relationship between the dendritic changes in cortical projection neurons and the functional hyperconnectivity between cortical and subcortical regions the mice with somatic *Tsc2*^{+/-} mutation. The increased synchrony between prefrontal cortex, insular cortex and striatum could be attributed to enhanced local neural activities in each of these three regions, and mTOR hyperactivation may contribute to neural hyperactivities in different neuronal types within each brain region through different mechanisms, e.g., membrane excitability, synaptic changes and metabolic activities, all of which may impact the fMRI BOLD signal.

We thank the reviewer for this comment. We have now more explicitly mentioned a possible contribution of other mTOR-related maturational to our findings as follows:

Discussion: "Collectively, these observations support a view in which the density and functionality of dendritic spines can crucially contribute to etiologically-relevant functional dysconnectivity in autism. However, the direction and location of the ensuing dysfunctional coupling are likely to be strongly biased by additional pathophysiological components within the ASD spectrum, including maladaptive microcircuit homeostasis (i.e. E/I imbalance (Filipello et al., 2018; Trakoshis et al., 2020)), developmental miswiring (Liska et al., 2018) or alterations in modulatory activity governing brain-wide coupling (Gutierrez-Barragan et al., 2019). Moreover, more subtle maturational mechanisms could conceivably compound synaptic abnormalities, or independently contribute to the generation of the aberrant functional coupling. These might include mTOR-related maturational alterations in striatal excitability (Lieberman et al., 2020a; Lieberman et al., 2020b) or maladaptive synaptic plasticity and metabolic activity (Li et al., 2016), all of which may impact rsfMRI BOLD signal."

5) The increase in mature dendritic spine density could either enhance or impair (PMID: 26929363) synaptic plasticity.

We thank the reviewer for this comment. In the revised manuscript we have included this reference in the discussion as follows:

Discussion: “Collectively, these observations support a view in which the density and functionality of dendritic spines can crucially contribute to etiologically-relevant functional dysconnectivity in autism. However, the direction and location of the ensuing dysfunctional coupling are likely to be strongly biased by additional pathophysiological components within the ASD spectrum, including maladaptive microcircuit homeostasis (i.e. E/I imbalance (Filipello et al., 2018; Trakoshis et al., 2020)), developmental miswiring (Liska et al., 2018) or alterations in modulatory activity governing brain-wide coupling (Gutierrez-Barragan et al., 2019). Moreover, more subtle maturational mechanisms could conceivably compound synaptic abnormalities, or independently contribute to the generation of the aberrant functional coupling. These might include mTOR-related maturational alterations in striatal excitability (Lieberman et al., 2020a; Lieberman et al., 2020b) or maladaptive synaptic plasticity and metabolic activity (Li et al., 2016), all of which may impact rsfMRI BOLD signal. ”

6) How sensitive are the fractional anisotropy and retrograde tracing in finding changes in synaptic coupling between regions? Do the authors think that the excess spines are synaptically connected to distal branches of axons, in which case, this would not show up in these techniques.

We thank the reviewer for this comment, which allows us to better clarify the goal of these anatomical investigations. Fractional anisotropy is an imaging parameter sensitive to microscale white matter integrity and myelination (Bertero et al., 2018). The relatively coarse resolution of diffusion tensor imaging (DTI) however makes FA measurements a reliable proxy for myelination integrity only in major white matter bundles of the mouse brain. This parameter is therefore virtually insensitive to changes in synaptic coupling and density, as these occur at a biophysical scale not accessible by diffusion tensor imaging. Similarly, between-group differences in synaptic density and coupling are unlikely to be detectable with the employed retrograde tracing, as the injection radius of the recombinant rabies virus employed is not sensitive to distal versus proximal dendritic synaptic buttons, and the between-group differences in synaptic density observed in *Tsc2* mutants are not large enough to warrant different retrograde labelling efficiency as a function of this parameter (Ugolini, 1995; Callaway and Luo, 2015). For these reasons, we employed FA mapping and retrograde tracing to assess whether macro- and meso-scale axonal organization of *Tsc2* mutant and control mice would be broadly comparable, without expecting either of these readouts to display any genotype-dependent synaptic bias. More subtle reorganizations at the subcellular level, including a possible distal expression of excess synaptic density in *Tsc2* mice would therefore not be detectable with these methods.

In order to comply with editorial requirements about manuscript length, we have not been able to introduce a discussion of a number of relevant comments, including this specific point.

7) I’m surprised that the authors did not look for rsfMRI data on TSC patients with autistic behaviors, since they use a mouse that’s a model for TSC. Perhaps TSC patients would show

similarities to “Group 2” of the ASD group. This would further strengthen the correlation. Were all of the ASD patients tested for TSC? Or did any have TSC-like stigmata?

We thank the reviewer for this comment. The scans of ASD patients included in our study were downloaded from the ABIDE collection (Di Martino et al., 2014), and are devoid of genetic testing or clinical screening for pathological traits that can be attributed to dysfunctional Tsc2-related mutations (e.g. TSC-like stigmata). We are aware that rsfMRI network mapping in infants harboring TSc2 mutations has been recently published (Ahtam et al., 2019). However, we did not consider requesting those authors their scans because their study is crucially devoid of a reference, control population (i.e. typically developing infants) to be used as contrast for brain connectivity maps in Tsc2-harboring individuals. The presence of carefully-balanced control scans in TD cohorts is unfortunately a key prerequisite for any meaningful and interpretable form of brain mapping in pathological populations. It should also be pointed out that Ahtam’s TSc2 patients sample is composed by infants (age range: 3-36 months), an age window not comparable to the ABIDE sample we selected here (age range 6-13 years) and that we cannot probe in mice for technical reasons (that would P7 or so, an age in which we cannot properly control physiological parameters for fMRI in the mouse).

In the revised manuscript we have mentioned this point as follows:

Discussion: “Notably, findings in mouse models are paralleled by the observation in the present study of four distinct connectivity profile in the insular cortex, some of which exhibiting clearly opposing functional topography. Future applications of the novel cross-species, multi-scale research platform we describe here might help unravel the connectional heterogeneity of ASD by permitting the identification of ASD subtypes (Lombardo et al., 2019). Future case-control imaging investigations of individuals harboring Tsc2 mutations (Ahtam et al., 2019) might also crucially help test the validity of our predictions and their mechanistic significance in relevant patient populations.”

8) References need to be updated. For example, “Zerbi et al., 2020 (Line 414)”± and “Davis et al., 2019 (Line 405)”± could not be located.

In the revised manuscript we have updated the references to (Zerbi et al., 2021) and (Davis et al., 2019).

9) As a reader, I am unfamiliar with the use of the terms “dyadic” and “non-dyadic” in the context used and would appreciate a definition.

By using “dyadic” we meant “monotonic” or “biunivocal”. We reworded the manuscript accordingly as follows:

Discussion: “...consistent with a **monotonic** relationship...”

Discussion: “A non-biunivocal relationship may involve also mTOR-related signaling...”

10) “Synapsis” is misspelled in at least a couple of instances.

We thank the reviewer to point out this typos. In the revised manuscript we have edited as following:

Discussion: “... Mutations in genes affecting synaptic pruning and homeostasis represent a ...”

Discussion: “... that removal of unnecessary synapses is required for the refinement ...”

Discussion: “... Fmr1 mice are characterized by strongly immature synapses ...”

11) Figure 1 – I can’t see the +/- spine figure in A

Thanks for pointing out this omission. We apologize, a technical glitch may have occurred during resubmission #1. The image was however visible in the originally submitted manuscript. The panel illustrating spine density in +/- mice in Figure 1A has now been reinstated.

12) The authors do rsfMRI imaging of mice at P28, a good time for this study. In future work, it would be interesting to do it again after the normal period of spine pruning to see if the connectivity normalizes or persists.

We thank the reviewer for the comment. We do agree that could be of interest for future studies to study rsfMRI connectivity and its link to synaptic pruning at later stages of development. Here we scanned Tsc2 mice at p28, when surplus of spine density was clearly reported in (Tang et al., 2014) and replicated in our tsc2 mice (Figure 1).

Reviewer #3 (Remarks to the Author):

The authors have addressed all of my comments. This is a very nice cross-species study that will be an important contribution to the field. Only one minor comment: Spelling error line 370: 'synapsis' should be 'synapses'

We would like to thank the Reviewer #3 for the constructive comments that have improved our work.

In the revised manuscript we have corrected the typos as following:

Discussion: “... Mutations in genes affecting synaptic pruning and homeostasis represent a ...”

Discussion: "... that removal of unnecessary **synapses** is required for the refinement ..."

Discussion: "... Fmr1 mice are characterized by strongly immature **synapses** ..."

Reviewer #4 (Remarks to the Author):

The authors did a good job in addressing my concerns. In particular, additional fMRI data analyses are performed that incorporate different strategies in handling motion artifacts, a practical problem that is likely inherent in imaging ASD patients, and recapitulate the fMRI finding, which is reassuring. I have no further comments.

We would like to thank the Reviewer #4 for the constructive comments that have improved our work.

References

- Zerbi V, Pagani M, Markicevic M, Matteoli M, Pozzi D, Fagiolini M, Bozzi Y, Galbusera A, Scattoni ML, Provenzano G (2021) Brain mapping across 16 autism mouse models reveals a spectrum of functional connectivity subtypes. *Molecular Psychiatry*:1-11.
- Lieberman OJ, Cartocci V, Pigulevskiy I, Molinari M, Carbonell J, Broseta MB, Post MR, Sulzer D, Borgkvist A, Santini E (2020a) mTOR suppresses macroautophagy during striatal postnatal development and is hyperactive in mouse models of autism spectrum disorders. *Frontiers in Cellular Neuroscience* 14:70.
- Lieberman OJ, Frier MD, McGuirt AF, Griffey CJ, Rafikian E, Yang M, Yamamoto A, Borgkvist A, Santini E, Sulzer D (2020b) Cell-type-specific regulation of neuronal intrinsic excitability by macroautophagy. *eLife* 9:e50843.
- Trakoshis S, Martínez-Cañada P, Rocchi F, Canella C, You W, Chakrabarti B, Ruigrok ANV, Bullmore ET, Suckling J, Markicevic M, Zerbi V, Baron-Cohen S, Panzeri S, Gozzi A, Lai M-C, Lombardo MV (2020) Intrinsic excitation-inhibition imbalance affects medial prefrontal cortex differently in autistic men versus women. *eLife*:2020.01.16.909531.
- Ahtam B, Dehaes M, Sliva DD, Peters JM, Krueger DA, Bebin EM, Northrup H, Wu JY, Warfield SK, Sahin M (2019) Resting-State fMRI Networks in Children with Tuberous Sclerosis Complex. *Journal of Neuroimaging* 29:750-9.
- Davis PE, Kapur K, Filip-Dhima R, Trowbridge SK, Little E, Wilson A, Leuchter A, Bebin EM, Krueger D, Northrup H (2019) Increased electroencephalography connectivity precedes epileptic spasm onset in infants with tuberous sclerosis complex. *Epilepsia* 60:1721-32.
- Gutierrez-Barragan D, Basson MA, Panzeri S, Gozzi A (2019) Infralow State Fluctuations Govern Spontaneous fMRI Network Dynamics. *Current Biology* 29:2295-306.e5.
- Lombardo MV, Lai M-C, Baron-Cohen S (2019) Big data approaches to decomposing heterogeneity across the autism spectrum. *Molecular Psychiatry* 24:1435-50.
- Bertero A, Liska A, Pagani M, Parolisi R, Masferrer ME, Gritti M, Pedrazzoli M, Galbusera A, Sarica A, Cerasa A, Buffelli M, Tonini R, Buffo A, Gross C, Pasqualetti M, Gozzi A (2018) Autism-associated 16p11.2 microdeletion impairs prefrontal functional connectivity in mouse and human Brain 141:2055–65.
- Filipello F et al. (2018) The Microglial Innate Immune Receptor TREM2 Is Required for Synapse Elimination and Normal Brain Connectivity. *Immunity* 48:979-91.e8.

- Liska A, Bertero A, Gomolka R, Sabbioni M, Galbusera A, Barsotti N, Panzeri S, Scattoni ML, Pasqualetti M, Gozzi A (2018) Homozygous Loss of Autism-Risk Gene CNTNAP2 Results in Reduced Local and Long-Range Prefrontal Functional Connectivity. *Cereb Cortex* 10:1-13.
- Li W, Xu X, Pozzo-Miller L (2016) Excitatory synapses are stronger in the hippocampus of Rett syndrome mice due to altered synaptic trafficking of AMPA-type glutamate receptors. *Proceedings of the National Academy of Sciences* 113:E1575-E84.
- Callaway EM, Luo L (2015) Monosynaptic circuit tracing with glycoprotein-deleted rabies viruses. *Journal of Neuroscience* 35:8979-85.
- Di Martino A et al. (2014) The autism brain imaging data exchange: towards a large-scale evaluation of the intrinsic brain architecture in autism. *Mol Psychiatry* 19:659-67.
- Tang G, Gudsnuk K, Kuo SH, Cotrina M-á, Rosoklija G, Sosunov A, Sonders M-á, Kanter E, Castagna C, Yamamoto A, Yue Z, Arancio O, Peterson B-á, Champagne F, Dwork A-á, Goldman J, Sulzer D (2014) Loss of mTOR-Dependent Macroautophagy Causes Autistic-like Synaptic Pruning Deficits. *Neuron* 83:1131-43.
- Ugolini G (1995) Specificity of rabies virus as a transneuronal tracer of motor networks: transfer from hypoglossal motoneurons to connected second-order and higher order central nervous system cell groups. *Journal of Comparative Neurology* 356:457-80.